

# The effect of atmospheric nudging on the stratospheric residual circulation in chemistry-climate models

Andreas Chrysanthou[1], Amanda C. Maycock[1], Martyn P. Chipperfield[1], Sandip Dhomse[1], Hella Garny[2,3], Douglas Kinnison[4], Hideharu Akiyoshi[5], Makoto Deushi[6], Rolando R. Garcia[4], Patrick Jöckel[2], Oliver Kirner[7], Olaf Morgenstern[8], Giovanni Pitari[9], David A. Plummer[10], Laura Revell[11], Eugene Rozanov[12,13], Andrea Stenke[12], Taichu Y. Tanaka[6], Daniele Visioni[14], Yousuke Yamashita[15,16] and Guang Zeng[8]

[1] School of Earth and Environment, University of Leeds, Leeds, UK
[2] Deutsches Zentrum für Luft- und Raumfahrt (DLR), Institut für Physik der Atmosphäre, Oberpfaffenhofen, Germany
[3] Ludwig Maximilians University of Munich, Meteorological Institute Munich, Munich, Germany
[4] National Center for Atmospheric Research (NCAR), Boulder, Colorado, USA
[5] National Institute of Environmental Studies (NIES), Tsukuba, Japan
[6] Meteorological Research Institute (MRI), Tsukuba, Japan
[7] Steinbuch Centre for Computing, Karlsruhe Institute of Technology, Karlsruhe, Germany
[8] National Institute of Water and Atmospheric Research (NIWA), Wellington, New Zealand
[9] Department of Physical and Chemical Sciences, Università dell'Aquila, L'Aquila, Italy
[10] Environment and Climate Change Canada, Climate Research Division, Montréal, QC, Canada
[11] School of Physical and Chemical Sciences, University of Canterbury, Christchurch, New Zealand
[12] Institute for Atmospheric and Climate Science, ETH Zürich (ETHZ), Zürich, Switzerland
[13] Physical-Meteorological Observatory/World Radiation Center, Davos, Switzerland
[14] Sibley School of Mechanical and Aerospace Engineering, Cornell University, Ithaca, NY, USA
[15] Climate Modelling and Analysis Section, Center for Global Environmental Research, National Institute for Environmental Studies, Tsukuba, Japan
[16] Japan Agency for Marine-Earth Science and Technology (JAMSTEC), Yokohama, Japan

**Correspondence**: Andreas Chrysanthou (eeac@leeds.ac.uk)

## Abstract

We perform the first multi-model comparison of the impact of nudged meteorology on the stratospheric residual circulation using hindcast simulations from the Chemistry Climate Model Initiative (CCMI). We examine simulations over the period 1980-2009 from 5 models in which the meteorological fields are nudged towards reanalysis data and compare with equivalent free-running simulations from 9 models. We show that nudging meteorology does not constrain the mean strength of the stratospheric residual circulation and that the inter-model spread is similar, or even larger, than in the free-running simulations. The nudged simulations also simulate stronger upwelling in the tropical lower stratosphere compared to the residual circulation estimated directly from the reanalyses they are nudged towards. Downward control calculations reveal substantial differences between the mean lower stratospheric tropical upward mass flux (TUMF) computed from the modeled wave forcing and that calculated directly from the residual circulation. Although the mean circulation is poorly constrained, the nudged simulations show a high degree of consistency in the interannual variability of the TUMF in the lower stratosphere, which is related to the contribution to variability from the resolved wave forcing. We apply a multiple linear regression (MLR) model to separate the drivers of interannual and long-term variations in the simulated TUMF. The MLR model explains up to ~75% of the variance in TUMF in the nudged simulations and reveals a statistically significant positive trend for most models in TUMF



over the period 1980-2009. Overall, nudging meteorological fields leads to increased inter-model spread

for most of the measures of the mean climatological stratospheric residual circulation assessed in this study. Our findings show that while nudged simulations by construction produce accurate temperatures and realistic representations of fast horizontal transport, this is not necessarily the case for the slower zonal mean vertical transport. Consequently, caution is required when using nudged simulations to interpret long-lived stratospheric tracers that are controlled by the residual circulation.

**1 Introduction**

The Brewer Dobson circulation (BDC) is characterized by upwelling of air in the tropics, poleward flow in the stratosphere, and downwelling at mid and high latitudes. The circulation can be separated into two branches: the shallow branch in the lower stratosphere and the deep branch in the middle and upper stratosphere (Plumb, 2002; Birner and Bönisch, 2011). The BDC affects the distribution of trace

species in the stratosphere, such as ozone, and its strength partly determines the lifetimes of long-lived gases such as chlorofluorocarbons (CFCs; Butchart and Scaife, 2001). It also determines stratosphere to troposphere exchange of ozone (Hegglin and Shepherd, 2009), which is important for the tropospheric ozone budget (Wild, 2007). In the tropical lower stratosphere, where the photochemical lifetime of ozone is long, variations and trends in the strength of the BDC are the main drivers of ozone within the

annual cycle (Weber et al., 2011), for interannual and longer term variability (Randel and Thompson, 2011) and in response to climate change (Keeble et al., 2017). It is important to note that the overall tracer transport in the stratosphere also accounts for the effect of turbulent eddy mixing, which has been evaluated separately in previous studies (Garny et al., 2014; Ploeger et al., 2015). Here we focus on the advective part of the BDC, or the residual circulation, which is driven by wave breaking in the

stratosphere from planetary scale Rossby waves and gravity waves (Holton et al., 1995). The residual circulation is commonly evaluated in model (Butchart et al., 2011) and reanalysis (Abalos et al., 2015; Kobayashi and Iwasaki, 2016) studies using the Transformed Eulerian Mean circulation (TEM; Andrews and McIntyre, 1976, 1978; Andrews et al., 1987).

Past studies have shown substantial spread across models in the mean strength of the residual

circulation (e.g. Butchart et al., 2010). Nevertheless, climate and chemistry-climate models (CCMs) consistently simulate a long-term strengthening of the residual circulation with an increase of ~2% decade$^{-1}$ (e.g. Butchart et al., 2010; Hardiman et al., 2014), though there are differences across models in the relative contribution to trends from resolved and parametrized waves. While subject to multiple caveats, reanalysis datasets also suggest a strengthening of the residual circulation over the past several

decades of the order 2-5% decade$^{-1}$ (Abalos et al., 2015; Miyazaki et al., 2016) apart from one (ERA-Interim) which shows a weakening trend in the deep branch of the BDC (Seviour et al., 2012). Comparing model and reanalysis estimates of BDC variability and trends with observations is challenging since there are no direct measurements of the residual circulation and there are limited





tracer measurements from which age-of-air (AoA) can be estimated (Waugh and Hall, 2002). Engel et
al. (2009) and Stiller et al. (2012) find a statistically significant increase in AoA in the middle
stratosphere at northern midlatitudes, which could suggest the deep branch of the residual circulation is
weakening. In contrast, CCMs forced with observed sea-surface temperatures (SSTs) show a decrease
in AoA throughout the depth of the stratosphere in northern midlatitudes (World Meteorological
Organization, 2018).

In an attempt to obtain a closer comparison with observed stratospheric trace species, some studies
have used model simulations with meteorological fields nudged or relaxed towards a reanalysis dataset
(Jeuken et al., 1996); these include many studies of ozone variability and trends (e.g. van Aalst et al.,
2003; Solomon et al., 2016; Hardiman et al., 2017b; Ball et al., 2018) and the chemical and climatic
effects of volcanic eruptions (Löffler et al., 2016; Solomon et al., 2016; Schmidt et al., 2018). Nudging
involves adding additional tendencies to the model equations to constrain the modeled variables to
reanalysis fields. Nudged variables can include horizontal winds (or divergence and vorticity),
temperature, surface pressure, latent and sensible heat fluxes. However, vertical winds, which are small
residual from horizontal divergence, are not nudged and the underlying model physics can yield quite
different results from the reanalysis that they are nudged towards (Telford et al., 2008; Hardiman et al.,
2017). A recent study by Orbe et al. (2018) analyzed tropospheric tracers in nudged CCM simulations
and found large differences in the distributions of the tracers, which could be partly traced to differences
in the model convection schemes. They urged users to adopt a cautious approach when interpreting
tracers in nudged simulations given their dependence on not only large-scale flow but also sub-grid
parameterizations.

The approach of nudging a CCM towards reanalysis data follows a similar philosophy to traditional
off-line chemical transport models (CTM), although there are fundamental differences between these
types of model in terms of their tracer advection. CTMs need to match the mass transport with the
evolution of the pressure field. This can be done exactly in isobaric coordinates (often used in the
stratosphere) but requires a correction in regions where grid box mass changes (e.g. as surface pressure
changes). CCMs are less affected by this mass-wind inconsistency than CTMs (Jöckel et al., 2001) but
nudging will add forcings which may be inconsistent with the model state. CTMs use the full 3-D
circulation from the (re-)analyses directly and have been widely developed and used over the past few
decades (e.g. Rood et al., 1988; Chipperfield et al., 1994; Lefèvre et al., 1994). They have proven to be
very successful at simulating stratospheric tracers on a range of timescales (Chipperfield, 1999),
including decadal changes (Mahieu et al., 2014). However, this success has been built on extensive
testing of the optimum way to use the reanalysis data to force the CTMs. For example, Chipperfield
(2006) showed how different approaches to calculating the vertical velocity in the
TOMCAT/SLIMCAT model could lead to very different distributions of stratospheric age of air, while
Monge-Sanz et al., (2013a) compared the performance of different European Centre for Medium-Range



Weather Forecasts (ECMWF) analyses within the same CTM framework. Krol et al., (2018) recently provided a summary of how current CTMs intercompare for tracer calculations. Monge-Sanz et al., (2013b) compared the approaches of using ECMWF analyses directly in a CTM with the ECMWF CCM nudged using the same analyses. They found that the CTM and nudged CCM were consistent in showing the degraded performance when using older ERA-40 reanalysis compared to the later ERA-

Interim. However, they also showed some differences between CTM and nudged-CCM tracers using the same analyses, with the nudged CCM showing stronger upward motion in the tropical stratosphere. Recently, Ball et al., (2018) showed 2 nudged CCMs which failed to capture the observed variations in the lower stratospheric ozone as measured by satellite observations, while Chipperfield et al., (2018) using the TOMCAT CTM simulated a better agreement of modeled ozone variations with the

observations. Overall, the success of some CTM simulations in simulating long-lived stratospheric tracers has been built on many years of model development and testing. In contrast, nudged CCMs are much newer tools and have not yet been evaluated to the same extent. Furthermore, with regards to the slow residual circulation, one cannot assume that a nudged CCM will behave in a similar way to a CTM even when using the same meteorological analyses.

To examine the effect of nudging on the stratospheric residual circulation we compare hindcast simulations from free-running and nudged versions of the same models that participated in the phase 1 of the Chemistry-Climate Model Initiative (CCMI; Morgenstern et al., 2017). Nudged experiments were not performed in previous chemistry-climate multi-model comparisons (Chemistry–Climate Model Validation Activity 2; CCMVal-2), so CCMI offers a timely opportunity to evaluate the effect of

nudging on mean biases, variability and long-term trends in the residual circulation. The manuscript is laid out as follows: Section 2 describes the CCMI and reanalysis data used in the present study along with the diagnostics for the residual circulation, section 3 presents results covering the mean circulation, annual cycle, interannual variability and trends, and section 4 summarizes the results and discusses the implications for using nudged simulations to study aspects of the observational record.

## 2 Data and Methods

### 2.1 Models and experiments

CCMI is the successor activity to CCMVal-2 and the Atmospheric Chemistry and Climate Model Intercomparison Project (ACCMIP; Lamarque et al., 2013). We use the hindcast free-running simulations, REF-C1 (C1), and the nudged specified dynamics (SD) simulations, REF-C1SD, which

cover the periods 1960-2010 and 1980-2010, respectively. Both experiments are run with prescribed observed SSTs and sea ice concentrations. The CCMI data are downloaded from the British Atmospheric Data Centre (Hegglin and Lamarque, 2015). We use results from a total of 9 models which differ from one another in various aspects such as their horizontal resolution, which ranges from 1.9° to





5.6°, their vertical resolution as well as their sub-grid parameterizations (see Table 1). For an extensive

overview of the CCMI models see Morgenstern et al. (2017). For the REF-C1 simulations we analyze

between 1-5 ensemble members (depending on what is available) and for REF-C1SD the one realization

submitted from each model. The REF-C1SD simulations nudge meteorological fields such as horizontal

winds and temperature, while the chemical fields are left to evolve freely (Table 1). The models use

different reanalysis fields for nudging taken from ERA-Interim (Dee et al., 2011), JRA-55 (Ebita et al.,

2011; Kobayashi et al., 2015) or MERRA (Rienecker et al., 2011). The differences in the residual

circulation diagnosed from the reanalyses have been identified and documented in previous studies

(Abalos et al., 2015). We analyze those CCMI models that output the necessary TEM diagnostics. At a

minimum, this requires the residual vertical velocity ($\overline{w}^*$) and the residual meridional velocity ($\overline{v}^*$)

(Andrews et al., 1987). Nine models provided these fields and are analysed here (Table 2); where

available we also use the resolved and parametrized wave forcing fields, as documented in Table 2.

**2.2. Model diagnostics**

**2.2.1 TEM residual circulation**

The TEM velocities $(\overline{v}^*, \overline{w}^*)$ are defined as (Andrews et al., 1987):

$$\overline{v}^* = -\frac{1}{\rho_0 \cdot a \cdot \cos\phi}\frac{\partial \overline{\Psi}^*}{\partial z}, \quad \overline{w}^* = \frac{1}{\rho_0 \cdot a \cdot \cos\phi}\frac{\partial \overline{\Psi}^*}{\partial \phi}, \qquad (1)$$


where $\overline{\Psi}^*(\varphi,z)$ is the residual meridional mass streamfunction, $\rho_0$ is the log-pressure density, $\alpha$ is the

Earth radius and $\phi$ is the latitude. As most of the models analyzed here use a hybrid-pressure vertical

coordinate, the primary variable is the pressure vertical velocity, $\overline{\omega}^*$ calculated in $Pa\ s^{-1}$, which must be

converted to $m\ s^{-1}$ in order to get the residual vertical velocity, $\overline{w}^*$. Converting $\omega$ to $w$ is given by the

following equation:

$$\omega = \frac{dp}{dt} = \frac{\partial z}{\partial t}\frac{\partial p}{\partial z} = w\frac{-pg}{RT} = w\frac{-p}{H}, \qquad (2)$$

where $p$ is pressure, $R = 287\ J\ K^{-1}\ Kg^{-1}$ is the gas constant for dry air and $H$ is a fixed scale height. Both

TEM velocity components were submitted as monthly mean fields to the CCMI data archive (Hegglin

and Lamarque, 2015). Upon close examination of the CCMI model output, some discrepancies were

found in the way that the residual vertical velocity was calculated among the models. Although a fixed

scale height of $H = 6950\ m$ was recommended in the CCMI data request, the model output from some

models (EMAC and SOCOL) was calculated incorrectly using a temperature-dependent density, $\rho_0 =$

$p/RT$, instead of the log-pressure density. This methodological error leads to artificial spread in the

model $\overline{w}^*$ fields (see Dietmüller et al., 2018). We note that previous multi-model comparisons of the

residual circulation that use $\overline{w}^*$ taken directly from models may have been subject to the same issue,





though we cannot confirm this (e.g. Butchart et al., 2010; SPARC, 2010). To avoid this methodological inconsistency, Dietmüller et al., (2018) recalculated $\overline{w}^*$ from $\overline{v}^*$ using the continuity equation, which consists of a vertical integration and a derivative along the meridional direction. The recalculation of $\overline{w}^*$ from $\overline{v}^*$ has also been explored for this study, but it was found to introduce additional errors affecting

the latitudinal structure of $\overline{w}^*$ (not shown), specifically because of the reduced number of CCMI requested pressure levels compared to the native model levels. We were able to overcome this discrepancy for the EMAC simulations by converting $\overline{\omega}^*$ to $\overline{w}^*$ as in equation 2, using the log-pressure density. However, for SOCOL3 this was not possible and hence the absolute values for that model should be treated with caution. For the models other than EMAC, the results presented in this study are

based on the original diagnostics submitted to the CCMI data archive.

We also compute the mass flux across a given pressure surface as (Rosenlof, 1995):

$$2\pi \int_{\phi}^{pole} \rho_0\, a^2 cos\phi\, \overline{w}^* d\phi = 2\pi a \Psi(\phi), \quad (3)$$

using the boundary condition that $\Psi = 0$ at the poles. By finding at each pressure level the latitude at which $\Psi_{max}$ and $\Psi_{min}$ occur, which correspond to the height-dependent turnaround (TA) latitudes, we

can calculate the net downward mass flux in each hemisphere. The net tropical upward mass flux, equal to the sum of the downward mass fluxes in each hemisphere, can then be expressed as (Rosenlof, 1995):

$$Tropical\ Upward\ Mass\ Flux\ (TUMF) = 2\pi a\ (\Psi_{max} - \Psi_{min}) \quad (4)$$

**2.2.2 Downward control principle calculations**

Under steady-state conditions, the $\overline{\Psi}^*(\varphi,z)$ at a specified latitude $\varphi$ and at a log(pressure)-height $z$ is given by the vertically integrated eddy-induced total zonal forces $\overline{F}$ above that level (Haynes et al., 1991):

$$\Psi(\varphi, z) = \int_{z}^{\infty} \left\{ \frac{\rho_0 a^2 \overline{F} cos^2\phi}{\overline{m}_\phi} \right\}_{\phi=\phi(z')}, \quad (5)$$


where in the quasi-geostrophic limit $\overline{m}_\phi \approx -2\Omega a^2 sin\phi cos\phi$. The above integration occurs along lines of constant zonal mean absolute angular momentum $\overline{m}=acos\phi(\overline{v}+a\Omega cos\phi)$ where $\overline{v}$ is the zonal mean zonal wind and $\Omega$ corresponds to Earth's rotation rate with boundary conditions of $\Psi \rightarrow 0$ and $\rho_0 \overline{w}^* \rightarrow 0$ as $z \rightarrow \infty$. These lines of constant angular momentum are almost vertical apart from near the equator (up to

$\sim \pm 20°$) such that we can approximate the solution of the above integral using constant $\varphi$ for the limits of the integral. In climate model simulations, $\overline{F}$ corresponds to contributions from resolved waves due



to the divergence of Eliassen-Palm flux (EPF) and/or contributions from parameterized gravity wave drag due to sub-grid scale waves that originate from orography, convection and frontal instabilities. This enables us to estimate the contribution of both resolved planetary wave driving (EPFD) along with

the orographic (OGW) and non-orographic (NOGW) parameterized gravity wave drag from the CCMI model output (Table 2) to the tropical upward mass flux and compare with the direct estimates from the residual vertical velocity $\overline{w}^*$.

Applying the downward control principle (DCP) can provide useful insights into the driving mechanisms of the stratospheric residual circulation and therefore explain part of the inter-model spread

found in both REF-C1 and REF-C1SD simulations. While the DCP enables the contributions of EPFD and OGW/NOGW to TUMF to be calculated under various assumptions (Haynes et al., 1991), one has to keep in mind that the different wave forcings can interact and thus are not independent of each other (Cohen et al., 2013).

**2.3 Multiple linear regression model**

To investigate the drivers of interannual variability in the residual circulation we apply a multiple linear regression (MLR) model (equation 6) to the annual mean TUMF. The model includes terms for known drivers of variations in tropical lower stratospheric upwelling: major volcanic eruptions (Pitari and Rizi, 1993), El Nino Southern Oscillation (ENSO) (García-Herrera et al., 2006; Marsh and Garcia,

2007; Randel et al., 2009), the Quasi-Biennial Oscillation (QBO) (Baldwin et al., 2001) and a long-term linear trend (Calvo et al., 2010).

$$TUMF(t) = \beta_0 + \beta_{VOL} \cdot x_{VOLC}(t) + \beta_{ENSO} \cdot x_{ENSO}(t) + \beta_{TREND} \cdot x_{TREND}(t) + \beta_{QBO1} \cdot x_{QBO1}(t) + \beta_{QBO2} \cdot x_{QBO2}(t) + \varepsilon(t),$$
$$(6)$$

where $\beta_0$ is a constant, $\beta_i$ is the regression coefficient for basis function $x_i$ and $\varepsilon(t)$ is the residual. Following Maycock et al., (2018), the volcanic basis function is defined as the tropical lower stratospheric volcanic surface area density (SAD), the ENSO term is based on east-central equatorial Pacific Ocean SST timeseries (Niño 3.4 index; 5°S to 5°N; 170°W to 120°W), the two orthogonal QBO terms as the first two principal components from an empirical orthogonal function (EOF) analysis on

the zonal mean zonal winds between 10°S-10°N and 70 to 5 hPa, and a linear trend. The first three regressors, volcanic, ENSO and the linear trend are identical for both REF-C1 and REF-C1SD runs while the QBO terms are calculated using the model winds for each experiment. For the REF-C1 runs, CMAM does not include a QBO, hence when we apply the MLR to the CMAM REF-C1 simulation the QBO terms are omitted. We opted not to include an equivalent effective stratospheric chlorine (EESC)

MLR term to account for the ozone depleting substances changes (Morgenstern et al., 2018; Polvani et al., 2018) as the period considered in the study is not sufficiently long for the linear trend to be





separated properly from EESC. Since we are regressing annual mean TUMF we do not consider a seasonal cycle term or any lag in the terms. The results in section 3.5 focus on the first ensemble member (in the rip-nomenclature, where r stands for realization, i for initialization and p for physics-

r1i1p1), but where applicable the results from the MLR model for the rest of the ensemble members of the REF-C1 runs are presented in the supplement (Supplement Figures S7-S12).

**2.4 Reanalysis Data**

In order to compare the REF-C1 and REF-C1SD simulations against the reanalysis datasets used for

the nudging, we use the SPARC Reanalysis Intercomparison Project (S-RIP) dataset (Martineau et al., 2018). This provides a common gridded version of the reanalysis TEM fields on a 2.5° × 2.5° grid up to 1 hPa. The pressure vertical velocity, $\overline{\omega}^*$, is converted to the residual vertical velocity, $\overline{w}^*$, using equation 2. A detailed comparison of the stratospheric residual circulation in reanalysis datasets has been given by Abalos et al. (2015).

**3 Results**

**3.1 Climatological residual circulation**

Figure 1 shows a latitude-pressure cross-section of the climatological (1980-2009) multi-model mean (MMM) annual mean $\overline{w}^*$ for the REF-C1 (Fig. 1a) and REF-C1SD (Fig. 1b) simulations and their absolute differences (Fig. 1c). In Figure 1c, positive values indicate where the magnitude of the

circulation in REF-C1SD (whether upwelling or downwelling) is larger than in REF-C1. As expected, upwelling occurs in the tropics between around 30°S to 30°N and downwelling at higher latitudes. Within the region of tropical upwelling, the REF-C1SD runs generally show larger $\overline{w}^*$ values in the low-to-mid stratosphere. In the upper stratosphere between ~4-2 hPa, REF-C1 shows a slightly narrower and more peaked region of upwelling than REF-C1SD, and vice versa near 1 hPa. In the mid-

latitudes, between ~30-50°, the REF-C1SD runs exhibit on average slightly stronger downwelling than in the REF-C1 simulations except in the upper stratosphere. In the polar and subpolar regions of both hemispheres throughout the stratosphere, the picture reverses as the REF-C1 runs show stronger downwelling over the poles and weaker downwelling near the edge of the polar region (60-70°). We note these differences are rather similar when the MMM for the REF-C1 simulations is calculated using

only the same 6 simulations as used in the REF-C1SD calculation (Supplement Figure S1) with some differences in the tropical upper stratosphere within the upwelling region (not shown). Additionally, the climatological TA latitudes, when calculated for the above-mentioned subset of REF-C1 MMM, reveal significant differences throughout the depth of the stratosphere when compared with the MMM for the REF-C1 and REF-C1SD runs respectively (Supplement Figure S2). Specifically, the REF-C1SD runs



simulate on average a wider Northern Hemisphere (NH) TA latitude throughout almost all the depth of the stratosphere than the REF-C1 runs while in the Southern Hemisphere (SH) TA latitude, the picture reverses in the middle and upper stratosphere (Supplement Figure S2). Hence, nudging meteorology significantly affects the strength and structure of the climatological residual circulation throughout the stratosphere.

Focusing on the lower stratosphere, Figure 2 shows the climatological annual mean $\overline{w}^*$ at 70 hPa in the individual models for the REF-C1 and REF-C1SD simulations and their difference. Also plotted in Fig 2b are $\overline{w}^*$ estimates from the three reanalysis datasets used by the nudged models. The nudged runs on average show a wider region of tropical upwelling especially in the lower stratosphere compared to their free-running counterparts (Supplement Figure S2). Within the upwelling region, the vast majority of the

models show a clear double peaked $\overline{w}^*$ structure in the tropics, with ULAQ-CCM being a notable outlier as it exhibits a rather different and single-peaked maximum and NIWA-UKCA showing a relative flat and not clearly defined structure. Both EMAC simulations, along with CMAM and SOCOL3 models, show a narrower double-peaked structure with EMAC-L47 exhibiting a rather pronounced NH local maximum. The remainder of the REF-C1 simulations exhibit a double-peaked structure as well, but to a

certain degree more symmetric, albeit with various amplitudes. CESM1-WACCM simulates a broader double-peaked structure with the SH local maximum occurring at higher latitudes compared with the rest of the models. A double-peaked $\overline{w}^*$ structure in the lower stratosphere has previously been shown in reanalysis datasets (Abalos et al., 2015; Ming et al., 2016) and some CCMs (Butchart et al., 2006, 2010). This can also be seen in Figure 2b for the three reanalysis datasets (ERA-I, JRA-55 and

MERRA), where ERA-I and JRA-55 show an asymmetrical double-peaked structure with stronger upwelling in the NH local maximum. As documented by Abalos et al. (2015), based on the direct calculation from the residual circulation definition, MERRA exhibits downwelling at the equator, an issue which was highlighted and linked with a negative cell in the streamfunction.

   Figure 2c shows the differences in 70 hPa $\overline{w}^*$ for the 6 models that ran both experiments. As in Figure

1c positive differences indicate that the magnitude of the circulation in REF-C1SD is larger than in REF-C1. The largest differences are found within the inner tropics, where CCSRNIES-MIROC3.2 and CMAM exhibit significantly stronger upwelling (up to 3 times more for CMAM) near the local $\overline{w}^*$ minimum at the equator. Similarly, for MRI-ESM1r1, which was nudged towards JRA-55, the differences near the equator reflect a reduction in the double-peaked structure of $\overline{w}^*$ in the REF-C1SD

experiment. Similarly, EMAC-L90 differences are a manifestation of having a more double-peaked structure in their REF-C1 simulation. EMAC-L47 and CESM1-WACCM although nudged towards ERA-I and MERRA, respectively, show similar and relatively small differences between REF-C1SD and REF-C1 both within the upwelling region and the downwelling regions in both hemispheres. REF-C1 shows stronger downwelling in the mid-latitudes (up to 50° in both hemispheres) compared to the



nudged models. In the subpolar and polar latitudes of the SH, the majority of the nudged models
simulate stronger downwelling than their free-running counterparts, while in the NH no consistent
picture emerges.

   A key result from Figure 2 is that the inter-model spread in $\overline{w}^*$ for both experiments is larger in the NH
downwelling region than in the equivalent region of the SH. Specifically, the inter-model spread is 0.15
mm s$^{-1}$ for the REF-C1 runs between 30°S - 80°S and 0.22 mm s$^{-1}$ between 30°N - 80°N, while for
REF-C1SD the values are 0.11 mm s$^{-1}$ and 0.18 mm s$^{-1}$, respectively. This also demonstrates that the
REF-C1 simulations exhibit slightly larger inter-model spread in $\overline{w}^*$ in the extratropics and is also true
when taking into account the same 6 simulations as used in the REF-C1SD experiment. In contrast, in
the tropics between 30°S - 30°N the REF-C1SD simulations exhibit a larger inter-model spread than the
free-running simulations (0.09 mm s$^{-1}$ vs. 0.07 mm s$^{-1}$). All the above show that nudging, as applied in
these simulations, rather weakly constrains the mean amplitude and structure of the residual circulation.

   Comparing the SD models against the reanalysis datasets they are nudged towards reveals some
distinct differences. For example, CESM1-WACCM is nudged towards MERRA as shown in Figure 2b,
but it does not show downwelling around the equator as the direct MERRA estimate, while it does
resemble the reanalysis in the northern subtropics quite closely. MRI-ESM1r1 in the REF-C1SD
experiment is nudged towards JRA-55 and shows a rather flat latitudinal $\overline{w}^*$ structure with no clearly
defined local maxima and does not resemble the reanalysis in the inner tropics. As for the models
nudged towards ERA-I, CCSRNIES-MIROC3.2 simulates a largely dissimilar structure to the
reanalysis, exhibiting a single-peaked structure without capturing the tropical maxima asymmetry,
which is a feature of EMAC-L47 and CMAM simulations of the REF-C1SD simulations to a lesser
degree. In summary, nudging does not strongly constrain the mean strength of the residual circulation
and in most cases the REF-C1SD simulations exhibit some notable differences compared to the residual
circulation directly estimated from the reanalysis that they were nudged towards.

**3.2 Tropical upward mass flux climatology**

Figure 3 shows the vertical profiles of the climatological TUMF from 100 hPa to 3 hPa, calculated
from annual means of $\overline{w}^*$. In most cases (apart from EMAC-L47 and EMAC-L90), the REF-C1SD runs
simulate stronger TUMF than the equivalent free-running REF-C1 simulations up to 10hPa (Figure 3c).
In the lower stratosphere between 100-30 hPa, the largest differences in TUMF are found in the CMAM
and CCSRNIES-MIROC3.2 nudged simulations, which show larger values at 90 hPa by more than 20%
and 15%, respectively. In CESM1-WACCM, the differences in TUMF between the REF-C1SD and
REF-C1 simulations are generally positive throughout the stratosphere and reach 20% at 20 hPa and
~25% at 3 hPa. The MRI-ESM1r1 model generally shows the smallest differences in TUMF between
the two experiments. In the upper stratosphere (above 10 hPa) the picture is mixed as half of the models
show higher TUMF in the nudged experiments, including EMAC-L47 and EMAC-L90 which show





opposite sign differences at higher pressures. Compared to the reanalysis that they were nudged towards (Figure 3b), three out of the four REF-C1SD simulations that were nudged towards ERA-I (CCSRNIES-MIROC3.2 and EMAC-L47/L90) simulate stronger upwelling than the reanalysis in the upper stratosphere, with differences reaching up to 50%. This could be partly due to the fact that both the CCMI and S-RIP fields have been interpolated from their native model levels to a set of predefined

common pressure levels which are rather sparse in that vicinity, hence the TUMF calculation could be different if it was performed on the native model grid of both CCMI models and the reanalysis. Additionally, for both EMAC REF-C1SD simulations the differences from the reanalysis in the TUMF above 10 hPa can also be explained by the fact that the nudging is only imposed strongly up to 10 hPa, while higher model layers have weakening nudging coefficients as they serve as transition layers.

CESM1-WACCM generally shows slightly larger TUMF values than MERRA apart from the upper stratosphere where they start to converge. MRI-ESM1r1 exhibits relatively better agreement of TUMF with JRA-55 throughout the depth of the stratosphere, while CMAM which is nudged towards ERA-I follows closely the reanalysis especially above 10 hPa but is generally biased low. In terms of the spread found in both sets of experiments, in the lower stratosphere (100-30 hPa) where the maximum

spread is located, the nudged simulations and their free-running counterparts show a total spread of $3.26 \times 10^9$ kg s$^{-1}$ and $3.1 \times 10^9$ kg s$^{-1}$ respectively, though note there are fewer REF-C1SD simulations to consider. Again, a key message is that the nudged REF-C1SD simulations show if not a slightly larger, a comparable spread in the climatological TUMF compared to the free-running REF-C1 simulations, throughout almost all the depth of the stratosphere.

To understand the dynamical factors that contribute to the modelled residual circulation and its spread, Figure 4 shows TUMF at 70 hPa along with the downward control calculations (section 2.2.2) to quantify the contribution of resolved and parameterized wave forcing to the TUMF. The black bars on the left show the TUMF diagnosed from $\overline{w}^*$ and the grey bars on the right show the estimated contribution to TUMF from the Eliassen-Palm flux divergence (EPFD, dark grey), the orographic (mid-

grey) and non-orographic (light grey) gravity wave drag. Note that SOCOL3 and ULAQ-CCM did not provide any wave forcing fields and NIWA-UKCA only provided the EPFD.

   In the free-running REF-C1 simulations (Fig. 4a), the estimated TUMF from the total wave forcing for the majority of the models (apart from CESM1-WACCM and EMAC-L90), exceeds the TUMF calculated directly from $\overline{w}^*$; this was not the case though for the CCMVal-2 models in SPARC (2010)

(Figure 4.10, p.121). Comparing that figure with the results in Figure 4a, the MMM TUMF ($5.9 \times 10^9$ kg s$^{-1}$) for the ten REF-C1 model simulations analysed here is in very close agreement when compared to the MMM of the fourteen CCMVal-2 models ($5.8 \times 10^9$ kg s$^{-1}$) (SPARC, 2010). In terms of the contribution of the resolved wave forcing to the TUMF, there appears to be a decreased model range ($3.26 - 5.33 \times 10^9$ kg s$^{-1}$) in the present study compared with the CCMVal-2 models ($1.5 - 5.5 \times 10^9$ kg s$^{-1}$)

(SPARC, 2010). Some CCMI models have increased their horizontal resolution by up to a factor of two



(CMAM, MRI-ESM1r1, SOCOL3 and ULAQ-CCM) and also their vertical resolution up to 80 vertical levels (MRI-ESM1r1) compared with CCMVal-2 models (Dietmüller et al., 2018), which could improve their ability to simulate resolved wave forcing. There is a notable feature of CMAM which shows that the NOGWD contributes negatively to TUMF (indicated with two red horizontal lines on

Figure 4 and Supplement Figure S3); this was also found for CMAM in CCMVal-2 (Figure 4.10; SPARC, 2010).

The MMM TUMF at 70 hPa in the REF-C1SD simulations (Figure 4b) is $6.3 \times 10^9$ kg s$^{-1}$. Interestingly, for the single simulations that were nudged towards MERRA and JRA-55 (CESM1-WACCM and MRI-ESM1r1, respectively) the TUMF in the REF-C1SD runs is closer to the estimates from the reanalyses

they are nudged towards. This may simply be a coincidence given that there remain substantial differences in the structure of $\overline{w}^*$ between the REF-C1SD simulations and reanalyses (Figure 2b) and this does not apply necessarily for all models that were nudged towards ERA-I. Another notable feature is that the contribution from the individual and total wave forcing contributions shows reduced inter-model spread in the REF-C1SD simulations (Figure 4b, grey bars). For example, the inter-model

standard deviation of the EPFD contribution to TUMF at 70 hPa is 43% smaller than in REF-C1 ($0.43 \times 10^9$ kg s$^{-1}$ and $0.76 \times 10^9$ kg s$^{-1}$, respectively). However, the residuals (i.e. the difference between the directly calculated TUMF and the total downward control estimated contribution from the wave forcing) are substantially larger and positive (except for EMAC-L90) in the REF-C1SD experiment than in REF-C1. The lower TUMF calculated directly from $\overline{w}^*$ in EMAC-L90 compared to EMAC-L47 is

consistent with the results of Revell et al. (2015b) who also find that an increase in the model vertical resolution for SOCOL3 results in a slow-down of the BDC. Comparison of the TUMF at 10 hPa for the REF-C1SD experiment (see Supplement Figure S3b), reveal that the residuals are smaller in the middle stratosphere and as stated previously for both EMAC simulations, 10 hPa is the maximum level that the nudging is applied. Nevertheless, the rest of the REF-C1SD models were nudged even above that level

also show smaller residuals and this may indicate differences in the effect of nudging on the shallow versus the deep branch of the circulation (Birner and Bönisch, 2011). In summary, the results reflect the fact that nudging imparts an external and non-physical tendency in the model equations, which in turn might cause violations of the normal constraints on the global circulation, such as conservation of momentum and energy. This alters the residual circulation and appears to limit the ability to close the

circulation through the integrated wave forcing as would ordinarily apply in the downward control principle (Haynes et al., 1991).

### 3.3 Annual cycle

We now evaluate the representation of the annual cycle in the residual circulation. Figure 5 shows the

MMM climatological annual cycle of $\overline{w}^*$ at 70 hPa for the REF-C1 and REF-C1SD simulations and





their difference. Note there are no significant variations in the results when the MMM of the REF-C1 experiment is computed only for those models used in the REF-C1SD analysis (Supplement Figure S4). Both experiments show similar broad features in the annual cycle, with stronger tropical upwelling in boreal winter, a latitudinal asymmetry in the region of upwelling with the turnaround latitude being

further poleward in the summer hemisphere, and stronger downwelling over the winter pole. These features resemble the annual cycle found in other multi-model studies (e.g. Hardiman et al., 2014). Figure 5c shows that on average the nudged models simulate stronger upwelling in the sub-tropics, particularly in the NH in boreal winter with a few exceptions; the most prominent one being the narrow band between the equator and 10°N where the REF-C1 simulations exhibit stronger upwelling in austral

winter. Consequently, the nudged models simulate substantially stronger downwelling in the midlatitudes in winter. In the NH mid-latitudes in the summer months nudged runs show stronger downwelling, which reverses for the SH mid-latitudes in the austral winter. At polar latitudes there is a distinct seasonality to the difference between the REF-C1SD and REF-C1 simulations, with the nudged models simulating stronger downwelling in winter and weaker downwelling in the Arctic during the rest

of the year, corresponding to an amplified annual cycle. Conversely in the Antarctic, the REF-C1SD simulations generally simulate weaker downwelling, particularly during austral summer and spring.

To compare the annual cycle in residual circulation across the models, Figures 6a and 6b show the mean tropical (30°N-30°S) $\overline{w}^*$ at 70 hPa for the REF-C1 and REF-C1SD simulations, respectively. In general, the amplitude and phasing of the annual cycle is rather weakly constrained across the REF-

C1SD simulations with the intermodel spread of amplitude as measured by the standard deviation being around 20% higher (0.062 mm s$^{-1}$ vs. 0.05 mm s$^{-1}$) across all months compared to the REF-C1 runs. Comparing the MMM annual cycle of the REF-C1 runs with the MMM REF-C1SD of Figure 6b reveals that on average the SD models show a slightly larger peak-to-peak amplitude in the annual cycle. The difference between the REF-C1SD runs and the respective reanalysis they are nudged

towards are generally larger in boreal winter than in boreal summer.

Figures 6c and 6d show the climatological annual cycle in the turnaround latitudes at 70 hPa for the REF-C1 and REF-C1SD runs, respectively. The inter-model spread in the REF-C1SD runs is also slightly higher in this metric (+6% for the NH TA latitude), as seen from Figure 6d, while the reanalyses also exhibit some differences. Note for this measure we exclude ULAQ-CCM when

calculating the inter-model spread for the REF-C1 runs because it lacks a realistic seasonal evolution in its TA latitudes. The REF-C1SD runs show a weaker annual cycle in the SH TA latitude, with the models showing a consistently more poleward TA latitude in the SH in austral winter (JJA). In the NH, the TA latitudes in the REF-C1SD runs show a slightly smaller annual cycle compared to the REF-C1 runs. In summary, the amplitude and phasing of the annual cycle in tropical lower stratospheric

$\overline{w}^*$ appears to be more consistent across the nudged REF-C1SD simulations compared to the free-running REF-C1. To summarize, there is substantial inter-model spread in the turnaround latitudes and



the amplitude of the annual cycle in both sets of simulations which is, if anything, higher in the nudged simulations than in the free running models.

**3.4 Interannual variability of the tropical upward mass flux**

Figure 7 shows timeseries over 1980-2009 for the annual, December-January-February (DJF), and June-July-August (JJA) mean TUMF at 70 hPa for the REF-C1 (left column) and REF-C1SD (right column) simulation. As expected, the TUMF is larger in DJF compared to the annual and JJA means in both the REF-C1 and REF-C1SD runs because the average tropical upwelling is stronger. The REF-

C1SD simulations show remarkably similar temporal variability in contrast to REF-C1 where the modeled interannual variability is very diverse despite the models all being forced with observed SSTs. Hence, although nudging does not constrain the mean TUMF in the lower stratosphere, it does constrain the interannual variability; this is even more apparent for the DJF and JJA seasonal means. Additionally, the REF-C1SD runs exhibit a close agreement in their temporal variability to the

reanalysis they were nudged towards, albeit with magnitude differences and different trends in the beginning of the 21$^{st}$ century where ERA-I and MERRA show a negative behaviour. Nevertheless, the inter-model spread is higher for the REF-C1SD runs especially in the annual means for the whole time period of the TUMF timeseries in the lower stratosphere, with the inter-model spread for the nudged experiment being 25% higher (standard deviation of $0.54 \times 10^9$ kg s$^{-1}$ vs. $0.4 \times 10^9$ kg s$^{-1}$ respectively)

than their free-running counterparts.

To investigate the cause of the high temporal correlation of the REF-C1SD annual mean TUMF timeseries, in Figure 8 we present the TUMF anomalies at 70 hPa along with the contributions to the interannual variations in TUMF from EPFD, OGWD, NOGWD and the total parameterized wave forcing (from top to bottom panels) for REF-C1 (left column) and REF-C1SD (right column),

respectively. Figure 8b shows the remarkably similar temporal variability in TUMF across the REF-C1SD runs which can be contrasted against the weak interannual coherence in the REF-C1 runs (Figure 8a). Figure 8d and 8j show that both the EPFD and the total parametrized wave forcing contributions to the TUMF show a high degree of temporal coherence in the SD simulations. This result indicates that although nudging does not constrain the mean residual circulation, it does constrain the interannual

variability through influencing both the resolved and parametrized waves. It should be noted that the reanalyses have been shown to exhibit strong similarities in their resolved EP fluxes as shown by the linear correlation in the timeseries of tropical upwelling at the 70 hPa level when considering the momentum balance estimates of $\overline{w}^*$ (Abalos et al., 2015). Nonetheless, nudging only u, v (in most cases) and T (in some cases) in the SD models leads to very similar inferred contributions from resolved

and parametrized wave forcing to the TUMF although these fields are not directly nudged. The fact that the individual OGW and NOGW terms do not show such a strong inter-model agreement, while the





total parametrized wave forcing does, could suggest there is some compensation between the resolved and parameterised wave forcing occurring (e.g. Cohen et al., 2013). The REF-C1 simulations show a highly variable pattern of the TUMF anomalies contributed by EPFD and OGWD (Figures 8c, 8e),

despite the fact they use the same observed SSTs and some nudge the QBO (CCSRNIES-MIRCO3.2, CESM1-WACCM, EMACL47/L90, SOCOL3 and ULAQ-CCM). In summary, the source of the remarkably coherent interannual variability in the annual TUMF timeseries in the REF-C1SD simulations is due to the interannual variability of both the resolved and parametrized wave forcing being constrained more tightly than the climatological strength of the TUMF (Figure 4b). The reasons

for the difference in the effect of nudging on the behaviour of the residual circulation between the long-term mean and interannual variability is unclear.

### 3.5 Multiple Linear Regression analysis

Figures 9 and 10 present timeseries of annual TUMF anomalies at 70 hPa attributed to each of the

basis functions in the MLR model described in section 2.3 and the residuals for the REF-C1 and REF-C1SD runs, respectively. Figure 9a shows a large spread in the diagnosed signal of volcanic eruptions in the TUMF timeseries. The majority of the REF-C1 simulations analyzed here show a negative TUMF anomaly around the time of the El Chichón (1982) and Mount Pinatubo (1991) eruptions, a result that can be affected by how well aerosol heating is represented in the various models. Consequently, it

highlights that in a free-running climate simulation, internal variability can be larger than the response to a transient forcing. This is further demonstrated by the range in the amplitude of the volcanic regressor among different ensemble members from the same model (see Supplement Figures S7-S12). In the REF-C1SD runs (Figure 10), most models show a positive anomaly in TUMF attributed to volcanic eruptions, consistent with earlier studies, but still with a considerable range in amplitudes with

MRI-ESM1r1 and CESM1-WACCM showing the largest volcanic responses. The reason that the EMACL47/L90 models show a negative TUMF anomaly to volcanic forcing in both REF-C1 and REF-C1SD runs is documented in Appendix B4 of Morgenstern et al. (2017) and relates to a unit conversion error where the extinction of stratospheric aerosols was too low, by a factor of ~500, hence the stratospheric dynamical effects of those eruptions were not represented (Jöckel et al., 2016). There are

differences amongst models in the amplitude of the variance in the TUMF attributed to ENSO and the linear trend, but they are all consistent in sign (i.e. positive TUMF anomaly for El Nino and positive long-term trend). Despite the inter-model spread, for each model that performed both REF-C1 and REF-C1SD experiments the ENSO and linear trend contributions to the TUMF anomalies are quite similar, although the magnitude varies. As expected, the variations in TUMF attributed to the QBO are quite

different in the REF-C1 and REF-C1SD runs for those models that do not nudge the QBO in REF-C1, as shown in Figures 9 and 10. The nudging of zonal winds in REF-C1SD constrains the phase of the





QBO, and hence there is strikingly similar variability in the TUMF anomalies attributed to the QBO in the REF-C1SD runs. The linear trend coefficient is statistically significantly different from zero at the 95% confidence level in five out of the six REF-C1SD models over the period 1980-2009, in contrast to

being significant in half of the REF-C1 simulations. The $R^2$ values for the REF-C1 simulations vary between 0.16 (CMAM) and 0.67 (CESM1-WACCM). REF-C1SD runs generally give more consistent $R^2$ values across the models ranging from 0.62 (CCSRNIES-MIROC3.2) to 0.77 (EMAC-L47). This means there is still a substantial fraction (>23%) of unexplained variance in the annual TUMF timeseries in the REFC1-SD simulations after applying the MLR model, which exhibits a remarkable

degree of temporal correlation. In contrast, the MLR residuals in the REF-C1 runs (Figure 9f) show much less temporal coherence apart from a drop around 1989, which is also apparent in the REF-C1SD runs (bottom panel Figure 10). In contrast, the residuals in the REF-C1SD simulations (Figure 10f) show a high degree of coherent interannual variability, another manifestation of the fact that the nudged runs do reproduce a much more consistent inter-annual variability. This makes a substantial

contribution to the coherence of the TUMF timeseries in Figure 8b, but it cannot be attributed to any of the terms included in the MLR model.

### 3.6 Trend sensitivity analysis

Following from the results of the MLR analysis in section 3.5, which showed a statistically significant

linear trend in some models for the 30-year period 1980-2009, we now explore the sensitivity of the linear trend to the time period considered. We apply the same MLR model as discussed in section 3.5 to the annual mean 70 hPa TUMF timeseries of the first ensemble member for both REF-C1 and REF-C1SD runs, but systematically varying the start and end dates to cover all time periods in the window 1980-2009 that are at least ten years in length. We then extract the linear trend coefficient and its

associated p-value. Figures 11 and 12 present the linear trend calculations for the REF-C1 and REF-C1SD runs, respectively, as a function of trend start and end date. Statistically significant trends at the 95% confidence level are marked with black stippling.

None of the periods considered in either the REF-C1 or REF-C1SD experiments shows a significant negative trend. A statistically significant positive trend emerges in almost all of the SD models for

trends beginning in the mid-1980s to early 1990s extending to the mid-2000s. The trends are mainly significant for periods of 20 years or more and no less than around 12 years. This result broadly corroborates the findings of Hardiman et al. (2017a) who used a control run to estimate the period required to detect a BDC trend with an amplitude of 2% per decade trend against the background internal variability. There is range of different structures in the diagnosed trends among models,

particularly for the REF-C1 simulations where a consistent pattern of positive trends only emerges across most models for the entire time period. This is because of the differences in internal variability



amongst the models that can mask BDC trends over short periods. However, the REF-C1SD runs simulate more consistent variations in trends as a function of time period but generally show weaker positive trends in lower stratospheric TUMF than their free-running counterparts. It should also be noted that any trend combination starting around the end of 1990s in almost all cases of both REF-C1 and REF-C1SD runs exhibit no statistical significance possibly pointing towards the role of declining ozone depleting substances (ODS; Polvani et al., 2018) due to the implementation of the Montreal Protocol.

## 4. Conclusions

This study provides the first multi-model comparison of the impact of nudged meteorology on the representation of the stratospheric residual circulation. We use hindcast CCMI runs with identical prescribed natural forcings in two configurations: REF-C1SD with meteorological fields nudged towards reanalysis data (specified dynamics, SD) and REF-C1 that is free-running over the period 1980-2009. The nudged simulations use various reanalysis datasets, nudge different variables (u, v, T, vorticity, surface pressure), and use different time constants to impose the additional nudging tendencies in the model equations. The key findings of this study are:

1. Nudging large-scale meteorology does not strongly constrain the mean strength of the residual circulation compared to free-running model simulations. In fact, for most of the metrics of the climatological mean residual circulation examined, including residual vertical velocities and mass fluxes, the inter-model spread for fewer simulations is comparable or even larger in the REF-C1SD runs than in the free-running REF-C1 simulations.

2. Nudging generally leads to REF-C1SD runs simulating slightly stronger upwelling and do not quite resemble the direct estimates from the reanalysis they were nudged towards.

3. In the nudged simulations there are larger differences among the directly simulated tropical upward mass flux in the lower stratosphere compared to their free-running counterparts which can be explained from the diagnosed wave forcing using the downward control principle (Haynes et al., 1991). Nevertheless, the spread in the contributions from the resolved and parametrized wave forcing to the tropical mass flux is reduced in the nudged simulations.

4. Despite the lack of consistency in the mean circulation, nudging tightly constrains the interannual variability in the tropical upward mass flux in the lower stratosphere compared to the free-running simulations. This is associated with constraints to the contributions from both the resolved and parametrized wave forcing despite the fact the models use different reanalysis datasets for nudging.

5. A multiple linear regression analysis showed that up to 77% of the interannual variance in the tropical upward mass flux timeseries in the lower stratosphere can be explained by factors



including volcanic eruptions, ENSO, the QBO and a linear trend. The remaining unexplained variance shows a high degree of a temporal covariance amongst models in the nudged simulations but not for the free-running simulations.

6. Nudged simulations show a statistically significant positive linear trend in tropical mass flux in the lower stratosphere over the period 1980-2009 despite the ERA-Interim reanalysis not showing a positive trend in upwelling. A linear trend sensitivity analysis for the period over which the trend is calculated has shown that a robust positive linear trend in the tropical upward mass flux takes at least 12 years and in most cases around 20 years to emerge in the

REF-C1SD runs.

Our findings highlight the fact that nudging strongly affects the representation of the stratospheric residual circulation in chemistry-climate model simulations but does not necessarily lead to improvements in the circulation. The differences found in the nudged runs compared with the free-running simulations suggest that although nudging horizontal fields can remove a model bias of, for

example, the temperature field (Hardiman et al., 2017c), the simulated vertical wind field will not necessarily be similar to the reanalysis. It is interesting that the nudging does not constrain the mean strength of the circulation and the trend, but it does constrain the interannual variability. The reasons for this distinction between timescales are currently unknown. The large spread in climatological residual circulation in nudged simulations is an important limitation for those wishing to use nudged simulations

to examine tracer transport, for example ozone trends (Solomon et al., 2016), volcanic aerosols (Schmidt et al., 2018), and diagnostics for age-of-air (Dietmüller et al., 2018). The relaxation timescale when applying the nudging has been found to play an important role in single model studies (Merryfield et al., 2013), but there is no general consensus for the value of the relaxation constant, which is model-specific for the simulations considered here (Morgenstern et al., 2017). The differences in the

stratospheric residual circulation between the REF-C1SD and the REF-C1 runs may not arise solely from the dynamics, but can also be partly influenced by the indirect effects of nudging the temperatures which in turn affects the diabatic heating (Ming et al., 2016a, 2016b). Our results highlight that the method by which the large-scale flow is specified and more specifically the choice of the reanalysis fields, the relaxation timescale and the vertical grid (pressure level versus model level) in which the

nudging is applied needs to be better understood and evaluated for their influence on the stratospheric circulation. Discrepancies between the vertical grid of the models and the reanalysis pressure levels they are interpolated onto or unbalanced dynamics are possible explanations for the differences found between the directly inferred circulation and that diagnosed from the wave forcing in the nudged simulations. Nudging would either violate continuity, or if continuity is maintained, it will come at the

expense of the vertical fluxes, which are not nudged. The interesting aspect here seems to be that this results in substantial change to the net fluxes across a range of timescales, i.e. it does not only increase numerical noise in the $\overline{w}^*$ component. Similar differences in tropospheric transport characteristics in



the CCMI specified dynamics have recently been reported (Orbe et al., 2018). In conclusion we urge caution in drawing quantitative comparisons of nudged CCM simulations against stratospheric
observational data.

*Author contributions*

AC performed the analysis and wrote the article, ACM and MPC made substantial contributions to the conception and design of the study and interpretation of the data. Moreover, they participated in drafting and revising the article. HG provided the correctly calculated EMAC data and SD contributed to the
discussion on the content. The other authors contributed information pertaining to their individual models and helped revise this paper.

*Acknowledgments*

AC was supported by a University of Leeds Anniversary Scholarship. ACM was supported by a NERC Independent Research Fellowship (grant NE/M018199/1). MPC and SD acknowledge support through
the NERC SISLAC grant NE/R001782/1. We acknowledge the modelling groups for making their simulations available for this analysis, the joint WCRP SPARC/IGAC Chemistry–Climate Model Initiative (CCMI) for organizing and coordinating the model data analysis activity and the British Atmospheric Data Centre (BADC) for collecting and archiving the CCMI model output. The EMAC simulations have been performed at the German Climate Computing Centre (DKRZ) through support
from the Bundesministerium für Bildung und Forschung (BMBF). DKRZ and its scientific steering committee are gratefully acknowledged for providing the HPC and data archiving resources for this consortial project ESCiMo (Earth System Chemistry Integrated Modelling). Olaf Morgenstern and Guang Zeng acknowledge the UK Met Office for use of the Met Office Unified Model (MetUM). This research was supported by the New Zealand Government's Strategic Science Investment Fund (SSIF)
through the NIWA programme CACV. Olaf Morgenstern acknowledges funding by the New Zealand Royal Society Marsden Fund (grant 12-NIW-006) and by the Deep South National Science Challenge (http://www.deepsouthchallenge.co.nz). The authors wish to acknowledge the contribution of New Zealand eScience Infrastructure (NeSI) high-performance computing facilities to the results of this research. New Zealand's national facilities are provided by NeSI and funded jointly by NeSI's
collaborator institutions and through the Ministry of Business, Innovation & Employment's Research Infrastructure programme (https://www.nesi.org.nz). CCSRNIES research was supported by the Environment Research and Technology Development Fund (2-1303 and 2-1709) of the Ministry of the Environment, Japan, and a grant-in-aid for scientific research from the Ministry of Education, Culture, Sports, Science and Technology (MEXT) of Japan (16H01183 and 18KK0289), and computations were
performed on NEC-SX9/A(ECO) and NEC-SXACE computers at the CGER, NIES. The GEOSCCM is




supported by the NASA MAP program and the high-performance computing resources were provided by the NASA Center for Climate Simulations (NCCS). The analysis and visualization of the study has been performed using NCAR Command Language (NCL).

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

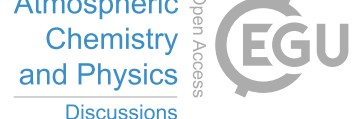


| Model Name | Reference(s) | Resolution | Top Level | REF-C1 ensemble members | Coord. Sys. | NOGWD Reference |
|---|---|---|---|---|---|---|
| CCSRNIES MIROC3.2 | Imai et al., (2013), Akiyoshi et al., (2016) | T42, L34 | 1.2 Pa | 3 | TP | Hines (1997) |
| CESM1 WACCM | Marsh et al., (2013), Solomon et al., (2015), Garcia et al., (2016) | 1.9° × 2.5°, L66 | 140 km | 5 | TP | Beres et al., (2005), Richter et al., (2010) |
| CMAM | Jonsson et al., (2004), Scinocca et al., (2008) | T47, L71 | 0.08 Pa | 3 | TP | Scinocca (2003) |
| EMAC (L47/L90) | Jöckel et al., (2010, 2016) | T42, L47/90 | 1 Pa | 2 | TP | Hines (1997a, b) |
| GEOSCCM | Molod et al., (2012, 2015), Oman et al., (2011, 2013) | ~2° × 2°, L72 | 1.5 Pa | 1 | TP | Garcia and Boville, (1994) |
| MRI-ESM1r1 | Yukimoto et al., (2011, 2012), Deushi and Shibata, (2011) | TL159, L80 | 1 Pa | 1 | TP | Hines (1997b) |
| NIWA-UKCA | Morgenstern et al., (2009, 2013), Stone et al.,(2016) | 3.75° × 2.5°, CP60 | 84 km | 3 | TA | Scaife et al., (2002) |
| SOCOL | Stenke et al., (2013), Revell et al., (2015) | T42, L39 | 1Pa | 4 | TP | Hines (1997a, b) |
| ULAQ CCM | Pitari et al., (2014) | T21, CP126 | 4 Pa | 3 | NTP | no OGWD |

**Table 1.** CCMI simulations that provided TEM diagnostics model output used in this study. CP is
Charney–Phillips; T21 ≈ 5.6° × 5.6°; T42 ≈ 2.8° × 2.8°; T47 ≈ 2.5° × 2.5°; TL159 ≈ 1.125° ×1.125°;
TA is hybrid terrain-following altitude; TP is hybrid terrain-following pressure; NTP is non-terrain-
following pressure.





| Model Name | Pressure/height range of nudging | Newtonian relaxation timescale | Spectral nudging (Y/N) | Nudged Variables | Source of nudging data | Reference |
|---|---|---|---|---|---|---|
| CCSRNIES MIROC3.2 | 1000-1 hPa<br>1-0.01 hPa | 1 day<br>1 day | Y<br>Y | u, v, T, zonal-mean<br>u and T | ERA–I<br>CIRA | Akiyoshi et al., (2016) |
| CESM1 WACCM | Surface – 50km (transition 40 – 50km) | 50 h | N | u, v, T, surface pressure, surface stress, latent/sensible heat flux | MERRA | Lamarque et al., (2012) |
| CMAM | Surface – 1hPa | 24 h | Y | Divergence, vorticity, temperature | ERA–I | McLandress et al., (2013) |
| EMAC (L47/L90) | 920 – 780 hPa (transition)<br>710 – 10 hPa (full)<br>10 – 6 hPa (transition) | 48 h<br>6 h<br>24 h<br>24 h | Y | Divergence, vorticity, T (with wave-0), (logarithm of) surface pressure | ERA–I | Jöckel et al., (2016) |
| MRI-ESM1r1 | 870 -1 hPa | 24 h (870 -40 hPa)<br>24-∞ h (40-1 hPa) | N | u, v, T | JRA55 | Deushi and Shibata, (2011) |


**Table 2**: Details of nudging in the CCMI REF-C1SD simulations that provided TEM diagnostics model output used in this study. ERA–I = ERA-Interim; CIRA = Cooperative Institute for Research in the Atmosphere; MERRA = Modern Era Retrospective ReAnalysis; JRA-55 = Japanese 55- year ReAnalysis.


| Model Name | REF-C1 | REF-C1SD |
|---|---|---|
| CCSRNIES MIROC3.2 | ✓❖✚★▲■ | ✓❖✚★▲■ |
| CESM1 WACCM | ✓❖✚▲■ | ✓❖✚▲■ |
| CMAM | ✓❖✚▲■ | ✓❖✚▲■ |
| EMAC (L47/L90) | ✓❖✚★▲■ | ✓❖✚★▲■ |
| GEOSCCM | ✓❖✚★▲■ | |
| MRI-ESM1r1 | ✓❖✚★▲■ | ✓❖✚★▲■ |
| NIWA-UKCA | ✓❖✚★ | |
| SOCOL | ✓❖ | |
| ULAQ CCM | ✓❖ | |

**Table 3.** Available TEM-related model output for each model from the CCMI-1 archive: $\overline{w}^*$ (✓), $\overline{v}^*$

(❖), EPFD (✚), GWD (OGWD+NOGWD) (★), OGWD (▲), NOGWD (■).



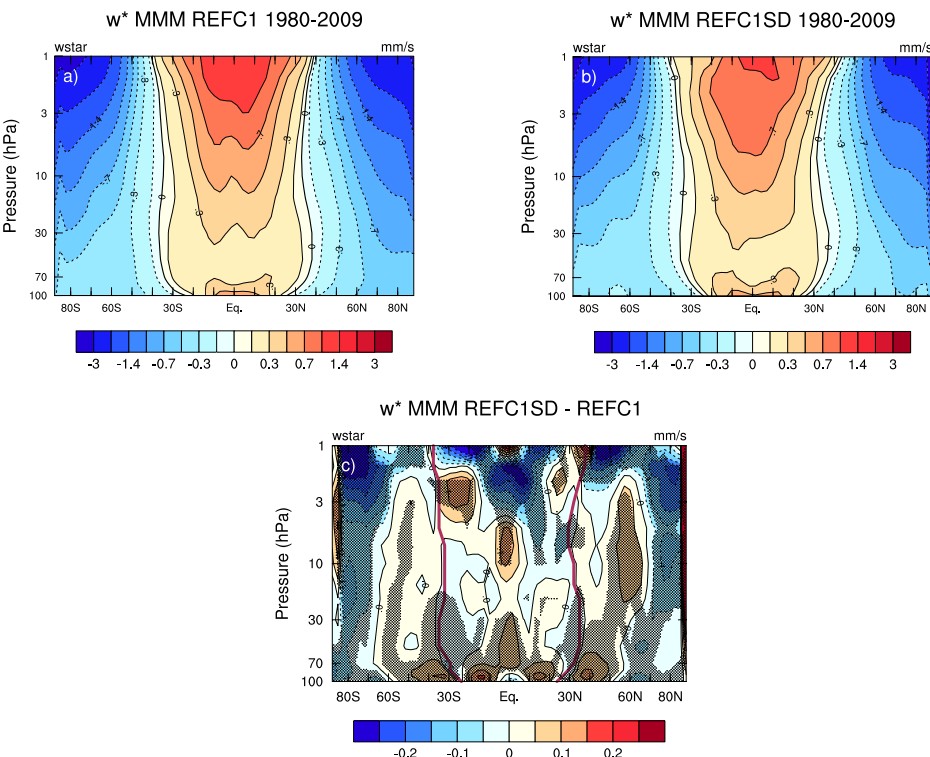

**Figure 1.** Latitude vs. pressure climatology (1980-2009) of MMM annual mean $\overline{w}^*$ for (a) REF-C1 simulations, (b) REF-C1SD simulations and (c) the REF-C1SD – REF-C1 differences. Stippling denotes statistical significance at the 95% confidence level and the red lines denote the climatological turnaround latitudes in REF-C1SD.





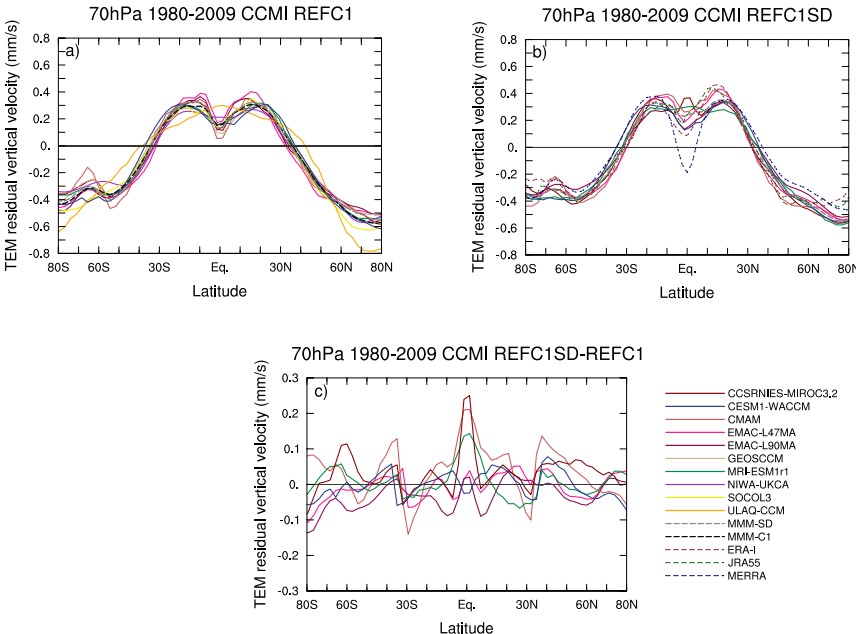


**Figure 2.** Mean strength of annual mean $\overline{w}^*$ [mm s$^{-1}$] at 70 hPa for (a) REF-C1 free-running models, (b) REF-C1SD nudged models, and (c) the REF-C1SD – REF-C1 difference.






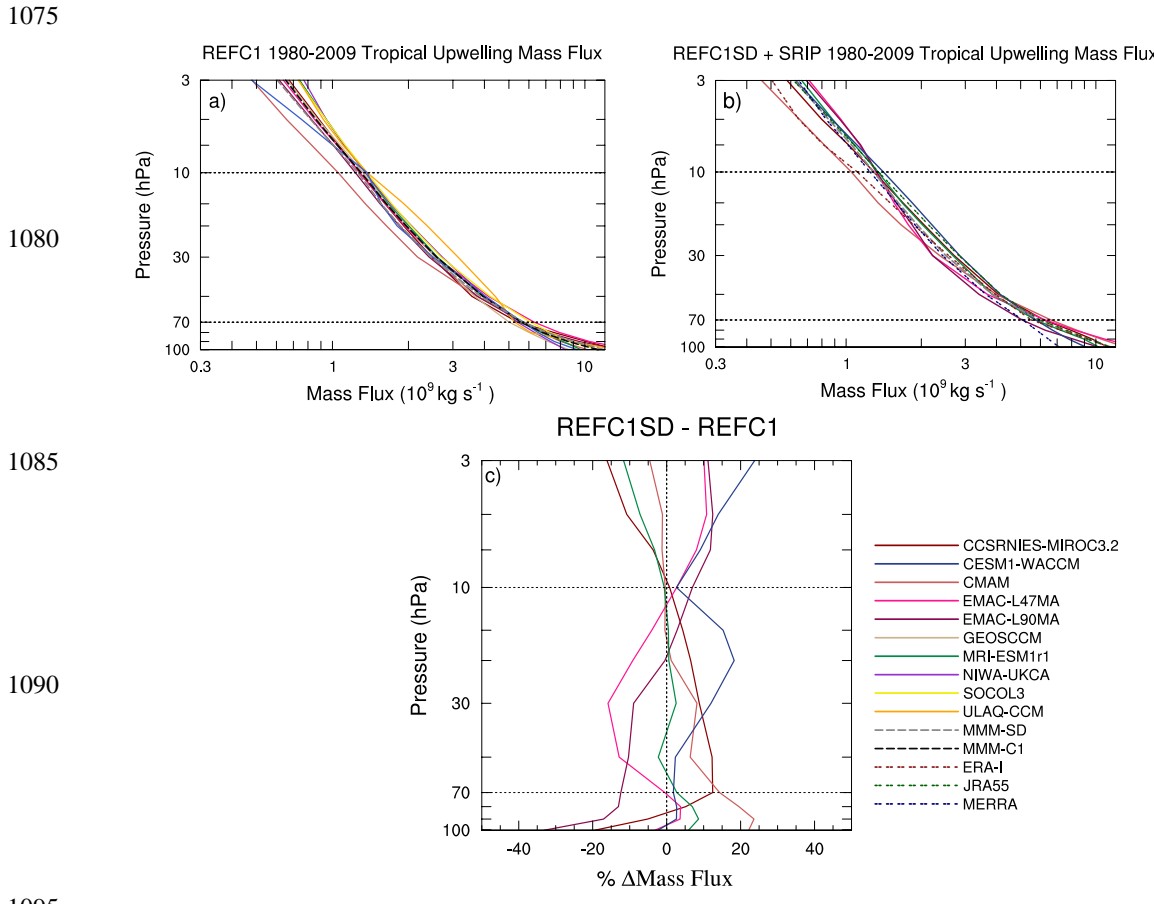





**Figure 3.** Vertical profiles of climatological (1980-2009) tropical upward mass flux [$10^9$ kg s$^{-1}$] averaged between the turnaround latitudes for (a) REF-C1 and (b) REF-C1SD, (c) % differences between REF-C1SD and REF-C1. Note the logarithmic x-axes in panels (a) and (b).




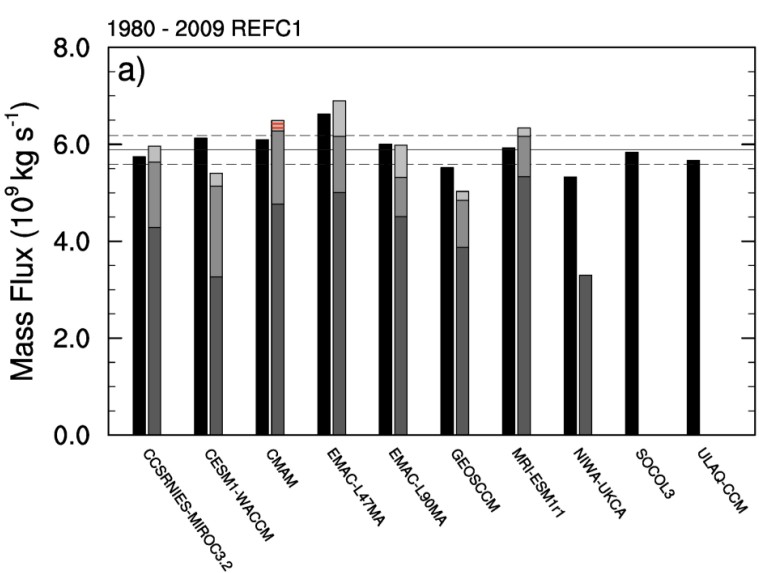

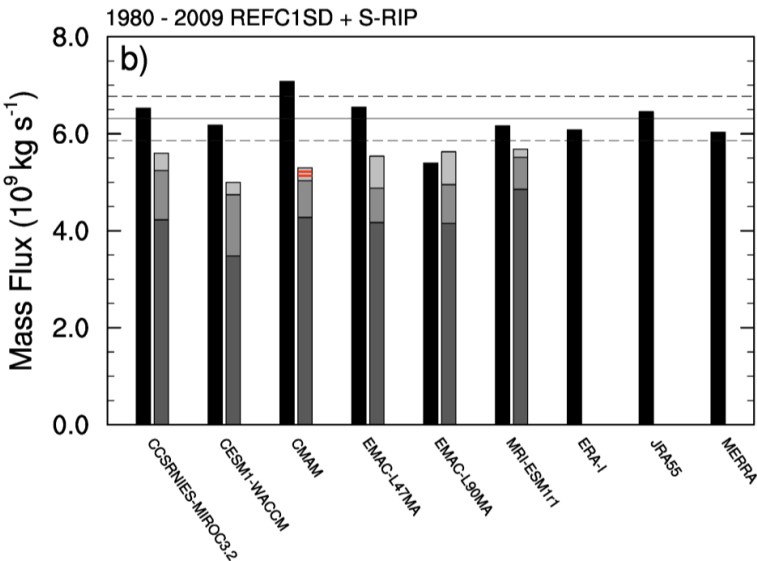

**Figure 4.** Tropical upward mass flux at 70 hPa (left bars) along with downward control calculations (right bars) showing contributions from EPFD (dark grey), OGW (mid-grey), and NOGW (light grey) for (a) REF-C1 and (b) REF-C1SD. For CMAM the NOGWD contributes negatively to TUMF and is indicated with two red horizontal lines inside the lighter grey bar.



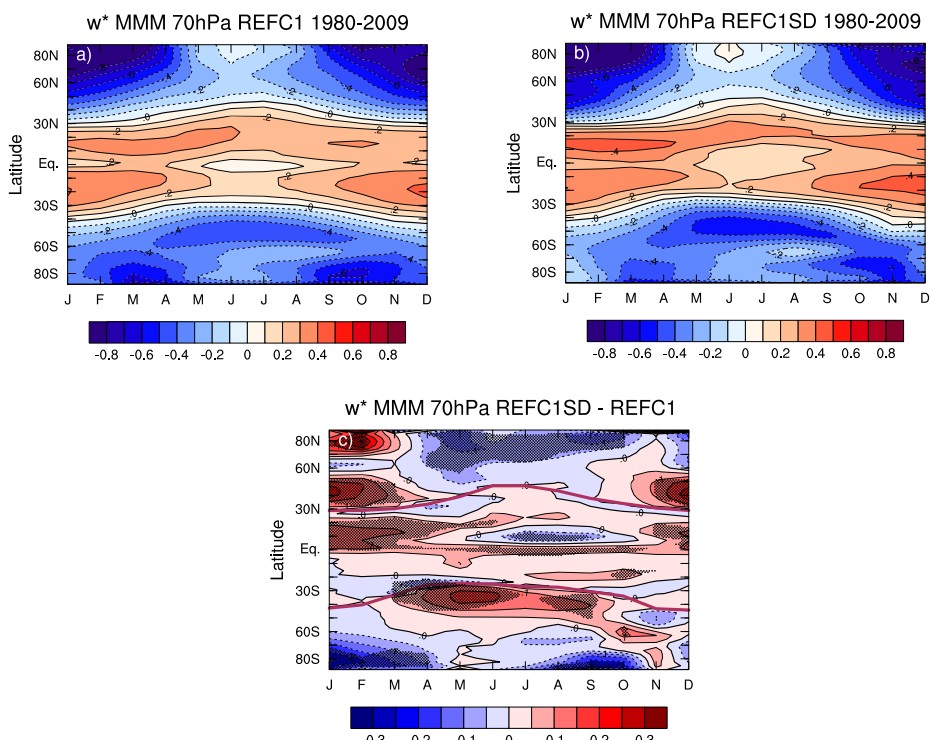

**Figure 5.** Climatological MMM annual cycle in $\overline{w}^*$ [mm s$^{-1}$] at 70 hPa for (a) REF-C1 simulations, (b) REF-C1SD simulations and (c) the REF-C1SD – REF-C1 difference overlaid with a student's t-test where stippling denotes statistical significance above 95%. The turnaround latitudes ($\overline{w}^* = 0$) are shown by the thick black lines in panels 5a and 5b and by the thick purple line for the MMM of REF-C1SD simulations in panel 5c.





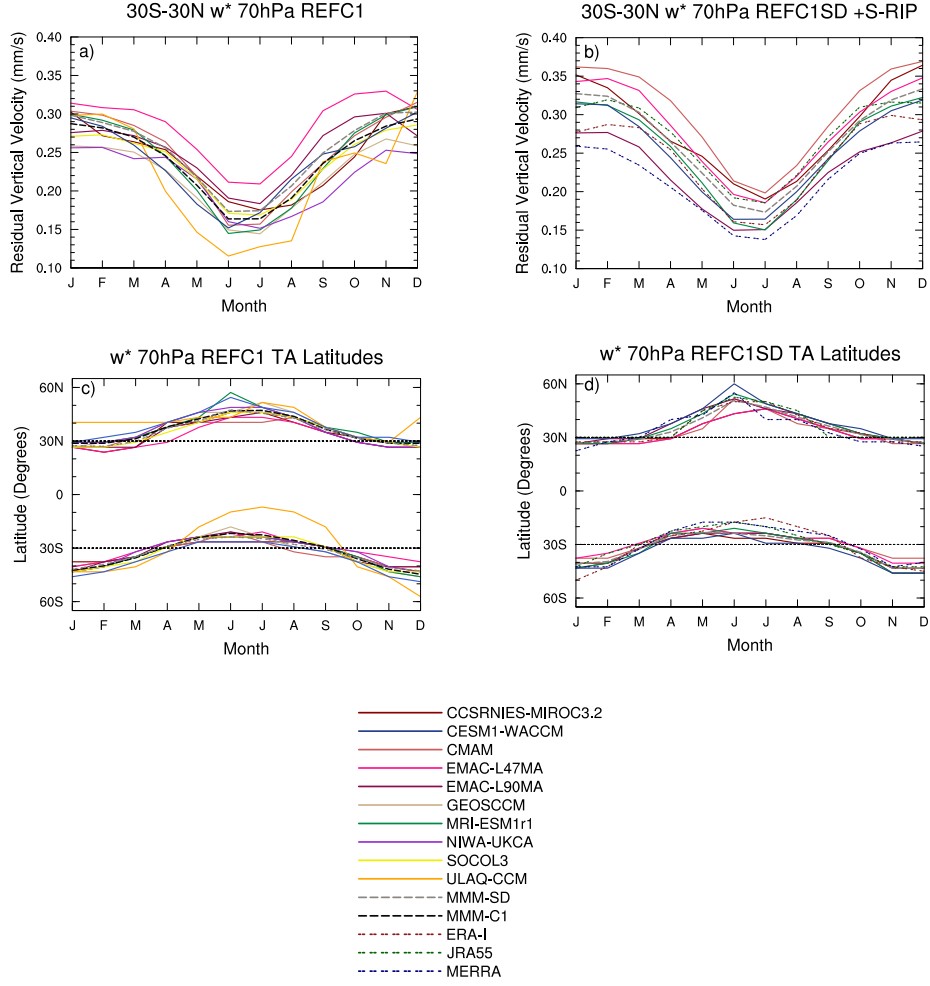

**Figure 6.** (Top) Climatological annual cycle in $\overline{w}^*$ [mm s⁻¹] at 70 hPa between 30°S-30°N in (a) REF-C1
and (b) REF-C1SD. (Bottom) Climatological annual cycle in turnaround latitudes at 70 hPa for each
model in (c) REF-C1 and (d) REF-C1SD.





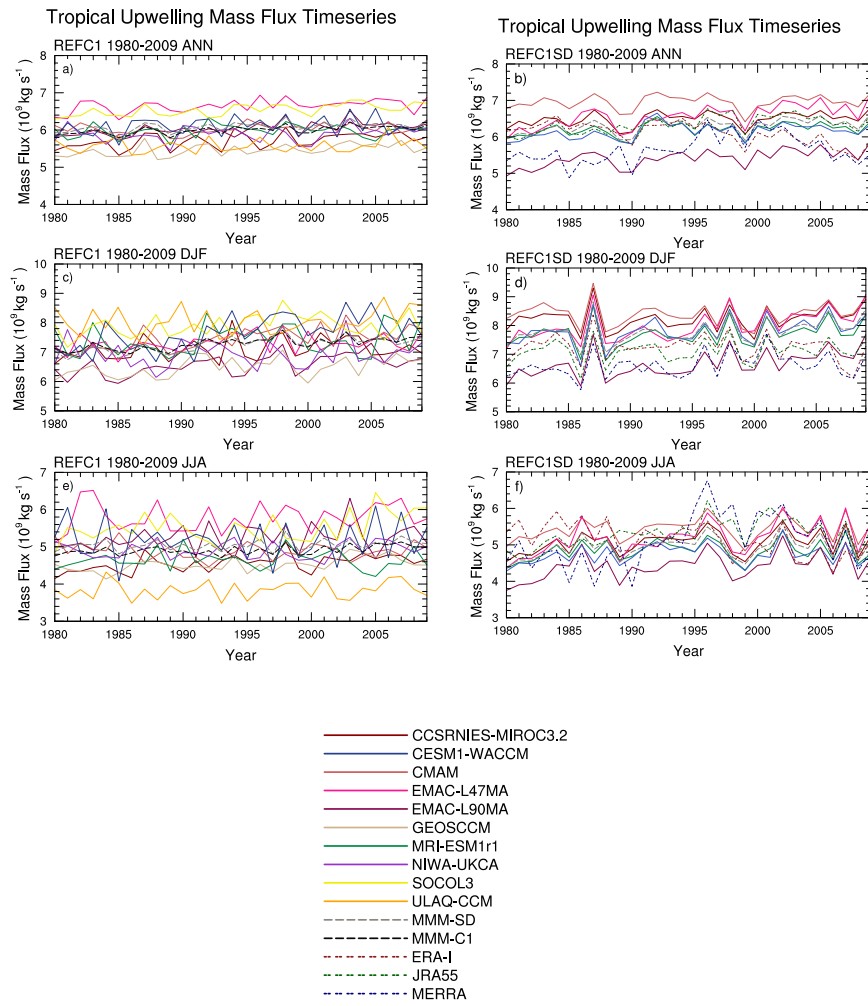

**Figure 7.** Annual (ANN), DJF and JJA means of tropical upward mass flux [$10^9$ kg s$^{-1}$] at 70 hPa for (left panels) REF-C1 simulations and (right panels) REF-C1SD simulations.



**Figure 8.** Timeseries of the annual tropical upward mass flux anomalies calculated from w* (a, b), and the DCP inferred contributions from resolved (EPF Divergence) wave driving (c, d), orographic gravity (OGW) wave drag (e, f), non-orographic gravity (NOGW) wave drag (g, h) and from the total parameterized (OGW+NOGW) gravity wave drag (i, j) for (left panels) REF-C1 simulations and (right panels) REF-C1SD simulations.





**Figure 9.** Timeseries for REF-C1 simulations of the components of the annual mean tropical upward mass flux attributed to (a) volcanic aerosol, (b) ENSO, (c) linear trend, (d, e) the QBO, and (f) the residuals from the mass flux timeseries and that reconstructed from the MLR.



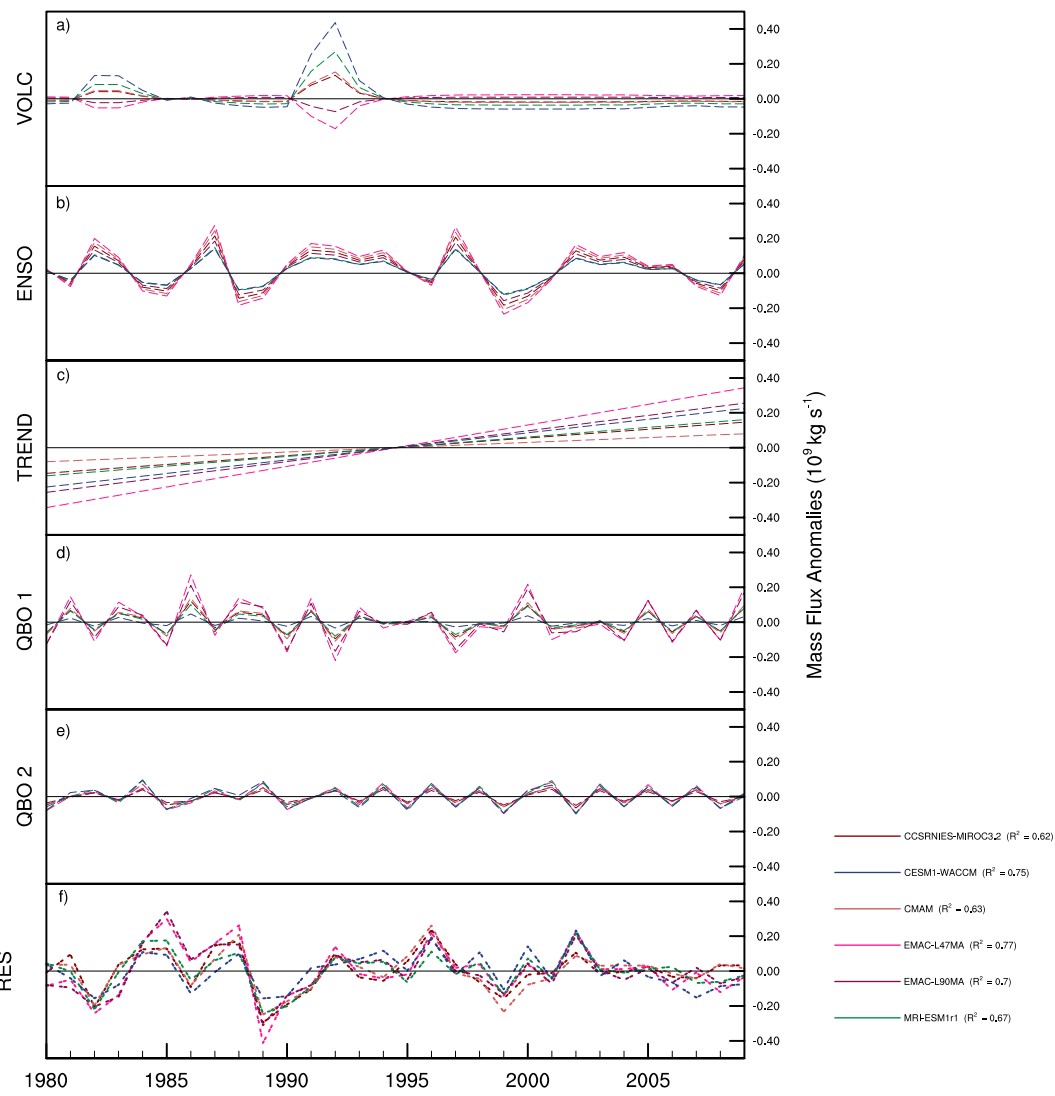

**Figure 10.** Timeseries for REF-C1SD simulations of the components of the annual mean tropical
upward mass flux attributed to (a) volcanic aerosol, (b) ENSO, (c) linear trend, (d, e) the QBO, and (f)
the residuals from the mass flux timeseries and that reconstructed from the MLR.




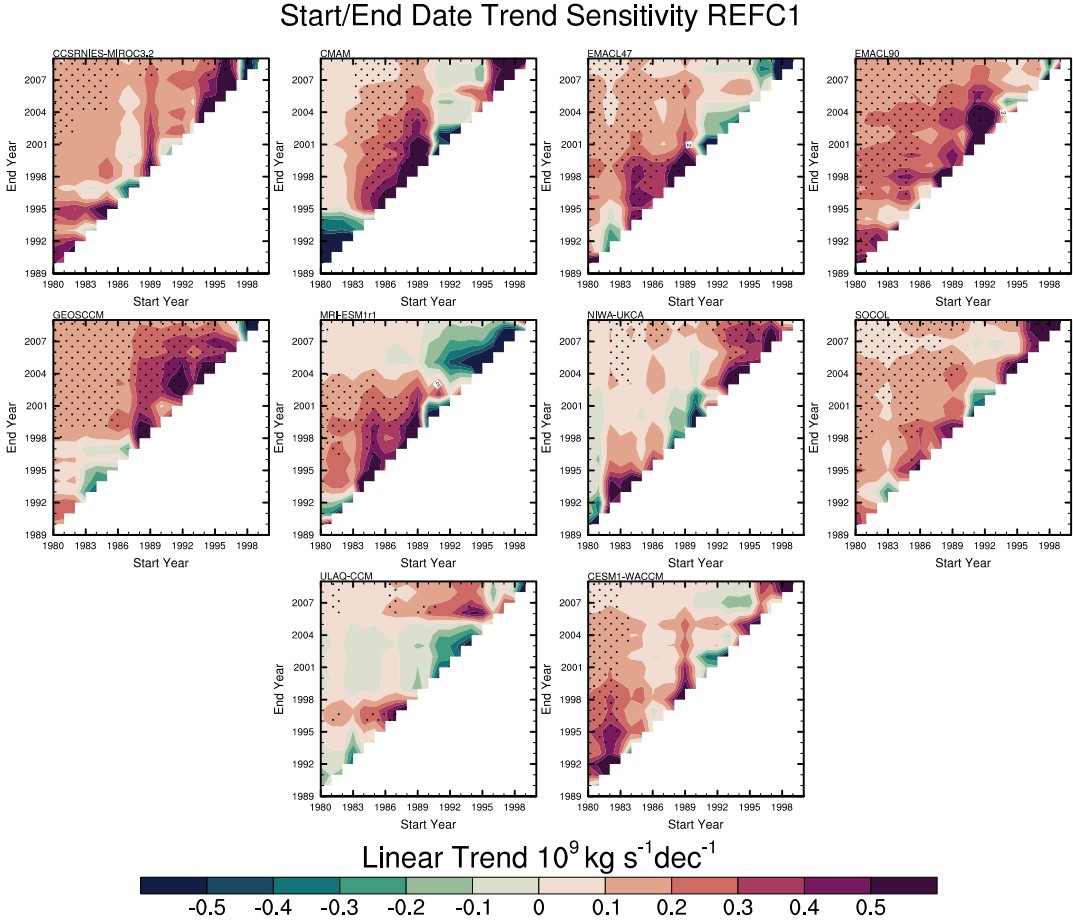

**Figure 11.** Mass flux linear trend partial regressor from the MLR sensitivity plots (values with

statistical significance are stippled) over period 1980-2009 for REF-C1 (r1i1p1) simulations.



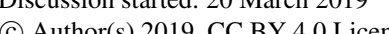

**Figure 12.** Mass flux linear trend partial regressor from the MLR sensitivity plots (values with statistical significance are stippled) over period 1980-2009 for REF-C1SD simulations.
