# Peer review of "The effect of atmospheric nudging on the stratospheric residual circulation in chemistry-climate models"

_Atmospheric Chemistry and Physics, 2019_

## Referee Comment (RC1) · Anonymous Referee #1 · 3 May 2019

Reviewer (Comments):
**Review of "The effect of atmospheric nudging on the stratospheric residual circulation in chemistry-climate models" by Andreas Chrysanthou et al.**

**Recommendation: Publication after minor revision**

The paper is very well organised and written. The topic discussed here – the effect of nudging CCMs on the stratospheric residual circulation – is of very high relevance, because nudged or specified dynamics CCM simulations are a common tool for analysing and interpreting observed changes of the stratospheric trace gas composition (e.g. Froidevaux et al. 2019). This work will most likely trigger many more studies related to the problems induced by driving CCMs (or more generally GCMs) with specified dynamics derived from meteorological reanalysis. However, some further steps to disentangle the main source(s) of the problem could already be done in this paper or the author could at least explain, why this could not be done here (see comments below).

The paper should be submitted after addressing the comments below.

**General comments:**

The topic is important and it is time to carefully analyse the effect of nudging CCMs towards meteorological reanalysis data (RA) on the simulated stratospheric residual circulation and consequently also on the transport of chemical species. The latter is not the focus of this paper here, but the results in this paper will hopefully stimulate further studies on the impact of nudging on stratospheric tracer transport and composition, e.g. on CCM-SD simulations of the recent lower stratospheric ozone trend reported by Ball et al. (2018).
I highly appreciate the topic and the great effort made by the authors and I agree in general to the conclusions derived from the results presented in this very valuable multi-model analysis. However, the manuscript should and could be focused more on the main topic and the new aspect that is clearly announced in the title: "**The effect of nudging** on the stratospheric …". The differences between specified dynamic (SD) and free-running (FR) simulations is to my point of view important, but only one (minor) aspect of this paper and should be contrasted with the differences between the SD simulations and the reanalysis datasets (RA) for the individual measures of the stratospheric residual circulation. This would describe the quantitative effect of nudging CCMs.
As outlined above, the paper is to my opinion focused too much on the difference between FR and SD simulations – this becomes obvious in most Figures where only the differences of REF-C1SD-REF-C1 (SD-FR) are shown and the differences REF-C1SD-Reanalysis (SD-RA) is missing. The systematic analyse of SD-RA and SD-FR for all the metrics of stratospheric residual circulation would help to get a more quantitative view and help to better understand the effect of nudging on the stratospheric residual circulation. It is clear that the underlying mechanisms that trigger the observed discrepancies in the stratospheric residual circulation between nudged CCMs and reanalysis is not in the scope of this paper, but the differences SD-RA should be shown in the same way as the differences SD-FR, so that the effect of nudging – reflected by the differences to the reanalysis – could be contrasted and discussed more quantitatively compared to the differences between specified dynamics and free-running simulations.
Frankly spoken, it would be better to streamline this paper and to concentrate only on the CCMs that have carried out SD and FR simulations in a consistent way. To my opinion, the

other non-nudged models could not add any information to the main topic of the paper – the effect of nudging CCMs. The consequence would be to skip 4 out of 9 models or 4 out of 10 simulations which do not provide SD simulations. Including 4 non-nudged CCMs to the multi-model-mean (MMM) of the FR simulations is not really helpful to analyse the effect of CCM nudging. In the best case, the results are not blurred by these models, but to my opinion the analysis of SD-FR vs. SD-RA for the 6 simulations providing all relevant information would help to get a much more stringent and systematic analysis of the role and impact of nudging CCMs on their stratospheric residual circulation. This paper would gain, if the also very interesting inter-model stratospheric residual circulation comparison of all participating CCMI models would be done in a separate paper.

**Specific comments:**

L.79: "*… from which age-of-air (AoA) can be estimated (Waugh and Hall, 2002).*"
For clarity, I would suggest to add: …, however AoA represents the combined effects of residual circulation and mixing.
Waugh and Hall (2002) is a review paper and not the original reference for AoA from tracer measurements and the credit should be given to the original work. To my knowledge, the first stratospheric AoA estimates from tracer measurements has been reported by Schmidt and Khedim (1991) and the concept of AoA was first applied to the stratosphere by Kida (1983).

L.79-80: Engel et al. (2009) found no decrease in AoA, but the observed increase of AoA was statistically non-significant.

L.191: The reference for the citation "(Rosenlof, 1995)" is missing.

L. 276-284: To my opinion, the Supplemental Figure 2 should be moved to the manuscript, because the topic is discussed here and the TA-latitudes are a simple measure and very indicative for the different structure of the residual circulations derived from SD and FR simulations. This Figure could be extended with the weighted mean TA-latitudes of the different reanalysis datasets (depending how often they are used for SD runs: 4x ERA-I, 1x MERRA and 1x JRA55). This would give a measure of the differences between the SD simulations and the reanalysis induced by nudging.

L327-338: Here, the author discuss some the differences between SD and reanalysis for wbar_star on the 70 hPa level as shown in Fig. 2b). As noted in the general comments, it would be helpful to add here the corresponding differences SD-RA.

L350-364: Here, the author discuss the discrepancies between SD and RA using Figure 3b. Again, I would suggest to add a fourth panel (Figure 3d) that clearly shows the differences SD-RA (same as panel 3c) for the vertical profile of the climatological TUMF.

L.364-366: It is not clear to me, what the total spread here really is. Is it the standard deviation of all models TUMF in the range 100-30 hPa? Could you please clarify?

L. 377-379: The second model, for which TUMF from wbar_star exceeds TUMF from total wave forcing for the REF-C1 simulations, is the GEOSCCM and not the EMAC-L90MA.

L392-397: It sounds not reasonable to me that the SD-RA discrepancies for TUMF should be related to the individual RA dataset, it is much more reasonable that the individual model itself and how the nudging is implemented is causing these differences.

L.401-404: Is the large positive difference between TUMF derived from wbar_star and derived from total wave forcing not what one would expect, if one assumes that nudging CCMs might lead to additional forcings to the residual stratospheric circulation induced by the inconsistencies with the modelled physics and not by wave breaking?
In my naive way of thinking, I would expect that TUMF from total wave forcing derived from the downward control principle (DCP) is at least equal or maybe slightly larger than the directly calculated TUMF from the residual vertical velocity due to the possible wave-wave interactions and the slight imperfections of the "exact downward control" in transient (non-steady state) cases. The internally more consistent free-running simulations without the additional tendencies induced by nudging seem to corroborate this hypothesis.
For SD simulations only the EMAC-L90 model behaves different (more realistic?) with a slightly larger TUMF derived from wave forcings. Could this be the consequence of a different nudging procedure? The EMAC-L90 and to a less degree the MRI-ESM1r1 are also the only models for which TUMF at 70 hPa derived from the SD simulation is smaller than that derived from the applied RA datasets, ERA-I and JRA55 respectively. Both models also show the smallest difference between directly calculated TUMF and TUMF derived from total wave forcing.
To my opinion, it would be worth to extend the discussion on these topics (RA vs. SD) a bit more to focus the paper more on the main topic stated in the paper's title.

L.404-406: The author states that the lower directly calculated TUMF values for the higher vertically resolved EMAC-L90 compared to -L47 is in line with the finding for the SOCOL3 model reported by Revell et al. (2015b), i.e. 90 vs. 39 layers. This is true for the FR simulations which has been used in the sensitivity experiments by Revell et al. (2015b), but does this conclusion also holds for both EMAC SD simulations? Does the vertical resolution also matters here?

L.427-436: The differences in the annual cycle of wbar_star at 70 hPa between SD and RA should be added to the Figure 5 and discussed here.

L.431-432: The author states that NH midlatitude downwelling during summer is stronger for the SD simulations. To my view, there are only blueish colours north of the TA-latitudes during JJA in Figure 5c indicating weaker and not stronger downwelling.

L.451-452: *"The REF-C1SD shows…"*
I assume that this sentence concerns the comparison between SD and RA, however it might be better to clarify this.

L.454-456: Why is the annual cycle of wbar_star in the tropical lower stratosphere more consistent for the SD compared to the FR simulations? Above you conclude that the annual cycle and the phasing of SD simulations are weakly constrained and the intermodal spread is 20% larger than for FR simulations. How does this fit together?

L.456-459: Again, I am missing in the summary here the SD vs. RA comparison relevant for analysing the nudging effect on stratospheric residual circulation in CCMs.

L.503-569 (Section 3.5 and 3.6): Would it be possible to add here also the MLR of TUMF at 70 hPa and the related trend analysis for the RA datasets? The latter might give a very interesting insight into the question, if the different linear trends of FR and SD simulations are mainly driven by the different stratospheric residual circulation of the RA datasets or if they are significantly influenced by the nudging of the CCMs.

L585-588: What is the explanation of the differences among the directly simulated TUMF for SD simulations that can be derived from the diagnosed wave forcing using the downward control principle? (See also my comments above: L.401-404)

L.601-602: What is the explanation that ERA-I shows no positive trend in tropical upwelling but a significant positive linear trend in TUMF at 70 hPa?

**References:**

Froidevaux, L., Kinnison, D. E., Wang, R., Anderson, J., and Fuller, R. A.: Evaluation of CESM1 (WACCM) free-running and specified dynamics atmospheric composition simulations using global multispecies satellite data records, Atmos. Chem. Phys., 19, 4783-4821, 10.5194/acp-19-4783-2019, 2019.

Kida, H.: General Circulation of Air Parcels and Transport Characteristics Derived from a Hemispheric GCM Part 2. Very Long-Term Motions of Air Parcels in the Troposphere and Stratosphere, Journal of the Meteorological Society of Japan. Ser. II, 61, 510-523, 10.2151/jmsj1965.61.4_510, 1983.

Rosenlof, K. H.: Seasonal Cycle of the Residual Mean Meridional Circulation in the Stratosphere, J. Geophys. Res., 100, 5173-5191, 1995.

Schmidt, U., and Khedim, A.: In situ measurements of carbon dioxide in the winter Arctic vortex and at midlatitudes: An indicator of the 'age' of stratospheric air, Geophys. Res. Lett., 18, 763-766, 10.1029/91gl00022, 1991.

---

## Referee Comment (RC2) · Anonymous Referee #2 · 14 May 2019

In this paper, the authors compare the free running and specified dynamics simulations of the CCMI models. They find that although the nudging involved in the SD does make the models short term variability line up, the residual circulation in the nudged runs matches neither the original models nor the reanalyses to which the models are nudged.

This paper is important. It is interesting and provides a cautionary tale against using these SD runs to examine transport of long-lived trace gases, which one would expect would be significantly impacted by these inconsistencies. It is well reasoned and well written. I recommend publication following some minor revisions. An expanded

introduction would be extremely valuable to the reader, and I have included recommendations for that below. The primary scientific points that would add significantly to the paper are:

1. Discuss the choice of metric for the BDC in more detail. (Line 191)

2. Reconsider the results of the multiple linear regression in light of the importance of the relative phasing of the annual cycle and the QBO.

3. Consider the differences between the models and the reanalyses they are being nudged towards, especially vertical profiles, to try to better understand why the results are what they are.

Details below: 1) Recent work (Abalos et al. 2015, Linz et al. 2019) has raised the issue that the differences due to different calculations of the residual circulation are as significant as (or more than) the differences between the reanalyses themselves (c.f. Seviour et al. 2012, also). Therefore, I would appreciate more discussion of the choice made here to use the TUMF and downward control (which is used to distinguish the different wave forcing components). How do these choices impact the results, if at all? A thermodynamic calculation might be very different, since the temperature is nudged, which will introduce spurious diabatic terms. How is TUMF different from w* averaged between turnaround latitudes?

2) Although the method used here to do the multiple linear regression is somewhat standard, it does not actually remove all of the variability that is related to these signals. In particular, the relative phasing of the QBO and the annual cycle will cause an enhanced wintertime upwelling in some years, so the result that the leftover variability is highly correlated could be related to this. Randel et al. 1999 have a nice treatment. I would consider using the method in that paper (described on p. 458, see also Randel and Wu 1996) to determine if the relative timing of the annual cycle and QBO is the cause of the covariation of the nudged time series. This phenomenon is also mentioned in the cited Baldwin et al. 2001 QBO review. 3) The authors find that the reanalyses

differ from the nudged runs which in turn differ from the free running models. I would like to see some presentation of the difference between the mean states of the reanalyses and the mean states of the models without nudging. The result that the models tend to have stronger upwelling in their SD runs could be caused by a systematic low bias in the model temperature in the tropics compared to the reanalysis temperature. The repeated nudging with a primarily positive value would then cause an apparent increase in upwelling, without changing the underlying model physics that are the cause of any systematic mean state bias. To understand why the nudging does what it does, I think the difference between the free running models and the reanalyses needs to be explored in much more detail–perhaps for just one model. A continuous spurious forcing because of a mismatch in the mean state (and likely also in the seasonal cycle) should impact the mean while leaving the interannual variability alone (line 612).

More minor comments below: I find the second paragraph of the introduction to be lacking. Although Abalos et al. (2015) did conclude that there was likely a 2-5%/decade trend in the circulation, they found substantial differences with different calculations for the circulation strength. Ploeger et al. (2019) have a new paper out on the inconsistency of the BDC trends for the different reanalyses, looking at the age spectra. Age of Air is mentioned somewhat abruptly and with no explanation. The results are also misrepresented. Engel et al. 2009 did not find a statistically significant increase in the mean age of air in the midlatitudes. They found no statistically significant change. There are limitations both on the availability of measurements and on the interpretation of age of air as a direct proxy for the residual circulation, and much more recent papers discuss this, rather than Waugh and Hall 2002. There are new satellite based measurements–ACE-FTS (e.g. Ray et al. 2016), MIPAS (Haenel et al. 2015), and MLS (Linz et al. 2017)–and new measurements from Air Core (Engel et al. 2017). The interpretation of the data is also difficult because of the separation of the residual circulation from the mixing (e.g. Garny et al. 2014, Ploeger et al 2015, as mentioned in the first paragraph–perhaps just move this comment to this paragraph?). The only attempt to calculate the residual circulation from tracer data did not calculate the resid-

ual circulation directly, but instead the diabatic circulation (Linz et al. 2017), and they did not calculate trends.

A more detailed explanation of what nudging is, how it is implemented, why it is used, and the motivation for this work would help the introduction. The current paragraph in the introduction (line 85) is good, and I think that including more specific examples would be very helpful. Similarly, in the conclusion, commentary on what these nudged runs could be useful for would be appreciated in addition to the entirely appropriate cautionary words.

Do any of the "free running" models have a nudged QBO, and if so, which ones?

Since the focus of this paper is really on the difference between SD and free running models, it might make sense to omit the models that do not include both. I respect the authors choice to have those comparisons in the supplementary information. However, I think I would find the plots more manageable without the additional lines.

Line 106: aren't they necessarily inconsistent? Lines 100-129: excellent discussion. Line 153: Do you mean Table 2? Line 155 etc.: More details here would be good. I don't understand what is meant by T (with wave-0), and I assume that "temperature" is also T (for CMAM), but if that's true then they all nudge temperature so maybe mention that? Including the sentence "Nudging timescales range from 6 - 50 hours." (or some equivalent) would also emphasize how different these treatments are. Line 159-160: Table 1? Line 168: "primary variable"–not sure what is meant by this Line 442: I have trouble seeing this from the plot. Could you provide the numbers? Is the difference in the peak-to-peak amplitude significant? Line 476...: How do these results relate to Abalos et al. 2014, who found such a strong dependence of the upwelling on the location of the EPFD? Line 488-90: I found this sentence confusing. Line 498: remove "due to" Section 3.6: Nice. Line 606...: This paragraph is very long and contains a lot of important information. Turn into two or even three shorter paragraphs?

Figures: Consider stippling the insignificant part of the plot. I read this on a number

of different devices, and on some it looked fine, but on one of them the plots were impossible to see because the colors weren't visible behind the stippling. This wasn't an issue for the last two figures for some reason.

It could be useful to have models that are nudged to the same reanalysis plotted in the same color family to aid interpretation of the complicated line plots.

References: Linz et al. 2019: https://www.atmos-chem-phys.net/19/5069/2019/acp-19-5069-2019.html Randel et al. 1999 https://acomstaff.acom.ucar.edu/randel/JAS%201999.pdf Randel and Wu 1996 https://journals.ametsoc.org/doi/10.1175/1520-0469%281996%29053%3C2546%3AIOTOQI%3E2.0.CO%3B2 Ploeger et al. 2019 https://www.atmos-chem-phys.net/19/6085/2019/ Ray et al. 2016 https://agupubs.onlinelibrary.wiley.com/doi/full/10.1002/2015JD024447 Haenel et al. 2015 https://www.atmos-chem-phys.net/15/13161/2015/acp-15-13161-2015.html Linz et al. 2017 https://www.nature.com/articles/ngeo3013 Abalos et al. 2014 https://journals.ametsoc.org/doi/10.1175/JAS-D-13-0366.1

---

## Author Comment (AC2) · 27 Jun 2019

**Author response to referee comments on Chrysanthou et al. "The effect of atmospheric nudging on the stratospheric residual circulation in chemistry-climate models" submitted to ACPD**

**Reply to Anonymous Referee #2**

In this paper, the authors compare the free running and specified dynamics simulations of the CCMI models. They find that although the nudging involved in the SD does make the models short term variability line up, the residual circulation in the nudged runs matches neither the original models nor the reanalyses to which the models are nudged.

This paper is important. It is interesting and provides a cautionary tale against using these SD runs to examine transport of long-lived trace gases, which one would expect would be significantly impacted by these inconsistencies. It is well reasoned and well written. I recommend publication following some minor revisions. An expanded introduction would be extremely valuable to the reader, and I have included recommendations for that below. The primary scientific points that would add significantly to the paper are:

1. Discuss the choice of metric for the BDC in more detail. (Line 191)

2. Reconsider the results of the multiple linear regression in light of the importance of the relative phasing of the annual cycle and the QBO.

3. Consider the differences between the models and the reanalyses they are being nudged towards, especially vertical profiles, to try to better understand why the results are what they are.

Thank you for your positive comments and suggestions for improving the paper. We reply to the specific points raised below in red.

**Specific comments:**

1) Recent work (Abalos et al. 2015, Linz et al. 2019) has raised the issue that the differences due to different calculations of the residual circulation are as significant as (or more than) the differences between the reanalyses themselves (c.f. Seviour et al. 2012, also). Therefore, I would appreciate more discussion of the choice made here to use the TUMF and downward control (which is used to distinguish the different wave forcing components). How do these choices impact the results, if at all? A thermodynamic calculation might be very different, since the temperature is nudged, which will introduce spurious diabatic terms. How is TUMF different from w* averaged between turnaround latitudes?

Thank you for this suggestion. We agree that some further discussion of the chosen measures of residual circulation is important to place the results into a broader context. As indicated by the reviewer, the choice of diagnostics is likely to be a particular issue for interpreting the nudged

models and reanalyses due to the additional tendencies imposed in the model equations. Part of our choice is pragmatic, based on the availability of output from the CCMI model dataset, which provides direct estimates of residual circulation diagnostics following Andrews et al. (1987) and resolved and parameterized wave forcing components. As TUMF is an integrated measure of w* between TA latitudes, the results regarding interannual variability and long-term trends are similar to if an average w* measure is used. Arguably, the TUMF is a first-order measure in order to evaluate whether the mass circulation is affected by nudging and it has been widely used in previous multi-model comparison and reanalysis studies and so we include it here (e.g. Butchart et al., 2006, Butchart et al., 2010, Butchart et al., 2014 and references therein; Rosenlof, 1995; Seviour et al., 2012; SPARC CCMVal, 2010). The important point is that we use consistent diagnostics in the free running and nudged experiments and the reanalyses, so that we are always comparing the same measures across the different simulations. Unfortunately, as heating rates are not provided by all models in the CCMI output, it would not be possible to compute a consistent (i.e. with the appropriate model radiation scheme) estimate of the residual circulation using the thermodynamic equation. It is possible that for the nudged simulations calculation of the residual circulation based on the thermodynamic equation might result in different behaviour to the direct TEM formulation used here. We now suggest this as an interesting topic for future research in the conclusions. The downward control principle (DCP; Haynes et al., 1991) calculations provide a framework to assess the extent to which the wave forcing and directly estimated residual circulation become decoupled as a consequence of the additional tendencies imposed by the nudging.

Based on the above we have expanded/edited the relevant parts of the text to the following :

End of section 2.2.1, (just before the introduction of DCP) :

*"The TUMF has been used widely as a measure of the strength of the BDC (Rosenlof, 1995; Butchart et al., 2006, Butchart et al., 2010, Butchart et al., 2014 and references therein; Seviour et al., 2012), so it's use here enables a direct comparison with earlier studies. Arguably, the strength of the TUMF is a first-order metric to evaluate changes to the mass circulation as a consequence of nudging. As mentioned above, by calculating the annual means of TUMF accounting for the seasonal cycle of the turnaround latitudes we capture the correct evolution of the intraseasonal (not shown) and interannual variability in the TUMF."*

End of section 2.2.2, (after introduction of DCP) :

*"Both the direct and DCP methods rely on the applicability of the quasi-geostrophic approximation to interpret the results. We note that in addition to the direct and DCP approaches used here, the residual circulation can also be estimated using the thermodynamic*

*equation. Unfortunately, heating rates were not available from all CCMI model simulations to perform this calculation. Studies have shown that the estimates from the different methods for evaluating the residual circulation can differ (Abalos et al., 2015; Linz et al., 2019; Seviour et al., 2012), particularly in reanalyses where the normal conservation laws are not required to be met. Similar issues are likely to beset the nudged model simulations owing to the additional tendencies included in the model equations. The differences between the calculation methods for the residual circulation can be as large as, or larger than, the differences between reanalysis datasets for the same diagnostic (Abalos et al., 2015; Linz et al., 2019), and may further depend on choices around averaging between fixed latitudes or the turnaround latitudes (Linz et al., 2019), so it is important to bear this in mind in interpretation of the results presented here. Nevertheless, we compute the diagnostics for the residual circulation in a self-consistent manner in the models and reanalyses to enable comparison with earlier multi-model studies (e.g. Butchart et al., 2006, Butchart et al., 2010, SPARC, 2010)."*

2) Although the method used here to do the multiple linear regression is somewhat standard, it does not actually remove all of the variability that is related to these signals. In particular, the relative phasing of the QBO and the annual cycle will cause an enhanced wintertime upwelling in some years, so the result that the leftover variability is highly correlated could be related to this. Randel et al. 1999 have a nice treatment. I would consider using the method in that paper (described on p. 458, see also Randel and Wu 1996) to determine if the relative timing of the annual cycle and QBO is the cause of the covariation of the nudged time series. This phenomenon is also mentioned in the cited Baldwin et al. 2001 QBO review.

Thank you for raising this interesting point. The MLR model is applied to the annual mean TUMF timeseries and hence the QBO terms are resolved at an annual timescale. Therefore, the method presented by Randel et al. (1999) to account for interannual variations in the relative phasing of the QBO with the annual cycle is not readily applicable in our analysis. We further note that all of the models except CMAM and MRI-ESMr1 also nudge the phase of the QBO in the REF-C1 simulations (Morgenstern et al., 2017). Hence, if this is a main factor for explaining the MLR residuals, we expect that such an effect would also be seen in the REF-C1 simulations. However, the temporal correlation of the residuals is much weaker in the REF-C1 experiments for those models that nudge the QBO, suggesting this is not a major factor for explaining the correlated residuals in the REFC1-SD experiments.

3) The authors find that the reanalyses differ from the nudged runs which in turn differ from the free running models. I would like to see some presentation of the difference between the mean states of the reanalyses and the mean states of the models without nudging. The result that the models tend to have stronger upwelling in their SD runs could be caused by a systematic low bias in the model temperature in the tropics compared to the reanalysis temperature.The repeated

nudging with a primarily positive value would then cause an apparent increase in upwelling, without changing the underlying model physics that are the cause of any systematic mean state bias. To understand why the nudging does what it does, I think the difference between the free running models and the reanalyses needs to be explored in much more detail–perhaps for just one model. A continuous spurious forcing because of a mismatch in the mean state (and likely also in the seasonal cycle) should impact the mean while leaving the interannual variability alone (line 612).

Thank you for this suggestion. We present below the percent differences in zonal mean temperature in the stratosphere between the REF-C1 simulations and the respective reanalysis they are nudged towards (stippling denotes statistical significance at 95% based on a two-tailed Student's t-test); the equivalent differences for the REFC1-SD simulations are also shown in the right panels (note the reduced colour scale).

Most models show a cold bias relative to the reanalyses in the tropical tropopause layer, except for MRI-ESMr1 which shows a warm bias. The two versions of EMAC show the largest relative cold bias in the TTL. Much of this bias is alleviated in the nudged simulations, as expected, but EMAC (l47/90), CESM1-WACCM and SOCOL3 still show a (smaller) cold bias in the TTL. Larger differences remain above the level where nudging ceased to be imposed (e.g. 10 hPa in EMAC and 40 hPa in MRI-ESM1r1). The opposite sign of the TTL temperature bias in the free-running MRI-ESMr1 model provides a useful test bed for the point raised by the reviewer. The REFC1-SD MRI-ESMr1 simulation still shows stronger upwelling near the equator compared to the free-running model (Figure 3c) despite having a warm bias; this result is similar to the effect of nudging in the other models that show a cold TTL bias. Hence from this cursory investigation, we do not identify a systematic relationship between the intrinsic model bias and the effect of the nudging on the residual circulation. To get a better handle on this, a detailed analysis of the sensitivities within one model to relaxation timescales and nudging parameters such as the altitude range etc. would be needed in order to robustly establish how nudging affects the mean residual circulation.

[Figure]

**Minor comments**

I find the second paragraph of the introduction to be lacking. Although Abalos et al. (2015) did conclude that there was likely a 2-5%/decade trend in the circulation, they found substantial differences with different calculations for the circulation strength. Ploeger et al. (2019) have a new paper out on the inconsistency of the BDC trends for the different reanalyses, looking at the age spectra. Age of Air is mentioned somewhat abruptly and with no explanation. The results are also misrepresented. Engel et al. 2009 did not find a statistically significant increase in the mean age of air in the midlatitudes. They found no statistically significant change. There are limitations both on the availability of measurements and on the interpretation of age of air as a direct proxy for the residual circulation, and much more recent papers discuss this, rather than Waugh and Hall 2002. There are new satellite based measurements–ACE-FTS (e.g. Ray et al. 2016), MIPAS (Haenel et al. 2015), and MLS (Linz et al. 2017)–and new measurements from Air Core (Engel et al. 2017). The interpretation of the data is also difficult because of the separation of the residual circulation from the mixing (e.g. Garny et al. 2014, Ploeger et al 2015, as mentioned in the first paragraph–perhaps just move this comment to this paragraph?). The only attempt to calculate the residual circulation from tracer data did not calculate the residual circulation directly, but instead the diabatic circulation (Linz et al. 2017), and they did not calculate trends.

Thank you for this comment. We have expanded this part of the introduction to give a more balanced and detailed synopsis of the literature:

*"Past studies have shown substantial spread across models in the mean strength of the residual circulation (e.g. Butchart et al., 2010). Nevertheless, climate and chemistry-climate models (CCMs) consistently simulate a long-term strengthening of the residual circulation with an increase of ~2% decade$^{-1}$ (e.g. Butchart et al., 2010; Hardiman et al., 2014), though there are differences across models in the relative contribution to trends from resolved and parameterized wave forcing. Reanalysis datasets also suggest a strengthening of the residual circulation over the past several decades of the order 2-5% decade$^{-1}$ (Abalos et al., 2015; Miyazaki et al., 2016) apart from one (ERA-Interim) which shows a weakening of the deep branch of the BDC (Seviour et al., 2012; Abalos et al., 2015). However, reanalyses are subject to multiple caveats, particularly in their suitability for trend studies, and there are substantial differences in residual circulation trends calculated from the same reanalysis using different methods (Abalos et al., 2015). Given the limitations of reanalyses, evaluating the fidelity of model estimates of residual circulation variability and trends is challenging since there are no direct measurements of the residual circulation.*

*The only direct estimates of the stratospheric mass circulation come from tracer measurements, which can be used to calculate stratospheric age-of-air (AoA) (Kida, 1983; Schmidt and Khedim 1991; Waugh and Hall, 2002). AoA represents the combined effects of advection by the residual circulation and turbulent mixing processes, and as such cannot be directly related to the residual circulation. While progress has been made in separating the relative effects of these two factors for AoA calculated from models (Garny et al., 2014; Dietmuller et al., 2018, Eichinger et al., 2019, Šácha et al., 2019), using Lagrangian models driven by reanalysis fields (Ploeger et al., 2015a, Ploeger et al., 2015b, Ploeger and Birner, 2016, Ploeger et al., 2019) and comparing these effects in both CCMs and Lagrangian models (Dietmuller et al., 2017), this is more difficult to achieve in observations. Engel et al. (2009) and Stiller et al. (2012) used balloon-borne measurements of stratospheric trace gases and found a statistically non-significant increase in AoA in the middle stratosphere at northern mid-latitudes, which has been corroborated by more recent study using balloon-borne measurements at two midlatitude sites in the Northern Hemisphere (Engel et al., 2017). It has been hypothesised based on analyses of more recent satellite datasets, which have greater spatial and temporal coverage, that the subtropical AoA trends can be explained by a weakening of the mixing barriers at the edge of the tropical pipe (Neu and Plumb, 1999) that is masking the effects of an increase in tropical upwelling on AoA (Stiller et al., 2012; Haenel et al., 2015). In contrast with the AoA trends derived from observations, CCMs forced with observed sea-surface temperatures (SSTs), greenhouse gases and ozone-depleting substances show a decrease in AoA throughout the stratosphere (World Meteorological Organization, 2018; Li et al., 2018; Morgenstern et al.,*

*2018; Abalos et al., 2019; Polvani et al., 2019). Theoretical approaches based on the tropical leaky pipe model (Neu and Plumb, 1999) have shown promise for bridging the information on the stratospheric circulation derived from observations with outputs from GCMs/CCMs (Ray et al., 2016).*

*As explained above, it is difficult to infer information on the residual circulation from AoA without carefully accounting for the effects of mixing. More recent theoretical developments offer a means of calculating the diabatic circulation using stratospheric tracers (Linz et al., 2017), which is a promising avenue as this is more closely related to the residual circulation than AoA. The results of Linz et al. (2017) showed consistent estimates of the diabatic circulation in the lower stratosphere, but large uncertainties of up to a factor of two in the mean strength in the upper stratosphere. Targeted measurement strategies to better characterise long-term changes in the observed stratospheric meridional circulation have been proposed (Moore et al., 2014; Ray et al., 2016)."*

A more detailed explanation of what nudging is, how it is implemented, why it is used, and the motivation for this work would help the introduction. The current paragraph in the introduction (line 85) is good, and I think that including more specific examples would be very helpful. Similarly, in the conclusion, commentary on what these nudged runs could be useful for would be appreciated in addition to the entirely appropriate cautionary words.

Thank you for this comment. We have expanded this paragraph to include your suggestions above. The relevant part of the manuscript is now (nb: black italic below denote unchanged parts of the manuscript) :

*"In an attempt to obtain a closer comparison with observed stratospheric trace species, some studies have used model simulations with meteorological fields nudged or relaxed towards a reanalysis dataset (Jeuken et al., 1996); these include many studies of ozone variability and trends (e.g. van Aalst et al., 2003; Solomon et al., 2016; Hardiman et al., 2017b; Ball et al., 2018),* comparisons between models and satellite-based multi-species observational records (Froidevaux et al. 2019) *as well as the chemical and climatic effects of volcanic eruptions (Löffler et al., 2016; Solomon et al., 2016; Schmidt et al., 2018).* Nudged simulations have also been used to study mechanisms for dynamical coupling between the stratosphere and troposphere (Hitchcock and Simpson, 2014). *Nudging involves adding additional tendencies to the model equations* by constraining the simulated meteorological fields to reanalysis fields (Kunz et al., 2011). *Nudged variables can include horizontal winds (or divergence and vorticity), temperature, surface pressure, latent and sensible heat fluxes. However, vertical winds, which are small residual from horizontal divergence, are not nudged and the underlying model physics can yield quite different results from the reanalysis that they are nudged towards (Telford et al., 2008; Hardiman et al., 2017). A recent study by Orbe et al. (2018) analyzed tropospheric tracers*

*in nudged CCM simulations and found large differences in the distributions of the tracers, which could be partly traced to differences in the model convection schemes. They urged users to adopt a cautious approach when interpreting tracers in nudged simulations given their dependence on not only large-scale flow but also sub-grid parameterizations.* In contrast, critical evaluation of the stratospheric residual circulation in models relaxed to reanalysis fields has been rather limited to date."

As for the conclusions the now expanded relevant part of the manuscript (second from last paragraph of the manuscript):

*"Despite the limitations described here, some success has been reported in studies that used nudged simulations to investigate specific meteorological events, such as Sudden Stratospheric Warmings, and in particular for exploring processes beyond the top of the nudging region in the Mesosphere-Lower Thermosphere (e.g., Tweedy et al., 2013; Chandran and Collins, 2014; Pedatella et al., 2014). In order to reduce discrepancies between nudged and free-running simulations, various nudging techniques have been investigated. The role of gravity waves in the error growth that the nudging introduces over time has been highlighted for a single model (Smith et al., 2017). Constraining just the horizontal winds without the temperature was found to be a good strategy when investigating the aerosol indirect effects without affecting significantly the mean state (Zhang et al., 2014)...*

*...A dedicated study of the sensitivities within one model to relaxation timescales, nudging parameters, nudging height range, and model resolution would be needed to offer a detailed explanation for these differences."*

Do any of the "free running" models have a nudged QBO, and if so, which ones?

All of the free-running models presented in the study apart from CMAM and MRI-ESMr1 impose a QBO "nudging" in their simulations as mentioned in the last paragraph of section 3.4 (Table S8 of supplement of Morgenstern et al., 2017).

Since the focus of this paper is really on the difference between SD and free running models, it might make sense to omit the models that do not include both. I respect the authors choice to have those comparisons in the supplementary information. However, I think I would find the plots more manageable without the additional lines.

Thank you for this recommendation. In response to this comment and also the comments of reviewer 1 we have removed the models which did not perform the SD simulation from the main body of the paper (GEOSCCM, NIWA-UKCA and ULAQ-CCM) and the main text now focuses

on the seven models for which both REF-C1 and REFC1-SD simulations are available. For completeness a subset of diagnostics are shown for those models in the supplementary material.

Line 106: aren't they necessarily inconsistent?

We have corrected this to: "*will add forcings that are inconsistent with the model state.*"

Lines 100-129: excellent discussion.

Thank you for the positive comment!

Line 153: Do you mean Table 2?

Thank you for spotting this, we have now corrected this.

Line 155 etc.: More details here would be good. I don't understand what is meant by T (with wave-0), and I assume that "temperature" is also T (for CMAM), but if that's true then they all nudge temperature so maybe mention that? Including the sentence "Nudging timescales range from 6 - 50 hours." (or some equivalent) would also emphasize how different these treatments are.

We have added some more details here as per your recommendation. In Table 2, temperature is referred to consistently as T. The wave-0 was intended to refer to the additional nudging of the global mean temperature in EMAC. We now state this explicitly. We have also changed L152-153 in the text to: "*The REF-C1SD simulations nudge temperature and other meteorological fields such as horizontal winds, vorticity and divergence and some surface fields (Table 2), while the chemical fields are left to evolve freely. The nudging timescales range from 6 - 50 hours and the height range over which nudging is applied varies (Table 2).*"

Line 159-160: Table 1?

Thank you for spotting this, we have now corrected this.

Line 168: "primary variable"–not sure what is meant by this

We meant prognostic. We have now corrected this.

Line 442: I have trouble seeing this from the plot. Could you provide the numbers? Is the difference in the peak-to-peak amplitude significant?

Yes, due to the congestion of the multiple lines is not that easy to see but the difference is not significant. This sentence now includes the numbers of the peak-to-peak amplitude

*"Comparing the MMM annual cycle of the REF-C1 runs with the MMM REF-C1SD of Figure 7b reveals that on average the REFC1-SD models show a slightly larger peak-to-peak amplitude (0.22 mm s⁻¹ vs. 0.19 mm s⁻¹) in the annual cycle."*

Line 476. . .: How do these results relate to Abalos et al. 2014, who found such a strong dependence of the upwelling on the location of the EPFD?

Thank you for this interesting point. We agree it would be interesting to examine how the patterns of EPFD vary between the free running and nudged simulations and how this relates to fluctuations in upwelling. This work is planned for a follow-up study.

Line 488-90: I found this sentence confusing.

Thank you for spotting this unclear sentence. We have removed this sentence as it was related to an earlier sentence on lines 483-485, which has been edited to:

*"This result indicates that although nudging does not constrain the mean residual circulation, it does constrain the interannual variability and produces very similar contributions to variability across models from both resolved and parameterized wave forcing.*

Line 498: remove "due to"

Done.

Section 3.6: Nice.

Thank you for the positive comment!

Line 606.: This paragraph is very long and contains a lot of important information. Turn into two or even three shorter paragraphs?

Thank you for this comment, we have now turned this paragraph into three shorter ones to improve the clarity.

Figures: Consider stippling the insignificant part of the plot. I read this on a number of different devices, and on some it looked fine, but on one of them the plots were impossible to see because the colors weren't visible behind the stippling. This wasn't an issue for the last two figures for some reason.

It may take a while for these plots to be loaded properly in a pdf viewer, as they are quite large files. We changed the subtleties of the stippling patterns as seen in the figures and hopefully they are more clear now.

It could be useful to have models that are nudged to the same reanalysis plotted in the same color family to aid interpretation of the complicated line plots.

We have now colour coded the model results nudged towards ERA-I in red(-ish) colours. We hope that without the additional lines of the extra models, the results are more clear.

**References:**
Linz et al. 2019: https://www.atmos-chemphys.net/19/5069/2019/acp-19-5069-2019.html
Randel et al. 1999 https://acomstaff.acom.ucar.edu/randel/JAS%201999.pdf
Randel and Wu 1996
https://journals.ametsoc.org/doi/10.1175/1520-0469%281996%29053%3C2546%3AIOTOQI%3E2.0.CO%3B2
Ploeger et al. 2019 https://www.atmos-chem-phys.net/19/6085/2019/
Ray et al. 2016 https://agupubs.onlinelibrary.wiley.com/doi/full/10.1002/2015JD024447
Haenel et al. 2015 https://www.atmos-chem-phys.net/15/13161/2015/acp-15-13161-2015.html
Linz et al. 2017 https://www.nature.com/articles/ngeo3013
Abalos et al. 2014 https://journals.ametsoc.org/doi/10.1175/JAS-D-13-0366.1

Abalos, M., Legras, B., Ploeger, F. and Randel, W. J.: Evaluating the advective Brewer-Dobson circulation in three reanalyses for the period 1979-2012, J. Geophys. Res. Atmos., 120(15), 7534–7554, doi:10.1002/2015JD023182, 2015.

Abalos, M., Polvani, L., Calvo, N., Kinnison, D., Ploeger, F., Randel, W. and Solomon, S.: New Insights on the Impact of Ozone-Depleting Substances on the Brewer-Dobson Circulation, J. Geophys. Res. Atmos., 124(5), 2435–2451, doi:10.1029/2018JD029301, 2019.

Andrews, D. G., Holton, J. R. and Leovy, C. B.: Middle Atmosphere Dynamics, International Geophysical Series, Vol. 40, 1987.

Butchart, N.: The Brewer-Dobson circulation, Rev. Geophys., 52(2), 157–184, doi:10.1002/2013RG000448, 2014.

Butchart, N., Scaife, A. A., Bourqui, M., Grandpré, J., Hare, S. H. E., Kettleborough, J., Langematz, U., Manzini, E., Sassi, F., Shibata, K., Shindell, D. and Sigmond, M.: Simulations of anthropogenic change in the strength of the Brewer-Dobson circulation, Clim. Dyn., 27(7–8), 727–741, doi:10.1007/s00382-006-0162-4, 2006.

Butchart, N., Cionni, I., Eyring, V., Shepherd, T. G., Waugh, D. W., Akiyoshi, H., Austin, J., Brühl, C., Chipperfield, M. P., Cordero, E., Dameris, M., Deckert, R., Dhomse, S., Frith, S. M., Garcia, R. R., Gettelman, A., Giorgetta, M. A., Kinnison, D. E., Li, F., Mancini, E., McLandress, C., Pawson, S., Pitari, G., Plummer, D. A., Rozanov, E., Sassi, F., Scinocca, J. F., Shibata, K., Steil, B. and Tian, W.: Chemistry–Climate Model Simulations of Twenty-First Century Stratospheric Climate and Circulation Changes, J. Clim., 23(20), 5349–5374, doi:10.1175/2010JCLI3404.1, 2010.

Chandran, A. and Collins, R. L.: Stratospheric sudden warming effects on winds and temperature in the middle atmosphere at middle and low latitudes: a study using WACCM, Ann. Geophys., 32(7), 859–874, doi:10.5194/angeo-32-859-2014, 2014.

Dietmüller, S., Garny, H., Plöger, F., Jöckel, P. and Cai, D.: Effects of mixing on resolved and unresolved scales on stratospheric age of air, Atmos. Chem. Phys., 17(12), 7703–7719, doi:10.5194/acp-17-7703-2017, 2017.

Dietmüller, S., Eichinger, R., Garny, H., Birner, T., Boenisch, H., Pitari, G., Mancini, E., Visioni, D., Stenke, A., Revell, L., Rozanov, E., Plummer, D. A., Scinocca, J., Jöckel, P., Oman, L., Deushi, M., Kiyotaka, S., Kinnison, D. E., Garcia, R., Morgenstern, O., Zeng, G., Stone, K. A. and Schofield, R.: Quantifying the effect of mixing on the mean age of air in CCMVal-2 and CCMI-1 models, Atmos. Chem. Phys., 18(9), 6699–6720, doi:10.5194/acp-18-6699-2018, 2018.

Engel, A., Möbius, T., Bönisch, H., Schmidt, U., Heinz, R., Levin, I., Atlas, E., Aoki, S., Nakazawa, T., Sugawara, S., Moore, F., Hurst, D., Elkins, J., Schauffler, S., Andrews, A. and Boering, K.: Age of stratospheric air unchanged within uncertainties over the past 30 years, Nat. Geosci., 2(1), 28–31, doi:10.1038/ngeo388, 2009.

Froidevaux, L., Kinnison, D. E., Wang, R., Anderson, J. and Fuller, R. A.: Evaluation of CESM1 (WACCM) free-running and specified dynamics atmospheric composition simulations using global multispecies satellite data records, Atmos. Chem. Phys., 19(7), 4783–4821, doi:10.5194/acp-19-4783-2019, 2019.

Garny, H., Birner, T., Bönisch, H. and Bunzel, F.: The effects of mixing on age of air, J. Geophys. Res., 119(12), 7015–7034, doi:10.1002/2013JD021417, 2014.

Haynes, P. H., McIntyre, M. E., Shepherd, T. G., Marks, C. J. and Shine, K. P.: On the "Downward Control" of Extratropical Diabatic Circulations by Eddy-Induced Mean Zonal Forces, J. Atmos. Sci., 48(4), 651–678, doi:10.1175/1520-0469(1991)048<0651:OTCOED>2.0.CO;2, 1991.

Hitchcock, P. and Simpson, I. R.: The Downward Influence of Stratospheric Sudden Warmings*, J. Atmos. Sci., 71(10), 3856–3876, doi:10.1175/jas-d-14-0012.1, 2014.

Kunz, A., Pan, L. L., Konopka, P., Kinnison, D. E. and Tilmes, S.: Chemical and dynamical discontinuity at the extratropical tropopause based on START08 and WACCM analyses, J. Geophys. Res. Atmos., 116(24), 1–15, doi:10.1029/2011JD016686, 2011.

Li, F., Newman, P., Pawson, S. and Perlwitz, J.: Effects of Greenhouse Gas Increase and Stratospheric Ozone Depletion on Stratospheric Mean Age of Air in 1960-2010, J. Geophys. Res. Atmos., (2010), 2098–2110, doi:10.1002/2017JD027562, 2018.

Miyazaki, K., Iwasaki, T., Kawatani, Y., Kobayashi, C., Sugawara, S. and Hegglin, M. I.: Inter-comparison of stratospheric mean-meridional circulation and eddy mixing among six reanalysis data sets, Atmos. Chem. Phys., 16(10), 6131–6152, doi:10.5194/acp-16-6131-2016, 2016.

Moore, F. L., Ray, E. A., Rosenlof, K. H., Elkins, J. W., Tans, P., Karion, A. and Sweeney, C.: A Cost-Effective Trace Gas Measurement Program for Long-Term Monitoring of the Stratospheric Circulation, Bull. Am. Meteorol. Soc., 95(1), 147–155, doi:10.1175/BAMS-D-12-00153.1, 2014.

Morgenstern, O., Hegglin, M. I., Rozanov, E., O'Connor, F. M., Abraham, N. L., Akiyoshi, H., Archibald, A. T., Bekki, S., Butchart, N., Chipperfield, M. P., Deushi, M., Dhomse, S. S., Garcia, R. R., Hardiman, S. C., Horowitz, L. W., Jöckel, P., Josse, B., Kinnison, D., Lin, M., Mancini, E., Manyin, M. E., Marchand, M., Marécal, V., Michou, M., Oman, L. D., Pitari, G., Plummer, D. A., Revell, L. E., Saint-Martin, D., Schofield, R., Stenke, A., Stone, K., Sudo, K., Tanaka, T. Y., Tilmes, S., Yamashita, Y., Yoshida, K. and Zeng, G.: Review of the global models used within phase 1 of the Chemistry–Climate Model Initiative (CCMI), Geosci. Model Dev., 10(2), 639–671, doi:10.5194/gmd-10-639-2017, 2017.

Morgenstern, O., Stone, K. A., Schofield, R., Akiyoshi, H., Yamashita, Y., Kinnison, D. E., Garcia, R. R., Sudo, K., Plummer, D. A., Scinocca, J., Oman, L. D., Manyin, M. E., Zeng, G.,

Rozanov, E., Stenke, A., Revell, L. E., Pitari, G., Mancini, E., Di Genova, G., Visioni, D., Dhomse, S. S. and Chipperfield, M. P.: Ozone sensitivity to varying greenhouse gases and ozone-depleting substances in CCMI-1 simulations, Atmos. Chem. Phys., 18(2), 1091–1114, doi:10.5194/acp-18-1091-2018, 2018.

Neu, J. L. and Plumb, R. A.: Age of air in a "leaky pipe" model of stratospheric transport, J. Geophys. Res. Atmos., 104(D16), 19243–19255, doi:10.1029/1999JD900251, 1999.

Pedatella, N. M., Fuller-Rowell, T., Wang, H., Jin, H., Miyoshi, Y., Fujiwara, H., Shinagawa, H., Liu, H.-L., Sassi, F., Schmidt, H., Matthias, V. and Goncharenko, L.: The neutral dynamics during the 2009 sudden stratosphere warming simulated by different whole atmosphere models, J. Geophys. Res. Sp. Phys., 119(2), 1306–1324, doi:10.1002/2013JA019421, 2014.

Ploeger, F. and Birner, T.: Seasonal and inter-annual variability of lower stratospheric age of air spectra, Atmos. Chem. Phys., 16(15), 10195–10213, doi:10.5194/acp-16-10195-2016, 2016.

Ploeger, F., Abalos, M., Birner, T., Konopka, P., Legras, B., Müller, R. and Riese, M.: Quantifying the effects of mixing and residual circulation on trends of stratospheric mean age of air, Geophys. Res. Lett., 42(6), 2047–2054, doi:10.1002/2014GL062927, 2015a.

Ploeger, F., Riese, M., Haenel, F., Konopka, P., M??ller, R. and Stiller, G.: Variability of stratospheric mean age of air and of the local effects of residual circulation and eddy mixing, J. Geophys. Res. Atmos., 120(2), 716–733, doi:10.1002/2014JD022468, 2015b.

Ploeger, F., Legras, B., Charlesworth, E., Yan, X., Diallo, M., Konopka, P., Birner, T., Tao, M., Engel, A. and Riese, M.: How robust are stratospheric age of air trends from different reanalyses?, Atmos. Chem. Phys. Discuss., 1–32, doi:10.5194/acp-2018-1281, 2019.

Polvani, L. M., Wang, L., Abalos, M., Butchart, N., Chipperfield, M. P., Dameris, M., Deushi, M., Dhomse, S. S., Jöckel, P., Kinnison, D., Michou, M., Morgenstern, O., Oman, L. D., Plummer, D. A. and Stone, K. A.: Large impacts, past and future, of ozone‐depleting substances on Brewer‐Dobson circulation trends: A multi‐model assessment, J. Geophys. Res. Atmos., 2018JD029516, doi:10.1029/2018JD029516, 2019.

Rosenlof, K. H.: Seasonal cycle of the residual mean meridional circulation in the stratosphere, J. Geophys. Res., 100(D3), 5173, doi:10.1029/94JD03122, 1995.

Šácha, P., Eichinger, R., Garny, H., Pišoft, P., Dietmüller, S., de la Torre, L., Plummer, D. A., Jöckel, P., Morgenstern, O., Zeng, G., Butchart, N. and Añel, J. A.: Extratropical Age of Air

trends and causative factors in climate projection simulations, Atmos. Chem. Phys. Discuss., (2018), 1–37, doi:10.5194/acp-2018-1310, 2019.

Seviour, W. J. M., Butchart, N. and Hardiman, S. C.: The Brewer-Dobson circulation inferred from ERA-Interim, Q. J. R. Meteorol. Soc., 138(665), 878–888, doi:10.1002/qj.966, 2012.

Smith, A. K., Pedatella, N. M., Marsh, D. R. and Matsuo, T.: On the Dynamical Control of the Mesosphere–Lower Thermosphere by the Lower and Middle Atmosphere, J. Atmos. Sci., 74(3), 933–947, doi:10.1175/JAS-D-16-0226.1, 2017.

SPARC: SPARC CCMVal Report on the Evaluation of Chemistry-Climate Models. V. Eyring, T. Shepherd and D. Waugh (Eds.), SPARC Rep. No. 5, WCRP-30/2010,WMO/TD-No.40, available at www.sparc-climate.org/publications/sp [online] Available from: http://www.sparc-climate.org/publications/sparc-reports/sparc-report-no5/, 2010.

Stiller, G. P., Von Clarmann, T., Haenel, F., Funke, B., Glatthor, N., Grabowski, U., Kellmann, S., Kiefer, M., Linden, A., Lossow, S. and López-Puertas, M.: Observed temporal evolution of global mean age of stratospheric air for the 2002 to 2010 period, Atmos. Chem. Phys., 12(7), 3311–3331, doi:10.5194/acp-12-3311-2012, 2012.

Tweedy, O. V., Limpasuvan, V., Orsolini, Y. J., Smith, A. K., Garcia, R. R., Kinnison, D., Randall, C. E., Kvissel, O. K., Stordal, F., Harvey, V. L. and Chandran, A.: Nighttime secondary ozone layer during major stratospheric sudden warmings in specified-dynamics WACCM, J. Geophys. Res. Atmos., 118(15), 8346–8358, doi:10.1002/jgrd.50651, 2013.

World Meteorological Organization (WMO): Scientific Assessment of Ozone Depletion: 2018, Global Ozone Research and Monitoring Project – Report No. 58, Geneva, Switzerland, 2018., 2018.

Zhang, K., Wan, H., Liu, X., Ghan, S. J., Kooperman, G. J., Ma, P. L., Rasch, P. J., Neubauer, D. and Lohmann, U.: Technical note: On the use of nudging for aerosol-climate model intercomparison studies, Atmos. Chem. Phys., 14(16), 8631–8645, doi:10.5194/acp-14-8631-2014, 2014.

---

## Author Response (AR1)

**Author response to referee comments on Chrysanthou et al. "The effect of atmospheric nudging on the stratospheric residual circulation in chemistry-climate models" submitted to ACPD**

**Reply to Anonymous Referee #1**

**Recommendation: Publication after minor revision**

The paper is very well organised and written. The topic discussed here – the effect of nudging CCMs on the stratospheric residual circulation – is of very high relevance, because nudged or specified dynamics CCM simulations are a common tool for analysing and interpreting observed changes of the stratospheric trace gas composition (e.g. Froidevaux et al. 2019). This work will most likely trigger many more studies related to the problems induced by driving CCMs (or more generally GCMs) with specified dynamics derived from meteorological reanalysis However, some further steps to disentangle the main source(s) of the problem could already be done in this paper or the author could at least explain, why this could not be done here (see comments below). The paper should be submitted after addressing the comments below.

Thank you for your positive comments and suggestions for improving the paper. We reply to the specific points raised below in red.

**General comments:**

The topic is important, and it is time to carefully analyse the effect of nudging CCMs towards meteorological reanalysis data (RA) on the simulated stratospheric residual circulation and consequently also on the transport of chemical species. The latter is not the focus of this paper here, but the results in this paper will hopefully stimulate further studies on the impact of nudging on stratospheric tracer transport and composition, e.g. on CCM-SD simulations of the recent lower stratospheric ozone trend reported by Ball et al. (2018).

I highly appreciate the topic and the great effort made by the authors and I agree in general to the conclusions derived from the results presented in this very valuable multi-model analysis. However, the manuscript should and could be focused more on the main topic and the new aspect that is clearly announced in the title: "The effect of nudging on the stratospheric …". The differences between specified dynamic (SD) and free-running (FR) simulations is to my point of view important, but only one (minor) aspect of this paper and should be contrasted with the differences between the SD simulations and the reanalysis datasets (RA) for the individual measures of the stratospheric residual circulation. This would describe the quantitative effect of nudging CCMs.

As outlined above, the paper is to my opinion focused too much on the difference between FR and SD simulations – this becomes obvious in most Figures where only the differences of REF-C1SD-REF-C1 (SD-FR) are shown and the differences REF-C1SD-Reanalysis (SD-RA) is missing. The systematic analysis of SD-RA and SD-FR for all the metrics of stratospheric residual circulation would help to get a more quantitative view and help to better understand the effect of nudging on the stratospheric residual circulation. It is clear that the underlying mechanisms that trigger the observed discrepancies in the stratospheric residual circulation between nudged CCMs and reanalysis is not in the scope of this paper, but the differences SD-RA should be shown in the same way as the differences SD-FR, so that the effect of nudging – reflected by the differences to the reanalysis – could be contrasted and discussed more quantitatively compared to the differences between specified dynamics and free-running simulations.

Frankly spoken, it would be better to streamline this paper and to concentrate only on the CCMs that have carried out SD and FR simulations in a consistent way. To my opinion, the other non-nudged models could not add any information to the main topic of the paper – the effect of nudging CCMs. The consequence would be to skip 4 out of 9 models or 4 out of 10 simulations which do not provide SD simulations. Including 4 non-nudged CCMs to the multi- model-mean (MMM) of the FR simulations is not really helpful to analyse the effect of CCM nudging. In the best case, the results are not blurred by these models, but to my opinion the analysis of SD-FR vs. SD-RA for the 6 simulations providing all relevant information would help to get a much more stringent and systematic analysis of the role and impact of nudging CCMs on their stratospheric residual circulation. This paper would gain, if the also very interesting inter-model stratospheric residual circulation comparison of all participating CCMI models would be done in a separate paper.

In response to this comment we have made several changes to the manuscript. As suggested, we have removed the models that did not perform the SD simulation from the main body of the paper (GEOSCCM, NIWA-UKCA and ULAQ-CCM). For completeness a subset of diagnostics is shown for those models in the supplementary material, but the main text now concentrates on the 7 models that performed both free running and nudged simulations (nb: since the initial submission we have now obtained SD data for SOCOL3).

Additionally, in line with the reviewer's recommendation, we now include additional figures showing differences between the SD experiments and the reanalysis (RA) datasets. Specifically, we have moved Figure S2 from the supplement into the main text and have added reanalysis data (now Figure 2), and have added additional panels to Figures 3 and 4 showing the SD - RA differences for comparison with the SD - REF-C1 results. Additionally, the supplement now

includes figures showing differences in the climatological wbar_star between each SD model and the respective reanalysis it was nudged towards (Figure S1). We also show the 70 hPa wbar_star differences in the climatological annual cycle between each SD model and the respective reanalysis it was nudged towards (Figure S3). The MLR analysis presented in section 3.5 and the trend analysis in section 3.6 was also performed for the RA datasets and the results are discussed in the main text with the relevant figures also presented in the Supplement (Figures S4 and S7). We hope that the reviewer finds that these changes to the manuscript address the comment about focusing the paper on the SD simulations and comparing them more closely with the RA data.

**Specific comments:**

L.79: "… from which age-of-air (AoA) can be estimated (Waugh and Hall, 2002)." For clarity, I would suggest to add: …, however AoA represents the combined effects of residual circulation and mixing. Waugh and Hall (2002) is a review paper and not the original reference for AoA from tracer measurements and the credit should be given to the original work. To my knowledge, the first stratospheric AoA estimates from tracer measurements has been reported by Schmidt and Khedim (1991) and the concept of AoA was first applied to the stratosphere by Kida (1983).

Thank you for highlighting the original work that defined the AoA concept. We have added the references to Kida (1983) and Schmidt and Khedim (1991). Additionally, we have added a word of caution regarding what AoA represents :

*"tracer measurements from which age-of-air (AoA) can be estimated (Kida, 1983; Schmidt and Khedim (1991); Waugh and Hall, 2002). Note that AoA represents the combined effects of both the residual circulation and mixing processes."*

L.79-80: Engel et al. (2009) found no decrease in AoA, but the observed increase of AoA was statistically non-significant.

Thank you for pointing this out. We have now corrected this sentence:

*"Engel et al. (2009) and Stiller et al. (2012) using observations from tracers found a statistically non-significant increase in AoA in the middle stratosphere at northern midlatitudes"*

L.191: The reference for the citation "(Rosenlof, 1995)" is missing.

Thank you for spotting this. We have now added the reference.

L. 276-284: To my opinion, the Supplemental Figure 2 should be moved to the manuscript, because the topic is discussed here and the TA-latitudes are a simple measure and very indicative for the different structure of the residual circulations derived from SD and FR simulations. This Figure could be extended with the weighted mean TA-latitudes of the different reanalysis datasets (depending how often they are used for SD runs: 4x ERA-I, 1x MERRA and 1x JRA55). This would give a measure of the differences between the SD simulations and the reanalysis induced by nudging.

We have moved the Supplemental Figure 2 to the main text and have added the reanalysis datasets to the figure. We decided not to add a "multi-reanalysis mean" as we think it is important to highlight the differences between the reanalysis themselves.

The relevant part of the manuscript that was referring to this figure has also been expanded:

*"To show the differences in vertical structure of the upwelling/downwelling regions between the REF-C1SD and REF-C1 experiments and between the REF-C1SD and RA, Figure 2 shows vertical profiles of the climatological TA latitudes for MMM-C1, MMM-SD and the three RA datasets used for nudging. It should be noted that since five of the REFC1-SD models were nudged towards ERA-I the MMM-SD is quite heavily weighted towards ERA-I. The REF-C1SD experiments consistently simulate a wider NH TA latitude (Figure 2b) throughout most of the stratosphere as compared to both the REF-C1 runs and the RA. In contrast, in the SH up to 30 hPa the TA latitude for MMM-SD is consistently wider than both MMM-C1 and RA (Figure 2a), while in the middle and upper stratosphere this reverses and MMM-SD shows a narrower Southern hemisphere upwelling region than both MMM-C1 and RA. Interestingly, above 10 hPa in the SH (Figure 2a) the MMM-SD does not show a widening of the upwelling region as seen in the RA. This is reflected in the structural differences in w\* in the SH upper stratosphere found in some models (Supplemental Figure S1). It should be noted though that the differences in TA latitudes between the REF-C1 and equivalent REF-C1SD simulations are comparable to the differences found between the three RA datasets."*

L327-338: Here, the authors discuss some of the differences between SD and reanalysis for wbar_star on the 70 hPa level as shown in Fig. 2b). As noted in the general comments, it would be helpful to add here the corresponding differences SD-RA.

As explained in the reply to comment 1, we have now added the differences between each SD model and the respective reanalysis it was nudged towards in Figure 3d.

The relevant part of the manuscript was also expanded to discuss this new result :

*"Figure 3d shows the absolute differences in w\* between the REF-C1SD simulations and the respective RA dataset used for nudging. In the upwelling region, the REF-C1SD experiments generally show stronger upwelling around the equator than in the RA. The differences near 10 - 15°N highlight a lack of inter-hemispheric asymmetry in the double-peaked w\* structure in the REF-C1SD experiment compared to the RA. Outside of the tropical pipe, the REF-C1SD experiments show weaker downwelling, particularly in the NH mid-latitudes, while at subpolar/polar latitudes (>65° latitude) the REF-C1SD runs simulate consistently stronger downwelling than the respective RA. This difference in w\* at high latitudes between the REFC1-SD and RA datasets extends throughout the depth of the stratosphere (see Supplemental Figure S2)."*

L350-364: Here, the authors discuss the discrepancies between SD and RA using Figure 3b. Again, I would suggest to add a fourth panel (Figure 3d) that clearly shows the differences SD-RA (same as panel 3c) for the vertical profile of the climatological TUMF.

Thank you for this, as per your suggestion we have now added the differences between each SD model and the respective reanalysis it was nudged towards in Figure 4d.

The relevant part of the manuscript was also expanded based on this new result :

*"In the lower and middle stratosphere, most of the REF-C1SD models show a smaller TUMF compared to the RA (Figure 4d), while in the upper stratosphere the majority of the REF-C1SD models simulate higher TUMF values than the respective RA."*

And:

*"There is a high degree of similarity in the vertical structure of the differences in TUMF between the REFC1-SD simulations for EMAC-L47/L90 and SOCOL3 as compared to ERA-I, which may be related to similarities in the implementation of nudging in these models; for example, vorticity and divergence were nudged with the same relaxation parameters (see Table 2)."*

L.364-366: It is not clear to me, what the total spread here really is. Is it the standard deviation of all models TUMF in the range 100-30 hPa? Could you please clarify?

Thank you for noting that this was not stated in the manuscript. The model spread is quantified using the inter-model standard deviation. The relevant part of the text was updated to:

*"The inter-model standard deviation in TUMF in the lower stratosphere (100-30 hPa), where the largest spread is found, is up to $3.5 \times 10^9$ kg s$^{-1}$ and $3.3 \times 10^9$ kg s$^{-1}$ in the REFC1-SD and REF-C1 experiments, respectively."*

L. 377-379: The second model, for which TUMF from wbar_star exceeds TUMF from total wave forcing for the REF-C1 simulations, is the GEOSCCM and not the EMAC-L90MA.

Thank you for pointing this out, we have corrected this. However, as GEOSCCM has been removed from the main part of the study, the text was updated to:

*"...the estimated TUMF from the total wave forcing for the majority of the models (apart from CESM1-WACCM), clearly exceeds the TUMF calculated directly from w\*..."*

L392-397: It sounds unreasonable to me that the SD-RA discrepancies for TUMF should be related to the individual RA dataset, it is much more reasonable that the individual model itself and how the nudging is implemented is causing these differences.

We agree that the SD-RA differences are not RA specific and that the method for implementing nudging is important. We have edited the text as follows :

*"However, given there are substantial differences in TUMF amongst REFC1-SD models nudged to the same RA dataset, it is likely that the differences between REFC1-SD and RA are related to how nudging was implemented in each model; a wide variety of relaxation timescales and vertical nudging ranges were used by the models."*

L.401-404: Is the large positive difference between TUMF derived from wbar_star and derived from total wave forcing not what one would expect, if one assumes that nudging CCMs might lead to additional forcings to the residual stratospheric circulation induced by the inconsistencies with the modelled physics and not by wave breaking? In my naive way of thinking, I would expect that TUMF from total wave forcing derived from the downward control principle (DCP) is at least equal or maybe slightly larger than the directly calculated TUMF from the residual vertical velocity due to the possible wave-wave interactions and the slight imperfections of the "exact downward control" in transient (nonsteady state) cases. The internally more consistent free-running simulations without the additional tendencies induced by nudging seem to corroborate this hypothesis. For SD simulations only the EMAC-L90 model behaves different (more realistic?) with a slightly larger TUMF derived from wave forcings. Could this be the consequence of a different nudging procedure? The EMAC-L90 and to a less degree the MRI-ESM1r1 are also the only models for which TUMF at 70 hPa derived from the SD simulation is smaller than that derived from the applied RA datasets, ERA-I and JRA55

respectively. Both models also show the smallest difference between directly calculated TUMF and TUMF derived from total wave forcing. To my opinion, it would be worth to extend the discussion on these topics (RA vs. SD) a bit more to focus the paper more on the main topic stated in the paper's title.

This is an interesting hypothesis and appears to be corroborated by most models from the 2010 SPARC CCMVal report (Figure 4.10(a), p.121, Chapter 4). The free-running simulations are internally consistent, and the imperfect match between the direct and DCP estimates shows the limitation of certain assumptions in the way the DCP is applied here, such as assuming a steady-state response to a steady mechanical forcing. As noted by the reviewer, the nudging procedure adds an additional non-physical tendency in the model equations, but from our perspective it is difficult to infer how this acts to decouple the wave forcing from the modelled residual circulation. For example, the nudging technique in EMAC-L90 is identical to that in EMAC-L47 (Morgenstern et al., 2017), so the different behaviour of those two models in Figure 4b indicate their must be sensitivities to multiple factors such as vertical resolution. In the EMAC-L90 there are far more levels throughout the stratosphere which might affect the results in that vicinity (Jockel et al 2016, ESCiMo_supplement, Figures S4 and S5). As a consequence, a dedicated study of the sensitivities of the residual circulation within one model to factors such as relaxation timescales, nudging parameters, nudging height range, vertical resolution etc would be needed to offer a detailed explanation for these differences. We therefore think it is prudent to adopt a cautious approach in trying to explain the differences between the direct and DCP calculations for the REFC1-SD experiments as these may be the result of different factors for different models.

To accommodate the points raised by the reviewer about the differences between the direct and DCP estimates in the REF-C1 experiments, we have added the following paragraph to the main text:

*"On the contrary, as seen from the smaller residuals in Figure 5a, the TUMF estimated from the total wave forcing is similar to or slightly larger than the directly calculated TUMF in most of the REF-C1 simulations. This is comparable to results for the free-running CCMVal-2 models in SPARC (2010) (Figure 4.10(a), p.121). Since these simulations are internally consistent, the imperfect match indicates that the downward control principle as applied here relies on the applicability of certain assumptions such as the system being in a steady-state in response to a steady mechanical forcing (Haynes et al., 1991). The larger differences between the direct and downward control principle TUMF calculations in the REFC1-SD simulations shows that nudging adds an additional non-physical tendency in the model equations which acts to decouple the wave forcing from the residual circulation; the details of how this decoupling is manifested is*

*likely to vary from one model to another depending on multiple factors such as nudging timescales, nudging parameters, nudging height range, and model resolution."*

L.404-406: The author states that the lower directly calculated TUMF values for the higher vertically resolved EMAC-L90 compared to -L47 is in line with the finding for the SOCOL3 model reported by Revell et al. (2015b), i.e. 90 vs. 39 layers. This is true for the FR simulations which has been used in the sensitivity experiments by Revell et al. (2015b), but does this conclusion also hold for both EMAC SD simulations? Does the vertical resolution also matters here?

Thank you for highlighting that this sentence was unclear. The higher vertical resolution version of EMAC (L90) simulates weaker directly estimated TUMF than its L47 counterpart in both the C1 and SD experiments. We have edited the text to clarify this :

*"...seen in both REF-C1 and REF-C1SD experiments..."*

L.427-436: The differences in the annual cycle of wbar_star at 70 hPa between SD and RA should be added to the Figure 5 and discussed here.

Thank you for your suggestion. Similar to your comment about calculating a "multi-reanalysis mean" for the previously Supplemental Figure 2, we think it is important to highlight the differences between each SD model and the respective reanalysis instead (now added as Supplemental Figure S3). The following plot illustrates that the SD models simulate a rather different annual cycle compared to the RA they were nudged towards.

REFC1SD vs RA 1980-2009

We have also expanded the relevant part of the manuscript based on this result :

*"However, it should be noted that the annual cycle in lower stratospheric w\* in the REF-C1SD runs generally differs from that in the respective RA they were nudged towards (Supplemental Figure S3). In fact, the REF-C1SD – RA comparison highlights a wide variety in both the magnitude and the spatial patterns of their absolute differences, exhibiting on average larger differences than the REF-C1SD – REF-C1. Within the tropical upwelling region similarly to the differences between REF-C1 and REF-C1SD (the narrow band between the equator and 10°N) the local minimum associated with the weaker upwelling simulated by REF-C1SD is extended in most cases throughout the year when compared with the RA (Supplemental Figure S3)."*

L.431-432: The author states that NH midlatitude downwelling during summer is stronger for the SD simulations. To my view, there are only blueish colours north of the TA-latitudes during JJA in Figure 5c indicating weaker and not stronger downwelling.

Thank you for pointing this out, we have now corrected this.

L.451-452: "The REF-C1SD shows…" I assume that this sentence concerns the comparison between SD and RA, however it might be better to clarify this.

Thank you for pointing out this unclear sentence. It was referring to the differences between C1 and SD. As some minor details of the results have changed due to the fact that we are now comparing the models that performed both sets of experiments we have altered this whole paragraph to better communicate the key points.

L.454-456: Why is the annual cycle of wbar_star in the tropical lower stratosphere more consistent for the SD compared to the FR simulations? Above you conclude that the annual cycle and the phasing of SD simulations are weakly constrained and the intermodal spread is 20% larger than for FR simulations. How does this fit together?

We thank you for highlighting this statement which was poorly formulated and did not adequately communicate the interhemispheric and seasonal aspects of this comparison. Based on Figures 7a/b we conclude that the nudging constrains the tropical mean w* in boreal summer (JJA), which exhibits ~20% less spread than the REF-C1 experiments, but it does not constrain the tropical mean upwelling in boreal winter (DJF), which shows a factor of two larger spread than the free-running models. With respect to the turnaround latitudes, the REFC1-SD experiments exhibit less spread (by up to ~3° or 25%) than the REF-C1 experiments in both hemispheres during boreal winter (DJF) and in the SH during austral winter (JJA). In contrast, in the NH during austral winter the turnaround latitude in the nudged simulations shows an increased intermodel spread by 5.5° (25%) compared to the REF-C1 runs. Hence, the picture emerging is rather more complex than was written in the initial manuscript. We have edited/expanded the paragraphs for figures 7a/b and 7c/d in the manuscript and the associated changes with respect to your comment are (nb: black italic below denote unchanged parts of the manuscript) :

*"Based on Figures 7a/b we conclude that the nudging constrains the tropical mean w\* in boreal summer (JJA), which exhibits ~20% less spread than the REF-C1 experiments, but it does not constrain the tropical mean upwelling in boreal winter (DJF), which shows a factor of two larger spread than the free running models. The differences between the REF-C1SD runs and the respective RA they are nudged towards are generally larger in boreal winter than in boreal summer for the majority of the REF-C1SD models. Figures 7c and 7d show the climatological annual cycle in the turnaround latitudes at 70 hPa for the REF-C1 and REF-C1SD runs, respectively. For both REF-C1 and REF-C1SD runs the NH TA latitude varies significantly more than the SH TA latitude during boreal summer (JJA). The NH TA latitude is displaced more poleward for the REF-C1SD runs during boreal summer (JJA) for most models, corroborating the results of section 3.1 associated with an asymmetrical and wider upwelling region (Figure 2)*

*in the lower stratosphere. The REF-C1SD runs exhibit less spread (by up to ~3° or 25%) than the REF-C1 experiments in both hemispheres during boreal winter (DJF) and in the SH during austral winter (JJA). In contrast, in the NH during austral winter the turnaround latitude in the nudged simulations shows an increased intermodel spread by 5.5° (25%) compared to the REF-C1 runs. Nevertheless, the amplitude of the annual cycle for the RA is in fact quite amplified when compared with the REF-C1SD (just for the SH) while the inter-reanalysis spread is significantly higher (more than 2.5° in the SH) than both REF-C1 and REF-C1SD runs. To summarize, there is substantial inter-model spread in the turnaround latitudes and the amplitude of the annual cycle highlighting significant interhemispheric differences in the upwelling region between both sets of simulations as well between the nudged experiment and the RA. Hence, the picture emerging is rather more complex one. Arguably, the nudged runs are closer to the free-running simulations rather than the reanalysis they were nudged towards, hence not constraining the annual cycle of the residual vertical velocity in the lower stratosphere."*

L.456-459: Again, I am missing in the summary here the SD vs. RA comparison relevant for analysing the nudging effect on stratospheric residual circulation in CCMs.

Thank you for highlighting this. We believe we have sufficiently covered this point in the replies in the two previous comments.

L.503-569 (Section 3.5 and 3.6): Would it be possible to add here also the MLR of TUMF at 70 hPa and the related trend analysis for the RA datasets? The latter might give a very interesting insight into the question, if the different linear trends of FR and SD simulations are mainly driven by the different stratospheric residual circulation of the RA datasets or if they are significantly influenced by the nudging of the CCMs.

Thank you for this suggestion. We have applied the MLR analysis and trend sensitivity analysis to the RA datasets and have included the results in the supplement with some discussion in the main text (Figures S4, S7, S8).

The analysis reveals some interesting behaviours. In terms of the MLR analysis, ERA-I shows the highest $R^2$ value (0.66) while in MERRA the variance explained by the MLR is substantially lower (0.3) than the other RA datasets and the SD models (Figure S4). In terms of long-term trends, none of the SD models nudged towards ERA-I simulate a negative trend in 70 hPa TUMF over 1980-2009 despite the ERA-I dataset showing a negative trend (Figure S8). The trend sensitivity analysis for the RA datasets highlights a common feature of a negative trend in TUMF from the mid-1990s onwards. This feature is not found in any of the SD models (Figure 13), but is captured by some free-running simulations (Figure 12).

The relevant parts of the manuscript have been expanded to discuss the MLR and trend results for the RA datasets:

*"The MLR analysis was also applied to the RA TUMF at 70 hPa (Supplemental Figures S4 and S7). This highlights significant discrepancies in attributing the variance in TUMF in the different RA datasets to the various proxies used in the MLR model. Both the volcanic activity and ENSO contributions to the variance in the TUMF is rather suppressed when compared to the REF-C1SD runs. The negative linear trend in ERA-I is in strong contrast to the positive trends found in the other RAs and the REF-C1SD models. The negative trend in ERA-I found in the TUMF in the lower stratosphere over 1980-2009 corroborates the findings of Abalos et al. (2015) who showed trends based on the residual circulation w\* estimate in ERA-I were also negative over 1979-2012 throughout the depth of the stratosphere. Despite this difference in the representation of long-term changes, ERA-I shows the highest percentage of TUMF variance explained (66%), with MERRA showing a substantially lower $R^2$ (0.3) compared to the other reanalyses and the REFC1-SD models. The residuals are generally less correlated between the reanalyses on interannual timescales than was found in the REFC1-SD simulations, but are similar on inter-decadal timescales; however, the residuals in the RAs show a different temporal behaviour from those found in the REF-C1SD simulations (Figure 11) (note that the y-axis scale for the residuals in Supplemental Figure S4 is double that for the CCMI models in Figures 10 and 11). In summary, although nudging constrains the interannual variability in the TUMF at 70 hPa, the REF-C1SD runs generally do not resemble the RA they were nudged towards."*

And

*"Interestingly, the RA trend sensitivity analysis highlights that nudging does not constrain the underlying trends of the REF-C1SD models in the TUMF at 70 hPa, as the RA datasets exhibit a wide range of trends (Supplemental Figure S8) when compared to the REF-C1SD runs. None of the REFC1-SD models simulate the statistically non-significant decrease in 70 hPa TUMF starting around the mid-1990s seen in the RA datasets."*

L585-588: What is the explanation of the differences among the directly simulated TUMF for SD simulations that can be derived from the diagnosed wave forcing using the downward control principle? (See also my comments above: L.401-404)

Please see our reply to the previous comment above. We have added the following sentence in this section of the manuscript:

*"The large intermodel spread in the differences between the direct and downward control estimates of lower stratospheric TUMF in the REFC1-SD simulations shows that multiple factors*

*are likely to affect the decoupling of the wave forcing and residual circulation; this includes nudging timescales, nudging parameters, nudging height range, and model resolution. Further nudging sensitivity studies to examine the effects of these factors within one model would help to offer a more detailed explanation of these differences."*

L.601-602: What is the explanation that ERA-I shows no positive trend in tropical upwelling but a significant positive linear trend in TUMF at 70 hPa?

This sentence was not phrased clearly in the original manuscript. We intended to communicate that ERA-Interim does not show a positive trend in both TUMF and tropical upwelling (see Figure S7 in the revised manuscript for TUMF and also Figure 11 of Abalos et al., (2015); for trends in tropical average upwelling). But the models that were nudged towards ERA-Interim do show a positive trend in TUMF, so there is an inconsistency between what the RA dataset and the nudged models show in terms of long-term trends.

We have rephrased this bullet to:

"6. Most nudged simulations show a statistically significant positive trend in tropical upward mass flux (TUMF) in the lower stratosphere over 1980-2009, which is on average larger than the trends simulated in the free-running models. This is despite the fact that five out of the seven models analysed were nudged towards ERA-Interim, which shows a negative trend in TUMF (also Abalos et al., 2015), while JRA-55 and MERRA show a positive trend."

Thank you for this suggestion. We agree that some further discussion of the chosen measures of residual circulation is important to place the results into a broader context. As indicated by the reviewer, the choice of diagnostics is likely to be a particular issue for interpreting the nudged

models and reanalyses due to the additional tendencies imposed in the model equations. Part of our choice is pragmatic, based on the availability of output from the CCMI model dataset, which provides direct estimates of residual circulation diagnostics following Andrews et al. (1987) and resolved and parameterized wave forcing components. As TUMF is an integrated measure of w* between TA latitudes, the results regarding interannual variability and long-term trends are similar to if an average w* measure is used. Arguably, the TUMF is a first-order measure in order to evaluate whether the mass circulation is affected by nudging and it has been widely used in previous multi-model comparison and reanalysis studies and so we include it here (e.g. Butchart et al., 2006, Butchart et al., 2010, Butchart et al., 2014 and references therein; Rosenlof, 1995; Seviour et al., 2012; SPARC CCMVal, 2010). The important point is that we use consistent diagnostics in the free running and nudged experiments and the reanalyses, so that we are always comparing the same measures across the different simulations. Unfortunately, as heating rates are not provided by all models in the CCMI output, it would not be possible to compute a consistent (i.e. with the appropriate model radiation scheme) estimate of the residual circulation using the thermodynamic equation. It is possible that for the nudged simulations calculation of the residual circulation based on the thermodynamic equation might result in different behaviour to the direct TEM formulation used here. We now suggest this as an interesting topic for future research in the conclusions. The downward control principle (DCP; Haynes et al., 1991) calculations provide a framework to assess the extent to which the wave forcing and directly estimated residual circulation become decoupled as a consequence of the additional tendencies imposed by the nudging.

Based on the above we have expanded/edited the relevant parts of the text to the following :

End of section 2.2.1, (just before the introduction of DCP) :

*"The TUMF has been used widely as a measure of the strength of the BDC (Rosenlof, 1995; Butchart et al., 2006, Butchart et al., 2010, Butchart et al., 2014 and references therein; Seviour et al., 2012), so it's use here enables a direct comparison with earlier studies. Arguably, the strength of the TUMF is a first-order metric to evaluate changes to the mass circulation as a consequence of nudging. As mentioned above, by calculating the annual means of TUMF accounting for the seasonal cycle of the turnaround latitudes we capture the correct evolution of the intraseasonal (not shown) and interannual variability in the TUMF."*

End of section 2.2.2, (after introduction of DCP) :

*"Both the direct and DCP methods rely on the applicability of the quasi-geostrophic approximation to interpret the results. We note that in addition to the direct and DCP approaches used here, the residual circulation can also be estimated using the thermodynamic*

*equation. Unfortunately, heating rates were not available from all CCMI model simulations to perform this calculation. Studies have shown that the estimates from the different methods for evaluating the residual circulation can differ (Abalos et al., 2015; Linz et al., 2019; Seviour et al., 2012), particularly in reanalyses where the normal conservation laws are not required to be met. Similar issues are likely to beset the nudged model simulations owing to the additional tendencies included in the model equations. The differences between the calculation methods for the residual circulation can be as large as, or larger than, the differences between reanalysis datasets for the same diagnostic (Abalos et al., 2015; Linz et al., 2019), and may further depend on choices around averaging between fixed latitudes or the turnaround latitudes (Linz et al., 2019), so it is important to bear this in mind in interpretation of the results presented here. Nevertheless, we compute the diagnostics for the residual circulation in a self-consistent manner in the models and reanalyses to enable comparison with earlier multi-model studies (e.g. Butchart et al., 2006, Butchart et al., 2010, SPARC, 2010)."*

2) Although the method used here to do the multiple linear regression is somewhat standard, it does not actually remove all of the variability that is related to these signals. In particular, the relative phasing of the QBO and the annual cycle will cause an enhanced wintertime upwelling in some years, so the result that the leftover variability is highly correlated could be related to this. Randel et al. 1999 have a nice treatment. I would consider using the method in that paper (described on p. 458, see also Randel and Wu 1996) to determine if the relative timing of the annual cycle and QBO is the cause of the covariation of the nudged time series. This phenomenon is also mentioned in the cited Baldwin et al. 2001 QBO review.

Thank you for raising this interesting point. The MLR model is applied to the annual mean TUMF timeseries and hence the QBO terms are resolved at an annual timescale. Therefore, the method presented by Randel et al. (1999) to account for interannual variations in the relative phasing of the QBO with the annual cycle is not readily applicable in our analysis. We further note that all of the models except CMAM and MRI-ESMr1 also nudge the phase of the QBO in the REF-C1 simulations (Morgenstern et al., 2017). Hence, if this is a main factor for explaining the MLR residuals, we expect that such an effect would also be seen in the REF-C1 simulations. However, the temporal correlation of the residuals is much weaker in the REF-C1 experiments for those models that nudge the QBO, suggesting this is not a major factor for explaining the correlated residuals in the REFC1-SD experiments.

3) The authors find that the reanalyses differ from the nudged runs which in turn differ from the free running models. I would like to see some presentation of the difference between the mean states of the reanalyses and the mean states of the models without nudging. The result that the models tend to have stronger upwelling in their SD runs could be caused by a systematic low bias in the model temperature in the tropics compared to the reanalysis temperature.The repeated

nudging with a primarily positive value would then cause an apparent increase in upwelling, without changing the underlying model physics that are the cause of any systematic mean state bias. To understand why the nudging does what it does, I think the difference between the free running models and the reanalyses needs to be explored in much more detail–perhaps for just one model. A continuous spurious forcing because of a mismatch in the mean state (and likely also in the seasonal cycle) should impact the mean while leaving the interannual variability alone (line 612).

Thank you for this suggestion. We present below the percent differences in zonal mean temperature in the stratosphere between the REF-C1 simulations and the respective reanalysis they are nudged towards (stippling denotes statistical significance at 95% based on a two-tailed Student's t-test); the equivalent differences for the REFC1-SD simulations are also shown in the right panels (note the reduced colour scale).

Most models show a cold bias relative to the reanalyses in the tropical tropopause layer, except for MRI-ESMr1 which shows a warm bias. The two versions of EMAC show the largest relative cold bias in the TTL. Much of this bias is alleviated in the nudged simulations, as expected, but EMAC (l47/90), CESM1-WACCM and SOCOL3 still show a (smaller) cold bias in the TTL. Larger differences remain above the level where nudging ceased to be imposed (e.g. 10 hPa in EMAC and 40 hPa in MRI-ESM1r1). The opposite sign of the TTL temperature bias in the free-running MRI-ESMr1 model provides a useful test bed for the point raised by the reviewer. The REFC1-SD MRI-ESMr1 simulation still shows stronger upwelling near the equator compared to the free-running model (Figure 3c) despite having a warm bias; this result is similar to the effect of nudging in the other models that show a cold TTL bias. Hence from this cursory investigation, we do not identify a systematic relationship between the intrinsic model bias and the effect of the nudging on the residual circulation. To get a better handle on this, a detailed analysis of the sensitivities within one model to relaxation timescales and nudging parameters such as the altitude range etc. would be needed in order to robustly establish how nudging affects the mean residual circulation.

[Figure]

**Minor comments**

I find the second paragraph of the introduction to be lacking. Although Abalos et al. (2015) did conclude that there was likely a 2-5%/decade trend in the circulation, they found substantial differences with different calculations for the circulation strength. Ploeger et al. (2019) have a new paper out on the inconsistency of the BDC trends for the different reanalyses, looking at the age spectra. Age of Air is mentioned somewhat abruptly and with no explanation. The results are also misrepresented. Engel et al. 2009 did not find a statistically significant increase in the mean age of air in the midlatitudes. They found no statistically significant change. There are limitations both on the availability of measurements and on the interpretation of age of air as a direct proxy for the residual circulation, and much more recent papers discuss this, rather than Waugh and Hall 2002. There are new satellite based measurements–ACE-FTS (e.g. Ray et al. 2016), MIPAS (Haenel et al. 2015), and MLS (Linz et al. 2017)–and new measurements from Air Core (Engel et al. 2017). The interpretation of the data is also difficult because of the separation of the residual circulation from the mixing (e.g. Garny et al. 2014, Ploeger et al 2015, as mentioned in the first paragraph–perhaps just move this comment to this paragraph?). The only attempt to calculate the residual circulation from tracer data did not calculate the residual circulation directly, but instead the diabatic circulation (Linz et al. 2017), and they did not calculate trends.

Thank you for this comment. We have expanded this part of the introduction to give a more balanced and detailed synopsis of the literature:

*"Past studies have shown substantial spread across models in the mean strength of the residual circulation (e.g. Butchart et al., 2010). Nevertheless, climate and chemistry-climate models (CCMs) consistently simulate a long-term strengthening of the residual circulation with an increase of ~2% decade$^{-1}$ (e.g. Butchart et al., 2010; Hardiman et al., 2014), though there are differences across models in the relative contribution to trends from resolved and parameterized wave forcing. Reanalysis datasets also suggest a strengthening of the residual circulation over the past several decades of the order 2-5% decade$^{-1}$ (Abalos et al., 2015; Miyazaki et al., 2016) apart from one (ERA-Interim) which shows a weakening of the deep branch of the BDC (Seviour et al., 2012; Abalos et al., 2015). However, reanalyses are subject to multiple caveats, particularly in their suitability for trend studies, and there are substantial differences in residual circulation trends calculated from the same reanalysis using different methods (Abalos et al., 2015). Given the limitations of reanalyses, evaluating the fidelity of model estimates of residual circulation variability and trends is challenging since there are no direct measurements of the residual circulation.*

*The only direct estimates of the stratospheric mass circulation come from tracer measurements, which can be used to calculate stratospheric age-of-air (AoA) (Kida, 1983; Schmidt and Khedim 1991; Waugh and Hall, 2002). AoA represents the combined effects of advection by the residual circulation and turbulent mixing processes, and as such cannot be directly related to the residual circulation. While progress has been made in separating the relative effects of these two factors for AoA calculated from models (Garny et al., 2014; Dietmuller et al., 2018, Eichinger et al., 2019, Šácha et al., 2019), using Lagrangian models driven by reanalysis fields (Ploeger et al., 2015a, Ploeger et al., 2015b, Ploeger and Birner, 2016, Ploeger et al., 2019) and comparing these effects in both CCMs and Lagrangian models (Dietmuller et al., 2017), this is more difficult to achieve in observations. Engel et al. (2009) and Stiller et al. (2012) used balloon-borne measurements of stratospheric trace gases and found a statistically non-significant increase in AoA in the middle stratosphere at northern mid-latitudes, which has been corroborated by more recent study using balloon-borne measurements at two midlatitude sites in the Northern Hemisphere (Engel et al., 2017). It has been hypothesised based on analyses of more recent satellite datasets, which have greater spatial and temporal coverage, that the subtropical AoA trends can be explained by a weakening of the mixing barriers at the edge of the tropical pipe (Neu and Plumb, 1999) that is masking the effects of an increase in tropical upwelling on AoA (Stiller et al., 2012; Haenel et al., 2015). In contrast with the AoA trends derived from observations, CCMs forced with observed sea-surface temperatures (SSTs), greenhouse gases and ozone-depleting substances show a decrease in AoA throughout the stratosphere (World Meteorological Organization, 2018; Li et al., 2018; Morgenstern et al.,*

*2018; Abalos et al., 2019; Polvani et al., 2019). Theoretical approaches based on the tropical leaky pipe model (Neu and Plumb, 1999) have shown promise for bridging the information on the stratospheric circulation derived from observations with outputs from GCMs/CCMs (Ray et al., 2016).*

*As explained above, it is difficult to infer information on the residual circulation from AoA without carefully accounting for the effects of mixing. More recent theoretical developments offer a means of calculating the diabatic circulation using stratospheric tracers (Linz et al., 2017), which is a promising avenue as this is more closely related to the residual circulation than AoA. The results of Linz et al. (2017) showed consistent estimates of the diabatic circulation in the lower stratosphere, but large uncertainties of up to a factor of two in the mean strength in the upper stratosphere. Targeted measurement strategies to better characterise long-term changes in the observed stratospheric meridional circulation have been proposed (Moore et al., 2014; Ray et al., 2016)."*

A more detailed explanation of what nudging is, how it is implemented, why it is used, and the motivation for this work would help the introduction. The current paragraph in the introduction (line 85) is good, and I think that including more specific examples would be very helpful. Similarly, in the conclusion, commentary on what these nudged runs could be useful for would be appreciated in addition to the entirely appropriate cautionary words.

Thank you for this comment. We have expanded this paragraph to include your suggestions above. The relevant part of the manuscript is now (nb: black italic below denote unchanged parts of the manuscript) :

*"In an attempt to obtain a closer comparison with observed stratospheric trace species, some studies have used model simulations with meteorological fields nudged or relaxed towards a reanalysis dataset (Jeuken et al., 1996); these include many studies of ozone variability and trends (e.g. van Aalst et al., 2003; Solomon et al., 2016; Hardiman et al., 2017b; Ball et al., 2018),* comparisons between models and satellite-based multi-species observational records (Froidevaux et al. 2019) *as well as the chemical and climatic effects of volcanic eruptions (Löffler et al., 2016; Solomon et al., 2016; Schmidt et al., 2018).* Nudged simulations have also been used to study mechanisms for dynamical coupling between the stratosphere and troposphere (Hitchcock and Simpson, 2014). *Nudging involves adding additional tendencies to the model equations* by constraining the simulated meteorological fields to reanalysis fields (Kunz et al., 2011). *Nudged variables can include horizontal winds (or divergence and vorticity), temperature, surface pressure, latent and sensible heat fluxes. However, vertical winds, which are small residual from horizontal divergence, are not nudged and the underlying model physics can yield quite different results from the reanalysis that they are nudged towards (Telford et al., 2008; Hardiman et al., 2017). A recent study by Orbe et al. (2018) analyzed tropospheric tracers*

*in nudged CCM simulations and found large differences in the distributions of the tracers, which could be partly traced to differences in the model convection schemes. They urged users to adopt a cautious approach when interpreting tracers in nudged simulations given their dependence on not only large-scale flow but also sub-grid parameterizations. In contrast, critical evaluation of the stratospheric residual circulation in models relaxed to reanalysis fields has been rather limited to date."*

As for the conclusions the now expanded relevant part of the manuscript (second from last paragraph of the manuscript):

*"Despite the limitations described here, some success has been reported in studies that used nudged simulations to investigate specific meteorological events, such as Sudden Stratospheric Warmings, and in particular for exploring processes beyond the top of the nudging region in the Mesosphere-Lower Thermosphere (e.g., Tweedy et al., 2013; Chandran and Collins, 2014; Pedatella et al., 2014). In order to reduce discrepancies between nudged and free-running simulations, various nudging techniques have been investigated. The role of gravity waves in the error growth that the nudging introduces over time has been highlighted for a single model (Smith et al., 2017). Constraining just the horizontal winds without the temperature was found to be a good strategy when investigating the aerosol indirect effects without affecting significantly the mean state (Zhang et al., 2014)...*

*...A dedicated study of the sensitivities within one model to relaxation timescales, nudging parameters, nudging height range, and model resolution would be needed to offer a detailed explanation for these differences."*

Do any of the "free running" models have a nudged QBO, and if so, which ones?

All of the free-running models presented in the study apart from CMAM and MRI-ESMr1 impose a QBO "nudging" in their simulations as mentioned in the last paragraph of section 3.4 (Table S8 of supplement of Morgenstern et al., 2017).

Since the focus of this paper is really on the difference between SD and free running models, it might make sense to omit the models that do not include both. I respect the authors choice to have those comparisons in the supplementary information. However, I think I would find the plots more manageable without the additional lines.

Thank you for this recommendation. In response to this comment and also the comments of reviewer 1 we have removed the models which did not perform the SD simulation from the main body of the paper (GEOSCCM, NIWA-UKCA and ULAQ-CCM) and the main text now focuses

on the seven models for which both REF-C1 and REFC1-SD simulations are available. For completeness a subset of diagnostics are shown for those models in the supplementary material.

Line 106: aren't they necessarily inconsistent?

We have corrected this to: "*will add forcings that are inconsistent with the model state.*"

Lines 100-129: excellent discussion.

Thank you for the positive comment!

Line 153: Do you mean Table 2?

Thank you for spotting this, we have now corrected this.

Line 155 etc.: More details here would be good. I don't understand what is meant by T (with wave-0), and I assume that "temperature" is also T (for CMAM), but if that's true then they all nudge temperature so maybe mention that? Including the sentence "Nudging timescales range from 6 - 50 hours." (or some equivalent) would also emphasize how different these treatments are.

We have added some more details here as per your recommendation. In Table 2, temperature is referred to consistently as T. The wave-0 was intended to refer to the additional nudging of the global mean temperature in EMAC. We now state this explicitly. We have also changed L152-153 in the text to: "*The REF-C1SD simulations nudge temperature and other meteorological fields such as horizontal winds, vorticity and divergence and some surface fields (Table 2), while the chemical fields are left to evolve freely. The nudging timescales range from 6 - 50 hours and the height range over which nudging is applied varies (Table 2).*"

Line 159-160: Table 1?

Thank you for spotting this, we have now corrected this.

Line 168: "primary variable"–not sure what is meant by this

We meant prognostic. We have now corrected this.

Line 442: I have trouble seeing this from the plot. Could you provide the numbers? Is the difference in the peak-to-peak amplitude significant?

Yes, due to the congestion of the multiple lines is not that easy to see but the difference is not significant. This sentence now includes the numbers of the peak-to-peak amplitude

*"Comparing the MMM annual cycle of the REF-C1 runs with the MMM REF-C1SD of Figure 7b reveals that on average the REFC1-SD models show a slightly larger peak-to-peak amplitude (0.22 mm s$^{-1}$ vs. 0.19 mm s$^{-1}$) in the annual cycle."*

Line 476. . .: How do these results relate to Abalos et al. 2014, who found such a strong dependence of the upwelling on the location of the EPFD?

Thank you for this interesting point. We agree it would be interesting to examine how the patterns of EPFD vary between the free running and nudged simulations and how this relates to fluctuations in upwelling. This work is planned for a follow-up study.

Line 488-90: I found this sentence confusing.

Thank you for spotting this unclear sentence. We have removed this sentence as it was related to an earlier sentence on lines 483-485, which has been edited to:

*"This result indicates that although nudging does not constrain the mean residual circulation, it does constrain the interannual variability and produces very similar contributions to variability across models from both resolved and parameterized wave forcing.*

Line 498: remove "due to"

Done.

Section 3.6: Nice.

Thank you for the positive comment!

Line 606.: This paragraph is very long and contains a lot of important information. Turn into two or even three shorter paragraphs?

Thank you for this comment, we have now turned this paragraph into three shorter ones to improve the clarity.

Figures: Consider stippling the insignificant part of the plot. I read this on a number of different devices, and on some it looked fine, but on one of them the plots were impossible to see because the colors weren't visible behind the stippling. This wasn't an issue for the last two figures for some reason.

It may take a while for these plots to be loaded properly in a pdf viewer, as they are quite large files. We changed the subtleties of the stippling patterns as seen in the figures and hopefully they are more clear now.

It could be useful to have models that are nudged to the same reanalysis plotted in the same color family to aid interpretation of the complicated line plots.

We have now colour coded the model results nudged towards ERA-I in red(-ish) colours. We hope that without the additional lines of the extra models, the results are more clear.

[14,5] Climate Modelling and Analysis Section, Center for Global Environmental Research, National Institute for Environmental  Studies, Tsukuba, Japan
[15,6] Japan Agency for Marine-Earth Science and Technology (JAMSTEC), Yokohama, Japan

**Correspondence**: Andreas Chrysanthou (eeac@leeds.ac.uk)

**Abstract**

We perform the first multi-model intercomparison of the impact of nudged meteorology on the stratospheric residual circulation using hindcast simulations from the Chemistry-Climate Model Initiative (CCMI). We examine simulations over the period 1980-2009 from 7 models in which the meteorological fields are nudged towards a reanalysis dataset  compare these with their equivalent free-running simulations and the reanalyses themselves.  We show that for the current implementations, nudging meteorology does not constrain the mean strength of the stratospheric residual circulation and the inter-model spread is similar, or even larger, than in the free-running simulations. The nudged  models generally show slightly stronger upwelling in the tropical lower stratosphere compared to the free-running versions and exhibit marked differences compared to the directly estimated residual circulation  from the reanalyses dataset they are nudged towards. Downward control calculations applied to the nudged simulations reveal substantial differences between the climatological lower stratospheric tropical upward mass flux (TUMF) computed from the modelled wave forcing and that calculated directly from the residual circulation. This explicitly shows that nudging decouples the wave forcing and the residual circulation, so that the divergence of the angular momentum flux due to the mean motion is not balanced by eddy motions, as would typically be expected in the time mean. Overall, nudging meteorological fields leads to increased

inter-model spread for most of the measures of the mean climatological stratospheric residual circulation assessed in this study. In contrast, the nudged simulations show a high degree of consistency in the interannual variability of the TUMF in the lower stratosphere, which is primarily related to the contribution to variability from the resolved wave forcing. The more consistent interannual variability in TUMF in the nudged models also compares more closely with the variability found in the reanalyses, particularly in boreal winter. We apply a multiple linear regression (MLR) model to separate the drivers of interannual and long-term variations in the simulated TUMF; this  explains up to ~75% of the variance in TUMF in the nudged simulations.  The MLR model reveals a statistically significant positive trend in TUMF for most models  over the period 1980-2009. The TUMF trend magnitude is generally larger in the nudged models compared to their free-running counterparts, but the intermodel range of trends doubles from around a factor of 2 to a factor of 4 due to nudging. Furthermore, the nudged models generally do not match the TUMF trends in the reanalysis they are nudged toward for trends over different periods in the interval 1980-2009. Hence, we conclude that nudging does not strongly constrain chemistry-climate model (CCM) simulated long-term trends in the residual circulation.  Our findings show that while nudged simulations may, by construction, produce accurate temperatures and realistic representations of fast horizontal transport, this is not  typically the case for the slower zonal mean vertical transport in the stratosphere. Consequently, caution is required when using nudged simulations to interpret the behaviour of  stratospheric tracers that are  affected by the residual circulation.

**1 Introduction**

The Brewer-Dobson circulation (BDC) is characterized by upwelling of air in the tropics, poleward flow in the stratosphere, and downwelling at mid and high latitudes. The circulation can be separated into two branches: the shallow branch in the lower stratosphere and the deep branch in the middle and upper stratosphere (Plumb, 2002; Birner and Bönisch, 2011). The BDC affects the distribution of trace species in the stratosphere, such as ozone, and its strength partly determines the lifetimes of long-lived gases such as chlorofluorocarbons (CFCs; Butchart and Scaife, 2001). It also determines stratosphere to troposphere exchange of ozone (Hegglin and Shepherd, 2009), which is important for the tropospheric ozone budget (Wild, 2007). In the tropical lower stratosphere, where the photochemical lifetime of ozone is long, variations and trends in the strength of the BDC are the main drivers of ozone within the annual cycle (Weber et al., 2011), for interannual and longer term variability (Randel and Thompson, 2011) and in response to climate change (e.g. Keeble et al., 2017).

80     Here we focus on the advective part of the BDC, or the residual circulation, which is driven by wave breaking in the stratosphere from planetary scale Rossby waves and gravity waves (Holton et al., 1995). It is important to note that the overall tracer transport in the stratosphere is also affected by turbulent eddy mixing, which has been evaluated separately in previous studies (Garny et al., 2014; Ploeger et al., 2015a, 2015b; Dietmüller et al., 2018;

85    Eichinger et al., 2019; Šácha et al., 2019). The residual circulation is commonly evaluated in model (Butchart et al., 2010, 2011) and reanalysis (Abalos et al., 2015; Kobayashi and Iwasaki, 2016) studies using the Transformed Eulerian Mean circulation (TEM; Andrews and McIntyre, 1976, 1978; Andrews et al., 1987).

Past studies have shown substantial spread across chemistry-climate models (CCMs) in the mean

90    strength of the residual circulation (e.g. Butchart et al., 2010). Nevertheless, CCMs consistently simulate a long-term strengthening of the residual circulation with an increase of ~2% decade$^{-1}$ (e.g. Butchart et al., 2010; Hardiman et al., 2014), though there are differences across models in the relative contribution to trends from resolved and parameterized wave forcings.  Reanalysisreanalysis datasets also suggest a strengthening

95    of the residual circulation over the past  few decades of the order 2-5% decade$^{-1}$ (Abalos et al., 2015; Miyazaki et al., 2016) apart from one (ERA-Interim) which shows a weakening of the deep branch of the BDC (Seviour et al., 2012; Abalos et al., 2015). However, reanalyses are subject to multiple caveats, particularly in their suitability for trend studies, and there can be substantial differences in residual circulation trends calculated from the same reanalysis using different methods

100   (Abalos et al., 2015).

Given the limitations of reanalyses, evaluating the fidelity of model estimates of residual circulation variability and trends is challenging since there are no direct measurements of the residual circulation. The only direct estimates of the stratospheric mass circulation come from tracer measurements, which can be used to calculate stratospheric age-of-air (AoA) (Kida, 1983; Schmidt and Khedim, 1991;

105   Waugh and Hall, 2002). AoA represents the combined effects of advection and mixing processes, and as such cannot be directly related to the residual circulation. While progress has been made in separating the relative effects of advection and mixing for AoA calculated from models (Garny et al., 2014; Dietmüller et al., 2018; Eichinger et al., 2019; Šácha et al., 2019), from Lagrangian models driven by reanalysis data (Ploeger et al., 2015a, 2015b, 2019; Ploeger and Birner, 2016), and comparing the

110   effects in both CCMs and Lagrangian models (Dietmüller et al., 2017), this is more difficult to achieve in observations.  Engel et al. (2009)  used balloon-borne measurements of stratospheric

115   trace gases and found  a statistically non--significant increase in AoA in the middle stratosphere at

northern midlatitudes; this has been corroborated in a more recent study using longer   It has been hypothesized based on analyses of recent satellite tracer datasets, which have greater spatial and temporal coverage, that subtropical AoA trends can be explained by a weakening of the mixing barriers at the edge of the tropical pipe (Neu and Plumb, 1999) that is masking the effects of an increase in tropical upwelling on AoA (Stiller et al., 2012; Haenel et al., 2015). In contrast with AoA trends derived from observations, CCMs forced with observed sea-surface temperatures (SSTs), greenhouse gases and ozone-depleting substances show a decrease in AoA throughout the stratosphere (Karpechko and Maycock, 2018; Li et al., 2018; Morgenstern et al., 2018; Abalos et al., 2019; Polvani et al., 2019). Theoretical approaches based on the tropical leaky pipe model (Neu and Plumb, 1999) have shown promise for bridging the information on the stratospheric circulation derived from observations with outputs from general circulation models (GCMs)/CCMs (Ray et al., 2016), but differences remain (Karpechko and Maycock, 2018).

More recent theoretical developments offer a means of calculating the diabatic circulation using stratospheric tracers (Linz et al., 2017), which is a promising avenue as this is more closely related to the residual circulation than AoA. Linz et al. (2017) showed consistent estimates of the diabatic circulation in the lower stratosphere based on two independent satellite tracer datasets but identified large uncertainties of up to a factor of two in the mean circulation strength in the upper stratosphere. Hence, the available tracer datasets are not yet suitable for characterizing trends in the diabatic circulation using these methods. Targeted measurement strategies to better characterize long-term changes in the stratospheric meridional circulation have been proposed (Moore et al., 2014; Ray et al., 2016).

In an attempt to obtain a closer comparison with observed stratospheric trace species, some studies have used model simulations with meteorological fields nudged or relaxed towards analysis or reanalysis datasets (Jeuken et al., 1996). These include  studies of stratospheric ozone variability and trends (e.g. van Aalst et al., 2004; Solomon et al., 2016; Hardiman et al., 2017b; Ball et al., 2018), comparisons between models and satellite-based multi-species observational records (Froidevaux et al., 2019) and in particular, focusing on specific meteorological events such as the Sudden Stratospheric Warming in the 2009/2010 winter (Akiyoshi et al., 2016),  as well as the chemical and climatic effects of volcanic eruptions (Löffler et al., 2016; Solomon et al., 2016; Schmidt et al., 2018). Nudged simulations have also been used to study mechanisms for dynamical coupling between the stratosphere and troposphere (Hitchcock and Simpson, 2014) and to examine the effects of different regions on atmospheric predictability (e.g. Douville, 2009; Jung et al., 2010). Nudging involves adding additional

[revised manuscript text omitted]

To examine the effect of nudging on the stratospheric residual circulation this study compares hindcast simulations from free-running and nudged versions of the same models that participated in the phase 1 of the Chemistry-Climate Model Initiative (CCMI; Morgenstern et al., 2017). Nudged experiments were not performed in previous chemistry-climate multi-model comparisons (Chemistry–Climate Model Validation Activity 2; CCMVal-2), so CCMI offers a timely opportunity to evaluate the effect of nudging on simulated mean biases, variability and long-term trends in the residual circulation. For completeness, we also present a comparison between the nudged simulations and the reanalysis datasets the models are nudged towards. The manuscript is laid out as follows: Section 2 describes the CCMI and the reanalysis data used in the present study along with the diagnostics for the residual circulation, section 3 presents results covering the mean circulation, annual cycle, interannual variability and trends, and section 4 summarizes the results and discusses the implications for using nudged simulations to study aspects of the observational record.

**2 Data and Methods**

**2.1 Models and experiments**

CCMI is the successor activity to CCMVal-2 and the Atmospheric Chemistry and Climate Model Intercomparison Project (ACCMIP; Lamarque et al., 2013). We use the hindcast free-running simulations, REF-C1 , and the nudged specified dynamics  simulations, REF-C1SD, which cover the periods 1960-2009 and 1980-2009, respectively. Here we analyze the common 30-year period 1980-2009 that was run by all models for both experiments  with prescribed observed SSTs and sea ice concentrations. The CCMI data were downloaded from the British Atmospheric Data Centre (Hegglin and Lamarque, 2015). For an extensive overview of the CCMI models see Morgenstern et al. (2017). We analyze those CCMI models that output the necessary TEM diagnostics. At a minimum, this requires the residual vertical velocity ($\overline{w}^*$) and the residual meridional velocity ($\overline{v}^*$) (Andrews et al., 1987); where available we also use the resolved and

parametrized wave forcing fields from the models. This gives results from a total of 10 models which differ from one another in various aspects such as their horizontal resolution,  ranges from 1.9° to 5.6°, their vertical resolution as well as their sub-grid parameterizations (see Table 1). The main text concentrates on the 7 out of 10 models that performed both the REF-C1 and REF-C1SD experiments (Table 2). However, the broad conclusions drawn in the main text for the characteristics of the 7-member REF-C1 ensemble are consistent with the behavior for all 10 models. Hence the 3 models that only performed the REF-C1 experiment (GEOSCCM, NIWA-UKCA and ULAQ-CCM) are not discussed further, but for completeness a subset of diagnostics from those models is shown in the supplement (Supplementary Figures S1-S5).

For the REF-C1 simulations we analyze between 1-5 ensemble members (depending on what was available) and for REF-C1SD the one realization submitted from each model. The REF-C1SD simulations nudge temperature and other meteorological fields such as horizontal winds, vorticity and divergence and some surface fields (Table 2), while the chemical fields are left to evolve freely. The nudging timescales range from 6 - 50 hours and the height range over which nudging is applied varies (Table 2). The TEM and related diagnostics that were available from each model are shown in Table 3. The models use different reanalysis fields for nudging taken from ERA-Interim (Dee et al., 2011), JRA-55 (Ebita et al., 2011; Kobayashi et al., 2015) or MERRA (Rienecker et al., 2011). The differences in the residual circulation diagnosed from  reanalyses have been identified and documented in previous studies (e.g. Abalos et al., 2015).

**2.2. Model diagnostics**

**2.2.1 TEM residual circulation**

The TEM velocities $(\bar{v}^*, \bar{w}^*)$ are defined as (Andrews et al., 1987):

$$\bar{v}^* = -\frac{1}{\rho_0 \cdot a \cdot cos\phi} \frac{\partial \overline{\Psi}^*}{\partial z}, \quad \bar{w}^* = \frac{1}{\rho_0 \cdot a \cdot cos\phi} \frac{\partial \overline{\Psi}^*}{\partial \phi}, \qquad (1)$$

where $\overline{\Psi}^*(\varphi,z)$ is the residual meridional mass streamfunction, $\rho_0$ is  log-pressure density, $a$ is  Earth's radius and $\phi$ is  latitude. As most of the models analyzed here use a hybrid-pressure vertical coordinate, the  prognostic variable is the pressure vertical velocity, $\bar{\omega}^*$ calculated in $Pa\ s^{-1}$, which must be converted to $m\ s^{-1}$ in order to get the residual vertical velocity, $\bar{w}^*$. The  conversion of $\omega$ to $w$ in isobaric coordinates is given by the following equation:

$$\omega = \frac{dp}{dt} = \frac{\partial z}{\partial t}\frac{\partial p}{\partial z} = w\frac{-pg}{RT} = w\frac{-p}{H}, \qquad (2)$$

where $p$ is pressure, $R = 287\ J\ K^{-1}\ kg^{-1}$ is the gas constant for dry air and $H$ is a fixed scale height. Both
255 TEM velocity components were submitted as monthly mean fields to the CCMI data archive . Upon close examination of the CCMI model output, some discrepancies were found in the way that the residual vertical velocity was calculated among the models. Although a fixed scale height of $H = 6950\ m$ was recommended in the CCMI data request (Eyring et al., 2013; Hegglin and Lamarque, 2015), the TEM  output from some models (EMAC and SOCOL3) was
260 calculated incorrectly using a temperature-dependent density, $\rho_0 = p/RT$, instead of the log-pressure definition of the density $\rho_0 = \rho_s \cdot e^{-z/H}$, such that $z$ has a unique 1:1 correspondence with $p$. This methodological error leads to artificial spread in the model $\bar{w}^*$ fields . We note that previous multi-model comparisons of the residual circulation that use $\bar{w}^*$ taken directly from models may have been subject to the same issue (e.g. Butchart et al., 2010; SPARC, 2010), though we
265 cannot confirm this . To avoid this methodological inconsistency, Dietmüller et al. (2018) recalculated $\bar{w}^*$ from $\bar{v}^*$ using the continuity equation, which requires a vertical integration and a derivative along the meridional direction. The recalculation of $\bar{w}^*$ from $\bar{v}^*$ was also  explored for this study, but it was found to introduce additional errors affecting the latitudinal structure of $\bar{w}^*$ (not shown), specifically because of the reduced
270 number of CCMI requested pressure levels compared to the native model levels. We were able to overcome the discrepancy in the submitted $\bar{w}^*$ fields for the EMAC simulations by reconverting high frequency $\bar{\omega}^*$ output to $\bar{w}^*$ using the log-pressure density as in equation 2. However, for SOCOL3 the required output for thiswas not available and hence we use the submitted $\bar{w}^*$ for which the absolute values  should be
275 treated with caution. For the other  models , the results presented in this study are based on the original diagnostics submitted to the CCMI data archive, which we have verified were calculated in the correct way.

We  compute the mass flux across a given pressure surface as (Rosenlof, 1995):

$$2\pi \int_{\phi}^{pole} \rho_0\, a^2 cos\phi\, \bar{w}^*\, d\phi = 2\pi a \Psi^*(\phi), \quad (3)$$

280 using the boundary condition that $\Psi = 0$ at the poles. By finding at each pressure level the latitude at which $\Psi_{max}$ and $\Psi_{min}$ occur, which correspond to the height-dependent turnaround (TA) latitudes, we can calculate the net downward mass flux in each hemisphere. The net tropical upward mass flux, equal to the sum of the downward mass fluxes in each hemisphere, can then be expressed as (Rosenlof, 1995):

285

$$\textit{Tropical Upward Mass Flux (TUMF)} = 2\pi a\, (\Psi^*_{max} - \Psi^*_{min}) \quad (4)$$

The TUMF has been used widely as a measure of the strength of the BDC (e.g. Rosenlof, 1995; Butchart and Scaife, 2001; Butchart et al., 2006, 2010, 2011; Butchart, 2014 and references therein; Seviour et al., 2012), so its use here enables a direct comparison with earlier studies. Arguably, the strength of the TUMF is a first-order metric for evaluating changes in the stratospheric mass circulation as a consequence of nudging. As mentioned above, by calculating the annual means of TUMF accounting for the seasonal cycle of the TA latitudes we capture the correct evolution of the intraseasonal (not shown) and interannual variability in the TUMF.

**2.2.2 Downward control principle calculations**

Under steady-state conditions, the $\overline{\Psi}^*(\varphi,z)$ at a specified latitude $\varphi$ and at a log-(pressure )-height $z$ is given by the vertically integrated eddy-induced total zonal forces, $\overline{F}$, above that level (Haynes et al., 1991):

$$\Psi^*(\varphi, z) = \int_z^\infty \left\{ \frac{\rho_0 a^2 \overline{F} cos^2\phi}{\overline{m}_\phi} \right\}_{\phi=\phi(z')} dz , \quad (5)$$

where in the quasi-geostrophic limit $\overline{m}_\phi \approx -2\Omega a^2 sin\phi cos\phi$. The above integration occurs applies along lines of constant zonal-mean absolute angular momentum per unit mass, $\overline{m}=acos\phi(\overline{u}+a\Omega cos\phi)$ where $\overline{u}$ is the zonal mean zonal wind and $\Omega$ corresponds to is Earth's rotation rate, with boundary conditions of $\Psi \to 0$ and $\rho_0\overline{w}^* \to 0$ as $z \to \infty$. These lines of constant angular momentum are almost approximately vertical apart from except near the equator (up to $\sim \pm20°$) such that we can approximate calculate the solution of the above integral using constant $\varphi$ for the limits of the integral (Haynes et al., 1991). In climate model simulations, $\overline{F}$ corresponds to has contributions from resolved waves due to the divergence of Eliassen -Palm flux divergence (EPFD) and/or contributions from parameterized gravity wave drag due to sub-grid scale waves that originate from orography, convection and frontal instabilities. This enables us to estimate the contribution to the tropical upward mass flux of both resolved planetary wave driving (EPFD) along with and the orographic (OGWD) and non-orographic (NOGWD) parameterized gravity wave drag from the CCMI model output (Table 2) to the tropical upward mass flux and compare with the direct estimates derived from the residual vertical velocity $\overline{w}^*$.

Applying the downward control principle (DCP Haynes et al., 1991) can provide useful insights into the driving mechanisms of the stratospheric residual circulation and therefore explain part of the inter-model spread found in both REF-C1 and REF-C1SD simulations. While the downward control principle DCP enables the contributions of EPFD and OGWD/NOGWD to TUMF to be calculated under

various assumptions (Haynes et al., 1991), one has to keep in mind that the different wave forcings can interact and thus are not independent of each other (Cohen et al., 2013).

It is important to note some possible limitations of the diagnostic approaches chosen for this study. Both the direct and downward control principle methods rely on the applicability of quasi-geostrophic theory to interpret the results. In addition to the two approaches used here, the residual circulation can also be estimated using the thermodynamic equation. Studies have shown that the estimates from the different methods for evaluating the residual circulation can differ (Seviour et al., 2012; Abalos et al., 2015; Linz et al., 2019), particularly in reanalyses where standard global conservation laws (e.g. conservation of mass) are generally not required to be met. Similar issues are likely to beset the nudged model simulations owing to the additional tendencies included in the model equations. The differences between the calculation methods for the residual circulation can be as large as, or larger than, the differences between reanalysis datasets for the same diagnostic (Abalos et al., 2015; Linz et al., 2019), and may further depend on choices around averaging between fixed latitudes or the TA latitudes (Linz et al., 2019), so it is important to bear this in mind in interpretation of the results presented here. Unfortunately, heating rates were not available from all CCMI model simulations to perform the thermodynamic equation calculation. Nevertheless, we compute the direct and downward control principle diagnostics for the residual circulation in a self-consistent manner in the models and reanalyses to enable comparison with earlier multi-model studies (Butchart et al., 2006, 2010; SPARC, 2010).

**2.3 Multiple linear regression model**

To investigate the drivers of interannual variability in the residual circulation we apply a multiple linear regression (MLR) model (equation 6) to the annual mean timeseries of TUMF. The model includes terms for known drivers of variations in tropical lower stratospheric upwelling: major volcanic eruptions (Pitari and Rizi, 1993), El Nino Southern Oscillation (ENSO) (García-Herrera et al., 2006; Marsh and Garcia, 2007; Randel et al., 2009), the Quasi-Biennial Oscillation (QBO) (Baldwin et al., 2001) and a long-term linear trend (Calvo et al., 2010).

$$TUMF(t) = \beta_0 + \beta_{VOL} \cdot x_{VOLC}(t) + \beta_{ENSO} \cdot x_{ENSO}(t) + \beta_{TREND} \cdot x_{TREND}(t) + \beta_{QBO1} \cdot x_{QBO1}(t) + \beta_{QBO2} \cdot x_{QBO2}(t) + \varepsilon(t),$$
$$(6)$$

where $\beta_0$ is a constant, $\beta_i$ is the regression coefficient for basis function $x_i$ and $\varepsilon(t)$ is the residual. Following Maycock et al., (2018), the volcanic basis function is defined as the tropical lower stratospheric average volcanic surface area density (SAD), the ENSO term basis function is the timeseries of based on east-central equatorial Pacific Ocean SST timeseries anomalies (Niño 3.4 index; 5°S to 5°N; 170°W to 120°W), the two orthogonal QBO terms ares the first two principal component

timeseriess from an empirical orthogonal function (EOF) analysis ofin the zonal mean zonal winds between 10°S-10°N and 70 to 5 hPa, and a linear trend. The first three regressors, volcanic, ENSO and the linear trend are identical for both REF-C1 and REF-C1SD runs, while the QBO terms are calculated using the model winds for each model and experiment.  For the REF-C1 runs, CMAM does not include a QBO, hence when we apply the MLR to the CMAM REF-C1 simulation the QBO terms are omitted. We opted not to include an equivalent effective stratospheric chlorine (EESC) MLR term to account for the changes in ozone depleting substances changes (Abalos et al., 2019; Morgenstern et al., 2018; Polvani et al., 2018, 2019) as the period considered in the study is may not be sufficiently long for the linear trend to be separated properly from EESC. Since we are regressing annual mean TUMF we do not consider a seasonal cycle term or any lag in the terms. The results in section 3.5 focus on the first ensemble member (in the rip-nomenclature, where r stands for realization, i for initialization and p for physics - r1i1p1), but where applicable the results from the MLR model for the rest of the ensemble members of the REF-C1 runs are presented in the supplement (Supplementary Figures S67-S12S9).

**2.4 Reanalysis Data**

In order to compare the REF-C1 and REF-C1SD simulations against the reanalysis datasets used for the nudging, we use the SPARC Reanalysis Intercomparison Project (S-RIP) dataset (Martineau, 2017; Martineau et al., 2018). This provides a common gridded version of the reanalysis TEM fields on a 2.5° × 2.5° grid up to 1 hPa. The pressure vertical velocity, $\overline{\omega}^*$, is converted to the residual vertical velocity, $\overline{w}^*$, using equation 2. A detailed comparison of the stratospheric residual circulation in reanalysis datasets has beenis given by  Abalos et al., (2015).

**3 Results**

**3.1 Climatological residual circulation: $\overline{w}^*$**

Figure 1 shows a latitude-pressure cross-sections of the climatological (1980-2009) multi-model mean (MMM) annual mean $\overline{w}^*$ for the REF-C1 (Fig. 1a) and REF-C1SD (Fig. 1b) simulations and their absolute differences (Fig. 1c). In Figure 1c, absolute differences are computed, so positive values indicate where the magnitude of the circulation in REF-C1SD (whether upwelling or downwelling) is larger than in REF-C1. As expected, the climatologies show upwelling occurs in the tropics between around 30°S to 30°N and downwelling at higher latitudes. In the lowermost stratosphere (100-80 hPa), wWithin the region of tropical upwelling, the REF-C1SD runs MMM generally shows larger $\overline{w}^*$ values in the low-to-mid stratospherein the subtropics and smaller values at the equator compared to REF-C1, indicating a tendency for a more double peaked $\overline{w}^*$ structure in the tropics in the lowermost stratosphere (Ming et al., 2016a). Above this, between ~70–4 hPa, the REF-C1SD MMM shows on average stronger

upwelling at the equator compared to REF-C1, indicating a less pronounced double peaked $\overline{w}^*$ structure in the REF-C1SD experiments in the lower to middle stratosphere. ~~In the upper stratosphere between ~4-2 hPa, REF-C1 shows a slightly narrower and more peaked region of upwelling than REF-C1SD, and vice versanear 1 hPa.In theAt~~ southern mid-latitudes, between ~30-60°S, the REF-C1SD  MMM exhibit on average slightly  weaker downwelling than in  REF-C1 , with the largest magnitude differences found in the upper stratosphere.  In the Arctic ,  the REF-C1SD  MMM shows significantly stronger downwelling over the poles than in REF-C1. In the Antarctic the picture is more complex, with the REF-C1SD MMM showing weaker downwelling right at the pole in the upper stratosphere (2-10 hPa), but stronger downwelling between around 75-88°S . In the middle stratosphere, from 50-10 hPa, the REF-C1SD MMM shows stronger downwelling between 60-80°S.

 To show the differences in the transition between regions of upwelling and downwelling motion, Figure 2 shows vertical profiles of the climatological annual mean turnaround (TA) latitudes in each hemisphere for the REF-C1 and REF-C1SD MMM and the three reanalysis datasets used for nudging.  Note that since five of the REF-C1SD models were nudged towards ERA-I, the REF-C1SD MMM may be more weighted towards ERA-I than the other reanalyses. In the Northern hemisphere (NH), the REF-C1SD  MMM  shows a  more poleward  TA latitude compared to both REF-C1 and the reanalyses throughout almost  the entire depth of the stratosphere (Figure 2b).  A more poleward TA latitude for REF-C1SD than in both REF-C1 and the reanalyses is also found in the Southern hemisphere (SH) at pressures greater than 30 hPa (Figure 2a). Hence the nudged simulations show, on average, a wider region of tropical upwelling in the lower stratosphere compared to their free-running counterparts by up to around 5° latitude. In the middle and upper stratosphere the  REF-C1SD MMM shows a narrower upwelling region in the SH . Interestingly, above 10 hPa in the SH (Figure 2a) the REF-C1SD does not show a progressive widening of the upwelling region with decreasing pressure as seen in the reanalyses. This is reflected in the structural differences in $\overline{w}^*$ in the SH upper stratosphere found in

some models (Supplementary Figure S10). It should be noted though that the differences in TA latitudes between the REF-C1 and REF-C1SD MMMs are comparable to the differences found between the three

425 reanalysis datasets.

Focusing on the lower stratosphere, Figure 3 shows the climatological annual mean $\overline{w}^*$ at 70 hPa in the individual models for the (a) REF-C1 and (b) REF-C1SD simulations and (c) their differences. Also plotted in Figb is $\overline{w}^*$  from the  reanalyses

430 and Figure 3d shows the difference between each REF-C1SD simulation and the reanalysis it was nudged toward.  Within the upwelling region, all the models show a clear double peaked $\overline{w}^*$ structure in the tropics with the exception of the CCSRNIES-MIROC3.2 and MRI-ESM1r1 models in

435 the REF-C1SD experiment. In those two cases, CCSRNIES-MIROC3.2 simulates a tri-modal $\overline{w}^*$ structure while MRI-ESM-r1 shows a relatively constant $\overline{w}^*$ across the tropics. For the REF-C1 experiment, Both EMAC simulations,  CMAM and SOCOL3  show a narrower double -

440 peaked structure, with EMAC-L47 exhibiting a rather pronounced NH subtropical maximum. Conversely,  CESM1-WACCM simulates  the broadestregion of tropical upwelling in the lower stratosphere, with the SH  subtropical maximum occurring at higher latitudes compared with the rest of the models. The

445 other REF-C1 simulations also exhibit a double peaked $\overline{w}^*$ structure, which is generally more hemispherically symmetric, but with varying amplitudes.

A double-peaked $\overline{w}^*$ structure in the lower stratosphere has previously been shown in reanalysis datasets (Abalos et al., 2015; Ming et al., 2016) and some CCMs (Butchart et al., 2006, 2010). This can also be seen in Figure 3b for the three reanalysis datasets (ERA-I, JRA-55 and MERRA), where ERA-

450 I and JRA-55 show an asymmetric double- peaked structure with stronger upwelling in the NH  subtropics compared to the SH. As documented by Abalos et al. (2015), based on the direct calculation of the residual circulation, MERRA exhibits downwelling at the equator, an issue which was highlighted in Abalos et al., (2015) manifested as  a negative cell in the streamfunction.

455 Figure 3c shows the absolute differences in  $\overline{w}^*$ at 70 hPa between the REF-C1SD and REF-C1 experiments. Positive values show where the differences magnitude of the circulation in REF-C1SD is larger than

in REF-C1. The largest differences are generally found within the inner tropics, where CCSRNIES-MIROC3.2 , CMAM and MRI-ESM1r1 exhibit significantly stronger upwelling (up to 3 times more for CMAM) near the local $\overline{w}^*$ minimum at the equator. ~~Similarly, for MRI-ESM1r1, which was nudged towards JRA-55, the differences near the equator reflect a reduction in the double-peaked structure of $\overline{w}^*$ in the REF-C1SD experiment. Similarly, EMAC-L90 differences are a manifestation of having a more double-peaked structure in their REF-C1 simulation. EMAC-L47 and CESM1-WACCM although nudged towards ERA-I and MERRA, respectively, show similar and relatively small differences between REF-C1SD and REF-C1 both within the upwelling region and the downwelling regions in both hemispheres.in the SH mid-latitudesREF-C1 shows stronger downwelling in the mid-latitudes (up to 50° in both hemispheres) compared to the nudged models-In the subpolar andnudgednudgedfree-running~~ counterparts, while in the NH extratropics no consistent picture emerges across the models. EMAC-L47 and EMAC-L90 show markedly different behaviours despite the fact they are nudged towards the same reanalysis (ERA-I) and differ only in their vertical resolution. This indicates that the effect of nudging on the mean residual circulation is likely to be sensitive to a great many factors that vary from model to model.

Another interesting  result from Figure 3 is that the inter-model spread in $\overline{w}^*$ for both experiments is larger in the NH downwelling region than in the equivalent region of the SH. Specifically, the inter-model spread is 0.14 mm s$^{-1}$ for the REF-C1 runs for all points between 30°S - 80°S and 0.2 mm s$^{-1}$ for points between 30°N - 80°N, while for REF-C1SD the values are 0.12 mm s$^{-1}$ and 0.19 mm s$^{-1}$, respectively. This also demonstrates that the inter-model spread in $\overline{w}^*$ in the REF-C1SD simulations is comparable to that in REF-C1   at  extratropical latitudes . In contrast, in the tropics between 30°S - 30°N the REF-C1SD simulations exhibit a slightly larger inter-model spread than the free-running simulations (0.09 mm s$^{-1}$ vs. 0.07 mm s$^{-1}$).

~~Comparing the SD models against the reanalysis datasets they are nudged towards reveals some distinct differences. For example, CESM1-WACCM is nudged towards MERRA as shown in Figure 2b, but it does not show downwelling around the equator as the direct MERRA estimate, while it does resemble the reanalysis in the northern subtropics quite closely. MRI-ESM1r1 in the REF-C1SD experiment is nudged towards JRA-55 and shows a rather flat latitudinal $\overline{w}^*$ structure with no clearly defined local maxima and does not resemble the reanalysis in the inner tropics. As for the models~~

 Figure 3d shows the absolute differences in $\overline{w}^*$ between the REF-C1SD simulations and the respective reanalysis dataset used for nudging. In the upwelling region, the REF-C1SD experiments generally show stronger upwelling near the equator than in the reanalyses. Although CESM1-WACCM is nudged towards MERRA, it does not simulate downwelling at the equator as seen in the MERRA direct estimate. The relatively larger $\overline{w}^*$ differences near 10---15°N in CCSRNIES MIROC3.2, EMAC-L90 and MRI-ESM-r1 reflect a lack of inter-hemispheric asymmetry in the double peaked $\overline{w}^*$ structure in the REF-C1SD experiment compared to the reanalyses. Outside of the tropics, the REF-C1SD experiments generally show weaker downwelling in the NH mid-latitudes, while at polar latitudes (>65°) the REF-C1SD runs consistently show stronger downwelling than in the reanalyses. The difference in $\overline{w}^*$ at high latitudes between the REFC1-SD and reanalysis datasets extends throughout the depth of the stratosphere (see Supplementary Figure S10). More generally, Figure 3b shows that the different models that all nudge towards ERA-I (CCSRNIES MIROC3.2, CMAM, EMAC-L47/90, SOCOL3) produce very different mean residual circulations.

In summary, we conclude based on the results in Figures 1 to 3 that nudging meteorology does affect the strength and structure of the climatological residual circulation throughout the stratosphere. However, as implemented in these simulations (Table 2), nudging does not strongly constrain the mean amplitude and structure of the residual circulation nor does it produce circulations that closely resemble the direct estimates from the reanalyses.

**3.2 Climatological residual circulation: tropical upward mass flux**

Figure 4 shows  vertical profiles of the climatological TUMF between 100 and 3 hPa calculated from annual means of $\overline{w}^*$  for the (a) REF-C1, (b) REF-C1SD experiments and (c) their difference. Note the logarithmic x-axis scale and that the CCMI and S-RIP fields have been interpolated from their native model levels to a set of predefined common pressure levels, which are rather sparse in the upper stratosphere; hence the TUMF calculation could be different if it was performed on the native model grid of both CCMI models and the reanalyses.

In terms of the differences between the REF-C1SD and REF-C1 simulations (Figure 4c), there is no consistent picture of the effect of nudging on the TUMF at different stratospheric levels. In the lower stratosphere between 70-100 hPa, most models  (apart from  EMAC-

L90),  simulate stronger TUMF in the REF-C1SD runs than  in REF-C1 . The largest TUMF differences  in the lower stratosphere due to nudging  occur in  EMAC-L90 and SOCOL3 , which show  differences at 90 hPa  of around -20% and +25%, respectively. In the middle  stratosphere, between 10-7 hPa, some models show almost no difference in TUMF due to nudging (MRI-ESMr1), some show a stronger mass flux (CCSRNIES-MIROC3.2, CESM1-WACCM, CMAM) and others a weaker mass flux (EMAC-L47, SOCOL3). ~~In CESM1-WACCM, the differences in TUMF between the REF-C1SD and REF-C1 simulations are generally positive throughout the stratosphere and reach 20% at 20 hPa and ~25% at 3 hPa. The MRI-ESM1r1 model generally shows the smallest differences in TUMF between the two experiments., including EMAC-L47 and EMAC-L90 which show opposite sign differences at higher altitudes~~pressures. CMAM shows the smallest change in TUMF in the upper stratosphere due to nudging. CESM1-WACCM is the only model to show a consistent sign of the TUMF differences between REF-C1SD and REF-C1 at all levels, with higher TUMF found throughout the stratosphere. There is no apparently simple relationship between the free-running model TUMF climatologies (Figure 4a) and the effect of nudging (Figure 4c).

We now cd the TUMF in each REF-C1SD experiment with  the  reanalysis they was nudged towards (Figureb and 44d). Taking at first a broad view of the entire profiles, there is a resemblance between the profiles of TUMF differences in EMAC-L47 and SOCOL3 as compared to ERA-I, which may be related to the similarities in the implementation of nudging in these models; for example, vorticity and divergence were nudged with the same relaxation parameters (see Table 2). The CESM1-WACCM REF-C1SD simulation generally shows larger TUMF values than MERRA by up to 10-15% apart from in the upper stratosphere where they start to converge. MRI-ESM1r1 exhibits relatively better agreement of TUMF with JRA-55 throughout the stratosphere. Looking across the models, most of the REF-C1SD simulations  simulate stronger upwelling than thei respective reanalysis in the upper stratosphere, with differences reaching up to 30-35% in the two EMAC models. In fact, EMAC-L47/L90 show a high degree of similarity in the vertical structure of the TUMF differences between REFC1-SD and ERA-I at pressures less than 30 hPa, despite showing substantial differences in the lower stratosphere. ~~This could be partly due to the fact that both the CCMI and S-RIP fields have been interpolated from their native model levels to a set of predefined common pressure levels which are rather sparse in that vicinity, hence the TUMF calculation could be different if it was performed on the native model grid of both CCMI models and the reanalysis. Additionally, for both EMAC REF-C1SD simulations the differences from the reanalysis in the TUMF~~

above 10 hPa can also This could be explained by the fact that because in EMAC the nudging is only imposed strongly up to 10 hPa, while higher model layers have weakening nudging coefficients as they serve as transition layers. In the middle stratosphere (50-20 hPa), most of the REF-C1SD models simulate a lower TUMF compared to the reanalysis. CESM1-WACCM generally shows slightly larger TUMF values than MERRA apart from the upper stratosphere where they start to converge. MRI-ESM1r1 exhibits relatively better agreement of TUMF with JRA-55 throughout the depth of the stratosphere, while CMAM which is nudged towards ERA-I follows closely the reanalysis especially above 10 hPa but is generally biased low In terms of the spread found in both sets of experiments, in the lower stratosphere (100-30 hPa) where the maximum spread is located, the nudged simulations and their free-running counterparts show a total spread of $3.26 \times 10^9$ kg s$^{-1}$ and $3.1 \times 10^9$ kg s$^{-1}$ respectively, though note there are fewer REF-C1SD simulations to consider. Again, a key message is that the nudged REF-C1SD simulations show if not a slightly larger, a comparable, if not a slightly larger, spread in the climatological TUMF compared to the free-running REF-C1 simulations, throughout almost all the depth of the stratosphere.

To understand the dynamical factors that contribute to the modelled climatological residual circulation and its spread, Figure 54 shows the annual mean TUMF at 70 hPa along with the downward control calculations (section 2.2.2) to quantify the contribution of resolved and parameterized wave forcing to the TUMF. The black bars on the left show the TUMF diagnosed from $\overline{w}^*$ and the grey bars on the right show the estimated contribution to TUMF from the Eliassen -Palm flux divergence (EPFD, dark grey), the orographic (mid-grey) and non-orographic (light grey) gravity wave drag. Note that SOCOL3 SOCOL3 and ULAQ-CCM did not provide any wave forcing fields (Table 3), and NIWA-UKCA only provided the EPFD so we cannot perform the downward control calculations for that model.

In the free-running REF-C1 simulations (Fig. 54a), the estimated TUMF from the total wave forcing for the majority of the models (apart from CESM1-WACCM and EMAC-L90), slightly exceeds the TUMF calculated directly from $\overline{w}^*$; this was not the case though for the CCMVal-2 models in SPARC (2010) (Figure 4.10, p.121). Since these simulations are internally consistent, the imperfect match indicates that the downward control principle as applied here relies on the close but inexact applicability of certain assumptions such as the system being in a steady-state in response to a steady mechanical forcing (Haynes et al., 1991). The REF-C1 inter-model range in TUMF at 70 hPa is $5.74 \times 10^9$ to $6.62 \times 10^9$ kg s$^{-1}$ (inter-model standard deviation = $0.29 \times 10^9$ kg s$^{-1}$). Comparing the CCMI results in Figure 5a with the that figure results from CCMVal-2 models (see Figure 4.10; SPARC, 2010) with the results in Figure 4a, the MMM TUMF at 70 hPa ($5.9 \times 10^9$ kg s$^{-1}$) for the ten seven REF-C1 model simulations analysed here ($6.05 \times 10^9$ kg s$^{-1}$) is in very close agreement within the inter-model range of the when compared to the MMM of the fourteen CCMVal-2 models, which show a MMM TUMF around 4% weaker ($5.8 \times 10^9$ kg s$^{-1}$) (SPARC, 2010). In terms of the contribution of the resolved wave

forcing to the TUMF in the free running simulations, there appears to be a decreased inter-model range ($3.26\times10^9$ to- $5.33\times10^9$ kg s$^{-1}$) in the present study compared with the CCMVal-2 models, albeit that study included more models ($1.5\times10^9$ to $5.5\times10^9$ kg s$^{-1}$) (SPARC, 2010). Some CCMI models have increased their horizontal resolution by up to a factor of two (CMAM, MRI-ESM1r1, SOCOL3 ) and also their vertical resolution up to 80 vertical levels (MRI-ESM1r1) compared with CCMVal-2 models (Dietmüller et al., 2018), which could improve their ability to simulate resolved wave forcing. There is a notable feature of CMAM which shows that the NOGWD contributes negatively to TUMF (indicated with two red horizontal lines on Figure 5 and Supplementary Figure S11); this was also found for CMAM in CCMVal-2 (Figure 4.10; SPARC, 2010).

The MMM TUMF at 70 hPa in the REF-C1SD simulations (Figure 5b) is $6.32\times10^9$ kg s$^{-1}$ or around 5% higher than in REF-C1. The REF-C1SD model range is larger than in REF-C1 being $5.39\times10^9$ to $7.08\times10^9$ kg s$^{-1}$ (inter-model standard deviation = $0.51\times10^9$ kg s$^{-1}$). A notable feature is that the contribution from the individual and total wave forcing contributions shows reduced inter-model spread in the REF-C1SD simulations (Figure 5b, darker grey bars). For example, the inter-model standard deviation of the EPFD contribution to TUMF at 70 hPa is around 40% smaller than in REF-C1 ($0.44\times10^9$ kg s$^{-1}$ and $0.72\times10^9$ kg s$^{-1}$, respectively). ~~Interestingly, for the single simulations that were nudged towards MERRA and JRA-55 (CESM1-WACCM and MRI-ESM1r1, respectively) the TUMF in the REF-C1SD runs is closer to the estimates from the reanalyses they are nudged towards. This may simply be a coincidence given that there remain substantial differences in the structure of $\overline{w}^*$ between the REF-C1SD simulations and reanalyses (Figure 2b) and this does not apply necessarily for all models that were nudged towards ERA-I. Another notable feature is that the contribution from the individual and total wave forcing contributions shows reduced inter-model spread in the REF-C1SD simulations (Figure 4b, grey bars). For example, the inter-model standard deviation of the EPFD contribution to TUMF at 70 hPa is 43% smaller than in REF-C1 ($0.43\times10^9$ kg s$^{-1}$ and $0.76\times10^9$ kg s$^{-1}$, respectively).~~

Nonetheless, the residuals (i.e. the difference between the directly calculated TUMF and the total downward control estimated contribution from the wave forcing) are substantially larger and positive (except for EMAC-L90) in the REF-C1SD experiment than in REF-C1. This shows that nudging adds an additional non-physical tendency in the model equations which acts to decouple the wave forcing from the residual circulation; this means the physical constraint that the divergence of the angular momentum flux due to the mean motion is balanced over some sufficient time average by that of all eddy motions does not apply in the nudged models (Haynes et al., 1991). The details of how this decoupling is manifested is likely to vary from one model to another depending on multiple factors such as nudging timescales, nudging parameters, nudging height range, and model resolution. Comparison of the TUMF at 10 hPa for the REF-C1SD experiment (see Supplementary Figure S11b) also reveals substantial differences in some models between the direct and downward control TUMF estimates in

the middle stratosphere. Variations in the residuals as a function of height may indicate differences in the effect of nudging on the connection between the climatological wave forcing and the shallow and deep branches of the circulation (Birner and Bönisch, 2011). However, the inter-model ranges in the directly calculated TUMF at 10 hPa are more comparable in the two experiments than was found at 70 hPa ($1.45\times10^9$ to $1.70\times10^9$ kg s$^{-1}$ and $1.51\times10^9$ to $1.72\times10^9$ kg s$^{-1}$ for REF-C1 and REF-C1SD, respectively) (Supplementary Figure S11b).

Interestingly, for the single simulations that were nudged towards MERRA and JRA-55 (CESM1-WACCM and MRI-ESM1r1, respectively) the TUMF at 70 hPa in the REF-C1SD runs appear to be close to the estimates from the reanalyses they are nudged towards (compare black bars in Figure 5b). This may simply be a coincidence given that there are substantial differences in the structure of $\overline{w}^*$ between the REF-C1SD simulations for those models and the reanalyses (Figures 3b and 3d), and this is not found for all 5 models that were nudged towards ERA-I. Indeed, given there is substantial spread in TUMF amongst the 5 REFC1-SD models nudged to ERA-I, it is likely that the differences between the REFC1-SD and reanalysis datasets are related to how nudging was implemented in each model; a wide variety of relaxation timescales and vertical nudging ranges were used by the models (Table 2). Despite this, t The lower TUMF calculated directly from $\overline{w}^*$ in EMAC-L90 compared to EMAC-L47 seen in both the REF-C1 and REF-C1SD experiments, is consistent with the results of Revell et al. (2015b) who also find that an increase in the model vertical resolution for SOCOL3SOCOL3 results in a slowdown of the BDC.

Comparison of the TUMF at 10 hPa for the REF-C1SD experiment (see Supplement Figure S23b), reveal that the residuals are smaller in the middle stratosphere and as stated previously for both EMAC simulations, 10 hPa is the maximum level that the nudging is applied. Nevertheless, the rest of the REF-C1SD models were nudged even above that level also show smaller residuals and this may indicate differences in the effect of nudging on the shallow versus the deep branch of the circulation (Birner and Bönisch, 2011). In summary, the results from Figures 4 and 5 reflect further demonstratethe fact that nudging imparts an external and non-physical tendency in the model equations, which in turn might cause violations of the normal constraints on the global circulation, such as conservation of momentum and energy. This is found to alters the residual circulation but in a manner that cannot be understood fromand appears to limit the ability to a closure ofe the circulation through the integrated wave forcing as would ordinarily apply in the downward control principle (Haynes et al., 1991).

**3.3 Annual cycle**

We now evaluate the representation of the annual cycle in the residual circulation. Figure 65 shows the MMM climatological annual cycle of $\overline{w}^*$ at 70 hPa for the REF-C1 and REF-C1SD simulations and their difference. Note there are no significant variations in the results when the MMM of the REF-C1

Both experiments show similar broad features in the annual cycle, with stronger tropical upwelling in boreal winter, a latitudinal asymmetry in the region of upwelling with the  TA latitude being further poleward in the summer hemisphere, and stronger downwelling over the winter pole. These

675 features resemble the annual cycle found in other multi-model studies (e.g. Hardiman et al., 2014). Figure 6c shows that on average the nudged models simulate stronger upwelling in the sub-tropics, particularly in the NH in boreal winter with a few exceptions; the most prominent one being the narrow band between the equator and 10°N where the REF-C1 simulations exhibit stronger upwelling in austral winter. Consequently, the nudged models simulate substantially stronger downwelling in the

680 midlatitudes in winter. In the NH mid-latitudes in the summer months nudged runs show  weaker downwelling, which reverses for the SH mid-latitudes in the austral winter. At polar latitudes there is a distinct seasonality to the differences between the REF-C1SD and REF-C1 simulations, with the nudged models simulating stronger downwelling in boreal winter and weaker downwelling in the Arctic during the rest of the year, corresponding to an amplified annual cycle. Conversely in the

685 Antarctic, the REF-C1SD simulations generally simulate weaker downwelling, particularly during austral summer and spring.

To compare the annual cycle in residual circulation  in the individual models, Figures 7a and 7b show the mean tropical (30°S - 30°N) $\overline{w}^*$ at 70 hPa for the REF-C1 and REF-C1SD simulations, respectively. Comparing the MMM annual cycle of the REF-C1 runs (Figure 7a) with the MMM REF-

690 C1SD (Figure 7b) reveals that on average the nudged models show a slightly larger peak-to-peak annual cycle amplitude (0.16 mm s$^{-1}$ vs. 0.13 mm s$^{-1}$). In general, the amplitude  of the annual cycle in tropical mean $\overline{w}^*$ is  slightly more constrained across the REF-C1SD simulations with the  spread in peak-to-peak amplitude, as measured by the inter-model standard deviation, being around 25%  smaller than

695 in  REF-C1 ($\sigma$ = 0.015 mm s$^{-1}$ vs. 0.020 mm s$^{-1}$, respectively) In terms of seasonal mean behaviour, the nudging appears to constrain the tropical mean  $\overline{w}^*$ in boreal summer (JJA), which exhibits ~20% less spread than in the REF-C1 experiments, but it does not constrain the tropical mean

700 $\overline{w}^*$ in boreal winter (DJF), which shows a factor of two larger spread than the free running models. Furthermore, the differences in tropical mean $\overline{w}^*$ between the REF-C1SD runs and the respective reanalysis they are nudged towards are generally larger in boreal winter than in boreal summer for most models. In terms of spatially-resolved differences in $\overline{w}^*$ between REF-C1SD and the reanalyses (Supplementary Figure S12), some consistent features include the REF-C1SD simulations showing

705 stronger downwelling in the Arctic in boreal winter compared to the reanalyses and showing weaker

upwelling in the northern subtropics in boreal summer and autumn. Overall, the REF-C1SD minus reanalysis differences for the individual models highlight a wide variety in both the magnitude and the spatial patterns of their absolute differences, with no consistent picture emerging even for the models nudged towards the same reanalysis dataset.

710    Figures 7c and 7d show the climatological annual cycle in the  TA latitudes at 70 hPa for the REF-C1 and REF-C1SD runs, respectively. This further breaks down the MMM annual mean perspective shown in Figure 2 by model and by season. In the SH, the spread in seasonal mean TA latitude across models, as measured by the intermodel standard deviation, is increased in the REF-C1SD experiment in all seasons by up to 30% compared to REF-C1. Conversely in the NH, the spread in

715 seasonal mean TA latitude is decreased for REF-C1SD in all seasons except boreal spring (MAM) where it is increased. There are also substantial differences between the TA latitudes in the REF-C1SD experiment and the reanalyses in all months, which shows that nudging does not produce consistent structures of regions of upwelling and downwelling to those in the reanalysis.

720 ~~Figure 6d, while the reanalyses also exhibit some differences. Note for this measure we exclude ULAQ-CCM when calculating the inter-model spread for the REF-C1 runs because it lacks a realistic seasonal evolution in its TA latitudes. The REF-C1SD runs show a weaker annual cycle in the SH TA latitude, with the models showing a consistently more poleward TA latitude in the SH in austral winter (JJA). In the NH, the TA latitudes in the REF-C1SD runs show a slightly smaller annual cycle compared to the~~

725   To summarize the results of Figure 7, there is substantial inter-model spread in the  TA latitudes and in the amplitude of the annual cycle in $\overline{w}^*$ highlighting significant interhemispheric differences in the upwelling region  between both sets of simulations as well

730 between the nudged experiment and the reanalyses.

**3.4 Interannual variability of the tropical upward mass flux**

Figure 8 shows timeseries over 1980-2009 for the annual, December-January-February (DJF), and

735 June-July-August (JJA) mean TUMF at 70 hPa for the REF-C1 (left column) and REF-C1SD (right column) simulations. As expected, the TUMF is larger in DJF compared to the annual and JJA means in both the REF-C1 and REF-C1SD runs because the average tropical upwelling is stronger in boreal winter. The individual REF-C1SD simulations show remarkably similar temporal variability in contrast to REF-C1 –where the modeled interannual variability is very diverse despite the models all being

740 forced with observed SSTs. Hence, although nudging does not constrain the mean TUMF in the lower

stratosphere, it does constrain the interannual variability; this is even more apparent for the DJF and JJA seasonal means (Figures 8d,f). Additionally, the REF-C1SD  simulations show  a  relatively high agreement in their temporal variability to the reanalysis datasets they were nudged towards, albeit with  differences in magnitude and  trends  at the beginning of the 21st century where ERA-I and MERRA show a decrease in TUMF.

To investigate the cause of the high temporal  coherence of the REF-C1SD  TUMF timeseries,  Figure 9  presents the annual mean TUMF anomalies at 70 hPa along with the relative contributions  from EPFD, OGWD, NOGWD and the total parameterized wave forcing (from top to bottom panels) for REF-C1 (left column) and REF-C1SD (right column), respectively. Figure 9b shows again the remarkably similar temporal variability in TUMF across the REF-C1SD runs, which can be contrasted against the weak interannual coherence in the REF-C1 runs (Figure 9a). Figure 9d and 9j show that both the EPFD and the total parametrized wave forcing contributions to the TUMF show a high degree of temporal coherence in the REF-C1SD simulations. The fact that the individual OGWD and NOGWD terms do not show such a strong inter-model agreement, while the total parametrized wave forcing does, could suggest there is some compensation occurring between the different parameterised wave forcing components (e.g. Cohen et al., 2013).  It should be noted that the reanalyses have been shown to exhibit strong similarities in their resolved EP fluxes as shown by the linear correlation in the timeseries of tropical upwelling at the 70 hPa level when considering the momentum balance estimates of $\overline{w}^*$ (Abalos et al., 2015). This result indicates that although nudging does not constrain the mean residual circulation, it does constrain the interannual variability and produces  similar contributions to variability across models from both resolved and parameterized wave forcing. In contrast, the REF-C1 simulations show a highly variable pattern of the estimated TUMF anomalies from EPFD and  parameterized wave forcing (Figures 9c, 9i), despite the fact they use the same observed SSTs and some nudge the phase of the QBO (CCSRNIES-MIRCO3.2, CESM1-WACCM, EMACL47/L90 and SOCOL3

ULAQ-CCM). In summary, the  remarkably coherent interannual variability in the annual TUMF timeseries in the REF-C1SD simulations  is due to both the resolved and parametrized wave forcing being constrained by nudging; this is in strong contrast to  the climatological strength of the TUMF where there were large differences between the directly calculated TUMF and that due to wave forces (Figure 5b). The reasons for the difference in the effect of nudging on the behaviour of the residual circulation between the long-term mean and interannual variability is unclear.

**3.5 Multiple Linear Regression analysis**

Figures 10 and 11 show  timeseries of annual TUMF anomalies at 70 hPa attributed to each of the basis functions in the MLR model described in section 2.3 and the regression residuals for the REF-C1 and REF-C1SD runs, respectively. Also shown in the supplementary Figures S13 and S14 are the regression coefficients for each term and for each model along with their uncertainties. Figure 10a shows a large spread in the diagnosed signal of volcanic eruptions in the TUMF timeseries. The majority of the REF-C1 simulations analyzed here show a negative TUMF anomaly around the time of the El Chichón (1982) and Mount Pinatubo (1991) eruptions; however, the magnitude is within the estimated uncertainty range for all models except SOCOL3 (Supplementary Figure S13). In contrast to the REF-C1 results, most REF-C1SD simulations (except EMAC L47/L90 - see above discussion) show a positive anomaly in TUMF attributed to volcanic eruptions (Figure 11), consistent with earlier studies (Garcia et al., 2010; Diallo et al., 2017). However, there is still a considerable range of amplitudes and only the CESM1-WACCM and MRI-ESM1r1 regression coefficients are highly significant (Supplementary Figure S14).  The issue of establishing a robust response of the TUMF to volcanic forcing over a short period is  demonstrated by the range in  amplitudes of the volcanic regressors   different REF-C1 ensemble members from the same model (see Supplement Figures S7-S12). This highlights that in a free-running climate simulation, internal variability can overwhelm the response to forcing over short timescales. The "true" volcanic signal in TUMF will also depend on the representation of stratospheric heating due to aerosol in the various models. We note that the EMACL47/L90 models contained a unit conversion error where the extinction of stratospheric aerosols was set too low by a factor of ~500 (see Appendix B4 of Morgenstern et al., 2017), hence the stratospheric dynamical effects of the eruptions were not properly represented in the EMAC simulations (Jöckel et al., 2016).

MRI-ESM1r1 and CESM1-WACCM showing the largest volcanic responses. The reason that the EMACL47/L90 models show a negative TUMF anomaly to volcanic forcing in both REF-C1 and REF-C1SD runs is documented in Appendix B4 of Morgenstern et al. (2017) and relates to a unit conversion error where the extinction of stratospheric aerosols was too low, by a factor of ~500, hence the stratospheric dynamical effects of those eruptions were not represented (Jöckel et al., 2016). The REF-C1 models all show a positive best estimate regression coefficient for the TUMF response to ENSO (Figure 10), which is quite consistent in amplitude, but it is only strongly statistically significant in CCSRNIES-MIROC3.2 and SOCOL3 (Figure S5). This is in contrast to the REF-C1SD models which all show a larger and more significant positive ENSO regression coefficient. re are differences amongst models in the amplitude of the variance in the TUMF attributed to ENSO and the linear trend, but they are all consistent in sign (i.e. positive TUMF anomaly for El Nino and positive long-term trend). Despite the inter-model spread, for each model that performed both REF-C1 and REF-C1SD experiments the ENSO and linear trend contributions to the TUMF anomalies are quite similar, although the magnitude varies. The linear trend regression coefficient over 1980-2009 is positive in all REF-C1 models and is statistically significantly different from zero at the 95% confidence level in five out of the seven models. The magnitude of the linear trend term varies by around a factor of 2 for REF-C1. In REF-C1SD, the amplitude of the linear trend regression coefficient increases in all models, but the intermodel spread increases to around a factor of 4. Hence, in these simulations nudging increases the disparity across models in the magnitude of the long-term TUMF trend.

As expected, the variations in TUMF attributed to the QBO are quite different in the REF-C1 and REF-C1SD runs for those models that do not nudge the QBO in REF-C1, as shown in Figures 109 and 110. The nudging of zonal winds in REF-C1SD constrains the phase of the QBO, and hence there is strikingly similar variability in the TUMF anomalies attributed to the QBO in the REF-C1SD runs. The linear trend coefficient is statistically significantly different from zero at the 95% confidence level in five out of the six REF-C1SD models over the period 1980-2009, in contrast to being significant in half four of the REF-C1 simulations.

The overall $R^2$ values from the MLR model for the REF-C1 simulations vary between 0.16 (CMAM) and 0.67 (CESM1-WACCM). REF-C1SD runs generally give more consistent $R^2$ values across the models ranging from 0.62 (CCSRNIES-MIROC3.2) to 0.77 (EMAC-L47). This means there is still a substantial fraction (>23%) of unexplained variance in the annual TUMF timeseries in the REFC1-SD simulations after applying the MLR model, which and the residuals exhibits a remarkable degree of temporal correlation. In contrast, the MLR residuals in the REF-C1 runs (Figure 109f) show much less temporal coherence apart from a drop around 1989. , which is also apparent in the REF-C1SD runs (bottom panel Figure 10). In contrast, theThe residuals in the REF-C1SD simulations (Figure 110f) show a high degree of coherent interannual variability, another manifestation of the fact that the nudged runs do reproduce a much more consistent inter-annual variability. This makes a substantial

contribution to the coherence of the TUMF timeseries in Figure 9b, but it cannot be attributed to any of the terms included in the MLR model.

For completeness, the MLR model was also applied to the reanalysis TUMF at 70 hPa (Supplementary Figures S15 and S16). This highlights significant discrepancies in attributing the variance in TUMF in the different reanalysis datasets to the various basis functions in the MLR model. Both the volcanic activity and ENSO contributions to the variance in the TUMF is rather weak compared to the REF-C1SD runs. The negative linear trend in ERA-I is in strong contrast to the positive trends found in the other reanalyses and the REF-C1SD models. The negative trend in ERA-I found in the TUMF in the lower stratosphere over 1980-2009 corroborates the findings of Abalos et al. (2015) who showed a negative trend in the direct $\overline{w}^*$ estimate in ERA-I over 1979-2012. Despite this difference in the representation of the long-term TUMF trend, ERA-I shows the highest percentage of TUMF variance explained by the MLR model (66%), with MERRA showing a substantially lower $R^2$ (0.3) compared to the other reanalyses and the REFC1-SD models. The residuals are generally less correlated between the reanalyses on interannual timescales than was found in the REFC1-SD simulations, but are broadly similar on inter-decadal timescales. However, the regression residuals in the reanalyses show a different temporal behavior from those in the REF-C1SD simulations (Figure 11) (note that the y-axis scale for the residuals in Supplementary Figure S15 is double that for the CCMI models in Figures 10 and 11). In summary, although nudging constrains the interannual variability in the TUMF at 70 hPa, the attribution to some specific drivers differs across the models and in comparison, to the reanalyses they were nudged towards.

**3.6 Trend sensitivity analysis**

Following  the results of the MLR analysis described in section 3.5, which showed a statistically significant positive linear trend in most  REF-C1 and REF-C1SD models for the 30-year period 1980-2009, we now explore the sensitivity of the linear trend to the time period considered. We apply the same MLR model as discussed in section 3.5 to the annual mean 70 hPa TUMF timeseries of the first ensemble member for both REF-C1 and REF-C1SD runs as well as the reanalyses, but systematically vary the start and end dates to cover all time periods in the window 1980-2009 that are at least ten years in length. We then extract the linear trend coefficient from the MLR model and its associated p-value. Figures 12 and 13 present the linear trend calculations for the REF-C1 and REF-C1SD runs, respectively, as a function of trend start and end date. The same trend sensitivity analysis for the reanalyses is presented in the supplement (Supplementary Figure S17). Statistically significant trends at the 95% confidence level are marked with black stippling.

None of the periods considered in either the REF-C1 or REF-C1SD experiments shows a significant negative TUMF trend. A statistically significant positive trend emerges in almost all of the SD models for trends beginning in the mid-1980s to early 1990s extending to the mid-2000s. The trends are mainly significant for periods of 20 years or more and no less than around 12 years. This result broadly corroborates the findings of Hardiman et al. (2017a) who used a control run to estimate the period required to detect a BDC trend with an amplitude of 2%  decade$^{-1}$  against the background internal variability. There is range of different structures in the diagnosed trends among models, particularly for the REF-C1 simulations where a consistent pattern of positive trends only emerges across most models for the entire time period. This is because  internal variability  can mask BDC trends over short periods (Hardiman et al., 2017a). However, the REF-C1SD runs simulate more consistent variations in TUMF trends as a function of time period, but generally show  stronger positive trends  than their free-running counterparts. Interestingly, the reanalysis trend sensitivity analysis highlights that nudging does not constrain the underlying trends of the REF-C1SD models in the TUMF at 70 hPa, as the reanalysis datasets exhibit a wide range of different trends from one another (Supplementary Figure S17) and differences compared to the trends in the REF-C1SD simulations (Figure 13). For example, none of the REFC1-SD models simulate a statistically non-significant negative trend in TUMF starting around mid-1990s up to 2009, as seen in all the reanalyses. However, it should also be noted that any trend combination starting around the end of 1990s in almost all cases of both REF-C1 and REF-C1SD runs exhibit no statistical significance possibly pointing towards the role of declining ozone depleting substances (ODS) due to the implementation of the Montreal Protocol (Polvani et al., 2018).

**4. Conclusions**

This study  has performed the first multi-model intercomparison of the impact of nudged meteorology on the representation of the stratospheric residual circulation. We use hindcast  simulations over 1980-2009 from CCMI with identical prescribed  external forcings in two configurations: REF-C1SD with meteorological fields nudged towards reanalysis data (specified dynamics, SD) and REF-C1 that is free-running . The nudged simulations use one of three different reanalysis datasets (ERA-Interim, JRA-55, MERRA), nudge different variables (u, v, T, vorticity, divergence, surface pressure), and use different time constants to impose the additional nudging tendencies in the model equations.

The key findings of this study are:

1. Nudging  meteorology does not  constrain the mean strength of the residual circulation compared to free-running  simulations. In fact, for most of the metrics of the

climatological  residual circulation examined, including residual vertical velocities and mass fluxes, the inter-model spread  is comparable or  in some cases larger in the REF-C1SD  simulations than in  REF-C1 .

920   1.   Nudging  leads to  the models simulating on average  stronger upwelling at the equator in the lower to middle stratosphere and a wider tropical pipe in the lower stratosphere. In most cases,  the magnitude and structure of the climatological residual circulation in the REF-C1SD experiments differs markedly from that estimated for the reanalysis they are nudged towards.
925

2.

3.   In  most of the nudged  models there are large differences of up to 25%  between the directly calculated  tropical upward mass flux in the lower stratosphere  and that calculated from
930    the diagnosed total wave forcing using the downward control principle (Haynes et al., 1991).  However, the spread in the contributions from the resolved and parametrized wave forcing to the tropical mass flux is slightly reduced in the  REF-C1SD simulations compared to REF-C1.

4. Despite the lack of  agreement in the mean circulation, nudging tightly constrains
935   the interannual variability in the tropical upward mass flux (TUMF) in the lower stratosphere . Thi is is associated with constraints to the contributions from both the resolved and parametrized wave forcing despite the fact the models use different reanalysis datasets for nudging. The reanalysis datasets themselves exhibit broadly similar interannual variability in TUMF in the lower stratosphere, albeit with different
940   long-term trends.

 A multiple linear regression (MLR) analysis showed that up to 77% (67%) of the interannual variance of the lower stratospheric  TUMF  in the  REF-C1SD (REF-C1) experiments can be explained by  volcanic eruptions, ENSO, the QBO and a linear trend. The remaining unexplained TUMF variance in
945   the nudged models shows a high degree of a temporal  coherence,  but this is not the case for the free-running simulations.

5.

6.    The results of the MLR analysis applied to the TUMF in the reanalyses show differences in the total
950    variance explained  and the attribution of variance to the different physical proxies . There are also marked differences between the

individual regression coefficients derived for the REF-C1SD models compared to the reanalysis dataset used for nudging.

7. Most nudged simulations show a statistically significant positive trend in TUMF in the lower stratosphere over 1980-2009, which is on average larger than the trends simulated in the free-running models. This is despite the fact that five out of the seven models analyzed were nudged towards ERA-Interim, which shows a negative long-term trend in TUMF (see also Abalos et al., 2015), while JRA-55 and MERRA show a positive trend. However, the magnitude of the TUMF trend varies by up to a factor of 4 across the nudged models, which is larger than the spread in the free-running simulations. This is an important limitation for using nudged CCM simulations to interpret long-term changes in stratospheric tracers.

 A  sensitivity analysis  of the time period  for calculating lower stratospheric TUMF trends  shows that a statistically significant (at the 95% confidence level)  positive  trend in TUMF takes at least 12 years and in most cases around 20 years to emerge in the REF-C1SD runs. Despite the three reanalysis datasets showing different 30-year trends (1980-2009) they show a striking agreement in the statistically non-significant negative trends starting from the late 1990s up to 2009.

8.

Our findings highlight  that nudging strongly affects the representation of the stratospheric residual circulation in chemistry-climate model simulations, but it does not necessarily lead to improvements in the circulation. Similar disagreement in the characteristics of tropospheric transport in the CCMI nudged simulations has also been reported (Orbe et al., 2018). The differences found in the nudged runs compared with the free-running simulations suggest that although nudging horizontal fields can remove  model bias in, for example,  temperature and horizontal wind fields (Hardiman et al., 2017b), the simulated vertical wind field will not necessarily be similar to the reanalysis. A particularly interesting finding of our study is that  nudging does not constrain the mean strength of the residual circulation,  it does constrain the interannual variability.

The reason for the distinct effects of nudging on the residual circulation across these different timescales  currently unknown.

Multiple factors are likely to determine the effect of nudging on the residual circulation in a given model including model biases, nudging timescales, nudging parameters, nudging height range, and model resolution. The differences in the stratospheric residual circulation between the REF-C1SD and the REF-C1 runs may not arise solely from the dynamics, but can also be partly influenced by the

indirect effects of nudging the temperatures which in turn affects the diabatic heating (Ming et al., 2016a, 2016b). In addition to nudging the horizontal winds (mechanical nudging), nudging the temperature (thermal nudging) might be systematically creating a spurious heat source in the model, which leads to a stronger BDC in the lower stratosphere as suggested by Miyazaki et al., (2005) with MRI GCM. Our results highlight that the method by which the large-scale flow is specified and more specifically the choice of the reanalysis fields, the relaxation timescale and the vertical grid (pressure level versus model level) in which the nudging is applied needs to be better understood and evaluated for their influence on the stratospheric circulation. Discrepancies between the vertical grid of the models and the reanalysis pressure levels they are interpolated onto or unbalanced dynamics are possible explanations for the differences found between the directly inferred circulation and that diagnosed from the wave forcing in the nudged simulations. Nudging would either violate continuity, or if continuity is maintained, it will come at the expense of the vertical fluxes, which are not nudged. The interesting aspect here seems to be that this results in substantial change to the net fluxes across a range of timescales, i.e. it does not only increase numerical noise in the $\overline{w}^{*}$ component.

In order to reduce discrepancies between nudged and free-running simulations, various nudging techniques have been investigated. The role of gravity waves in the error growth that the nudging introduces over time has been highlighted for a single model (Smith et al., 2017). Constraining just the horizontal winds without the temperature was found to be a good strategy when investigating the aerosol indirect effects without affecting significantly the mean state (Zhang et al., 2014). The relaxation timescale when applying the nudging has been found to play an important role in single model studies (Merryfield et al., 2013), but there is no general consensus for the value of the relaxation constant, which is model-specific for the simulations considered here (Morgenstern et al., 2017). Given the varying implementations of nudging in the models analysed here, our study is ill-suited to investigate in detail the mechanisms for how nudging affects the residual circulation. A dedicated study of the sensitivities within one model to relaxation timescales, nudging parameters, nudging height range, and vertical resolution  would help to offer a detailed explanation for these differences.

The large spread in climatological residual circulation in nudged CCM simulations is an important limitation for those wishing to use them to examine tracer transport, for example stratospheric ozone trends (Solomon et al., 2016), volcanic aerosols (Schmidt et al., 2018), and diagnostics for age-of-air (Dietmüller et al., 2018). Despite the limitations for transport within the stratosphere described here, some success has been reported in studies that used nudged simulations to investigate specific meteorological events such as Sudden Stratospheric Warmings, and in particular for exploring processes beyond the top of the nudging region in the Mesosphere-Lower Thermosphere (e.g., Tweedy et al., 2013; Chandran and Collins, 2014; Pedatella et al., 2014). In conclusion, owing to the limitations of the current techniques for nudging models highlighted here, we urge caution in drawing quantitative

comparisons of stratospheric tracers affected by the residual circulation in nudged  simulations against stratospheric observational data.

*Data availability*

The majority of CCMI-1 used in this study can be obtained through the British Atmospheric Data Centre (BADC) archive (ftp://ftp.ceda.ac.uk, last access: June 2018). For instructions for access to the archive, see http://blogs.reading.ac.uk/ccmi/badc-data-access, last access: July 2019). The correctly calculated EMAC-L47MA/EMAC-L90MA TEM model output for both REF-C1 and REF-C1SD were obtained directly from Hella Garny. The SOCOL3 REF-C1SD TEM model output was obtained from Andrea Stenke. The S-RIP data used in this study can be obtained through the through the British Atmospheric Data Centre (BADC) archive (ftp://ftp.ceda.ac.uk, last access: September 2018), see https://catalogue.ceda.ac.uk/uuid/b241a7f536a244749662360bd7839312

*Author contributions*

AC performed the analysis and wrote the article ACM and MPC  design  the study and made substantial contributions to the interpretation of the data. Moreover, they participated in drafting and revising the article. HG provided the correctly calculated EMAC data and SD contributed to the discussion  of the content. AS provided the SOCOL REF-C1SD data. The other co-authors contributed information pertaining to their individual models and helped edit  the paper.

*Competing Interests*

The authors declare that they have no conflict of interest

*Acknowledgments*

[revised manuscript text omitted]

Froidevaux, L., Kinnison, D. E., Wang, R., Anderson, J. and Fuller, R. A.: Evaluation of CESM1 (WACCM) free-running and specified dynamics atmospheric composition simulations using global

1200     multispecies satellite data records, Atmos. Chem. Phys., 19(7), 4783–4821, doi:10.5194/acp-19-4783-2019, 2019.

García-Herrera, R., Calvo, N., Garcia, R. R. and Giorgetta, M. A.: Propagation of ENSO temperature signals into the middle atmosphere: A comparison of two general circulation models and ERA-40 reanalysis data, J. Geophys. Res. Atmos., 111(6), 1–14, doi:10.1029/2005JD006061, 2006.

1205   Garcia, R. R. and Boville, B. A.: "Downward Control" of the Mean Meridional Circulation and Temperature Distribution of the Polar Winter Stratosphere, J. Atmos. Sci., 51(15), 2238–2245, doi:10.1175/1520-0469(1994)051<2238:cotmmc>2.0.co;2, 2002.

Garcia, R. R., Randel, W. J. and Kinnison, D. E.: On the Determination of Age of Air Trends from Atmospheric Trace Species, J. Atmos. Sci., 68(1), 139–154, doi:10.1175/2010jas3527.1, 2010.

1210   Garcia, R. R., Smith, A. K., Kinnison, D. E., Cámara, Á. de la and Murphy, D. J.: Modification of the Gravity Wave Parameterization in the Whole Atmosphere Community Climate Model: Motivation and Results, J. Atmos. Sci., 74(1), 275–291, doi:10.1175/jas-d-16-0104.1, 2016.

Garny, H., Birner, T., Bönisch, H. and Bunzel, F.: The effects of mixing on age of air, J. Geophys. Res., 119(12), 7015–7034, doi:10.1002/2013JD021417, 2014.

1215   Haenel, F. J., Stiller, G. P., Von Clarmann, T., Funke, B., Eckert, E., Glatthor, N., Grabowski, U., Kellmann, S., Kiefer, M., Linden, A. and Reddmann, T.: Reassessment of MIPAS age of air trends and variability, Atmos. Chem. Phys., 15(22), 13161–13176, doi:10.5194/acp-15-13161-2015, 2015.

Hardiman, S. C., Butchart, N. and Calvo, N.: The morphology of the Brewer-Dobson circulation and its response to climate change in CMIP5 simulations, Q. J. R. Meteorol. Soc., 140(683), 1958–1965,

1220     doi:10.1002/qj.2258, 2014.

[revised manuscript text omitted]

J. Geophys. Res. D Atmos., 109(24), 1–18, doi:10.1029/2004JD005093, 2004.

Jung, T., Miller, M. J. and Palmer, T. N.: Diagnosing the Origin of Extended-Range Forecast Errors, Mon. Weather Rev., 138(6), 2434–2446, doi:10.1175/2010mwr3255.1, 2010.

Karpechko, A.Yu and A.C. Maycock (Lead Authors), M. Abalos, H. Akiyoshi, J.M. Arblaster, C.I. Garfinkel, K.H. Rosenlof and M. Sigmond , Stratospheric Ozone Changes and Climate, Chapter 5 in Scientific Assessment of Ozone Depletion: 2018, Global Ozone Rese., 2018.

Keeble, J., Bednarz, E. M., Banerjee, A., Luke Abraham, N., Harris, N. R. P., Maycock, A. C. and Pyle, J. A.: Diagnosing the radiative and chemical contributions to future changes in tropical column ozone with the UM-UKCA chemistry-climate model, Atmos. Chem. Phys., 17(22), 13801–13818, doi:10.5194/acp-17-13801-2017, 2017.

Kida, H.: General Circulation of Air Parcels and Transport Characteristics Derived from a hemispheric GCM, J. Meteorol. Soc. Japan. Ser. II, 61(2), 171–187, doi:10.2151/jmsj1965.61.2_171, 1983.

Kobayashi, C. and Iwasaki, T.: Brewer-Dobson circulation diagnosed from JRA-55, J. Geophys. Res., 121(4), 1493–1510, doi:10.1002/2015JD023476, 2016.

Kobayashi, S., Ota, Y., Harada, Y., Ebita, A., Moriya, M., Onoda, H., Onogi, K., Kamahori, H., Kobayashi, C., Endo, H., Miyaoka, K. and Takahashi, K.: The JRA-55 Reanalysis: General Specifications and Basic Characteristics, J. Meteorol. Soc. Japan. Ser. II, 93(1), 5–48, doi:10.2151/jmsj.2015-001, 2015.

Krol, M., De Bruine, M., Killaars, L., Ouwersloot, H., Pozzer, A., Yin, Y., Chevallier, F., Bousquet, P., Patra, P., Belikov, D., Maksyutov, S., Dhomse, S., Feng, W. and Chipperfield, M. P.: Age of air as a diagnostic for transport timescales in global models, Geosci. Model Dev., 11(8), 3109–3130, doi:10.5194/gmd-11-3109-2018, 2018.

Lamarque, J. F., Emmons, L. K., Hess, P. G., Kinnison, D. E., Tilmes, S., Vitt, F., Heald, C. L., Holland, E. A., Lauritzen, P. H., Neu, J., Orlando, J. J., Rasch, P. J. and Tyndall, G. K.: CAM-chem: Description and evaluation of interactive atmospheric chemistry in the Community Earth System Model, Geosci. Model Dev., 5(2), 369–411, doi:10.5194/gmd-5-369-2012, 2012.

Lamarque, J. F., Shindell, D. T., Josse, B., Young, P. J., Cionni, I., Eyring, V., Bergmann, D., Cameron-Smith, P., Collins, W. J., Doherty, R., Dalsoren, S., Faluvegi, G., Folberth, G., Ghan, S. J., Horowitz, L. W., Lee, Y. H., MacKenzie, I. A., Nagashima, T., Naik, V., Plummer, D., Righi, M., Rumbold, S. T., Schulz, M., Skeie, R. B., Stevenson, D. S., Strode, S., Sudo, K., Szopa, S., Voulgarakis, A. and Zeng, G.: The atmospheric chemistry and climate model intercomparison Project (ACCMIP): Overview and description of models, simulations and climate diagnostics, Geosci. Model Dev., 6(1), 179–206, doi:10.5194/gmd-6-179-2013, 2013.

Lefevre, F., Brasseur, G. P., Folkins, I., Smith, A. K. and Simon, P.: Chemistry of the 1991-1992 stratospheric winter: three-dimensional model simulations, J. Geophys. Res., 99(D4), 8183–8195, doi:10.1029/93JD03476, 1994.

Li, F., Newman, P., Pawson, S. and Perlwitz, J.: Effects of Greenhouse Gas Increase and Stratospheric Ozone Depletion on Stratospheric Mean Age of Air in 1960–2010, J. Geophys. Res. Atmos., 123(4), 2098–2110, doi:10.1002/2017JD027562, 2018.

Linz, M., Plumb, R. A., Gerber, E. P., Haenel, F. J., Stiller, G., Kinnison, D. E., Ming, A. and Neu, J. L.: The strength of the meridional overturning circulation of the stratosphere, Nat. Geosci., 10(9), 663–667, doi:10.1038/ngeo3013, 2017.

Linz, M., Abalos, M., Sasha Glanville, A., Kinnison, D. E., Ming, A. and Neu, J. L.: The global diabatic circulation of the stratosphere as a metric for the brewer-dobson circulation, Atmos. Chem. Phys., 19(7), 5069–5090, doi:10.5194/acp-19-5069-2019, 2019.

Löffler, M., Brinkop, S. and Jöckel, P.: Impact of major volcanic eruptions on stratospheric water vapour, Atmos. Chem. Phys., 16(10), 6547–6562, doi:10.5194/acp-16-6547-2016, 2016.

Mahieu, E., Chipperfield, M. P., Notholt, J., Reddmann, T., Anderson, J., Bernath, P. F., Blumenstock, T., Coffey, M. T., Dhomse, S. S., Feng, W., Franco, B., Froidevaux, L., Griffith, D. W. T., Hannigan, J. W., Hase, F., Hossaini, R., Jones, N. B., Morino, I., Murata, I., Nakajima, H., Palm, M., Paton-Walsh, C., Russell, J. M., Schneider, M., Servais, C., Smale, D. and Walker, K. A.: Recent Northern Hemisphere stratospheric HCl increase due to atmospheric circulation changes, Nature, 515(7525), 104–107, doi:10.1038/nature13857, 2014.

Marsh, D. R. and Garcia, R. R.: Attribution of decadal variability in lower-stratospheric tropical ozone, Geophys. Res. Lett., 34(21), 1–5, doi:10.1029/2007GL030935, 2007.

Marsh, D. R., Mills, M. J., Kinnison, D. E., Lamarque, J. F., Calvo, N. and Polvani, L. M.: Climate change from 1850 to 2005 simulated in CESM1(WACCM), J. Clim., 26(19), 7372–7391, doi:10.1175/JCLI-D-12-00558.1, 2013.

Martineau, P.: S-RIP: Zonal-mean dynamical variables of global atmospheric reanalyses on pressure levels. Centre for Environmental Data Analysis, , doi:10.5285/b241a7f536a244749662360bd7839312, 2017.

Martineau, P., Wright, J. S., Zhu, N. and Fujiwara, M.: Zonal-mean data set of global atmospheric reanalyses on pressure levels, Earth Syst. Sci. Data, 10(4), 1925–1941, doi:10.5194/essd-10-1925-2018, 2018.

Maycock, A. C., Randel, W. J., Steiner, A. K., Karpechko, A. Y., Cristy, J., Saunders, R., Thompson, D. W. J., Zou, C.-Z., Chrysanthou, A., Abraham, N. L., Akiyoshi, H., Archibald, A. T., Butchart, N., Chipperfield, M., Dameris, M., Deushi, M., Dhomse, S., Di Genova, G., Jöckel, P., Kinnison, D. E., Kirner, O., Ladstädter, F., Michou, M., Morgenstern, O., O'Connor, F., Oman, L., Pitari, G., Plummer, D. A., Revell, L. E., Rozanov, E., Stenke, A., Visioni, D., Yamashita, Y. and Zeng, G.: Revisiting the mystery of recent stratospheric temperature trends, Geophys. Res. Lett., 1–15, doi:10.1029/2018GL078035, 2018.

McLandress, C., Scinocca, J. F., Shepherd, T. G., Reader, M. C. and Manney, G. L.: Dynamical Control

of the Mesosphere by Orographic and Nonorographic Gravity Wave Drag during the Extended Northern Winters of 2006 and 2009, J. Atmos. Sci., 70(7), 2152–2169, doi:10.1175/jas-d-12-0297.1, 2012.

1345 Merryfield, W. J., Lee, W. S., Boer, G. J., Kharin, V. V., Scinocca, J. F., Flato, G. M., Ajayamohan, R. S., Fyfe, J. C., Tang, Y. and Polavarapu, S.: The canadian seasonal to interannual prediction system. part I: Models and initialization, Mon. Weather Rev., 141(8), 2910–2945, doi:10.1175/MWR-D-12-00216.1, 2013.

Ming, A., Hitchcock, P. and Haynes, P.: The Double Peak in Upwelling and Heating in the Tropical 1350 Lower Stratosphere, J. Atmos. Sci., 73(5), 1889–1901, doi:10.1175/jas-d-15-0293.1, 2016a.

Ming, A., Hitchcock, P. and Haynes, P.: The Response of the Lower Stratosphere to Zonally Symmetric Thermal and Mechanical Forcing, J. Atmos. Sci., 73(5), 1903–1922, doi:10.1175/jas-d-15-0294.1, 2016b.

Miyazaki, K., Iwasaki, T., Shibata, K., Deushi, M. and Sekiyama, T. T.: The Impact of Changing 1355 Meteorological Variables to Be Assimilated into GCM on Ozone Simulation with MRI CTM, J. Meteorol. Soc. Japan. Ser. II, 83(5), 909–918, doi:10.2151/jmsj.83.909, 2005.

Miyazaki, K., Iwasaki, T., Kawatani, Y., Kobayashi, C., Sugawara, S. and Hegglin, M. I.: Inter-comparison of stratospheric mean-meridional circulation and eddy mixing among six reanalysis data sets, Atmos. Chem. Phys., 16(10), 6131–6152, doi:10.5194/acp-16-6131-2016, 2016.

1360 Molod, A., Takacs, L., Suarez, M., Bacmeister, J., Song, I.-S. and Eichmann, A.: The GEOS-5 Atmospheric General Circulation Model: Mean Climate and Development from MERRA to Fortuna. [online] Available from: https://ntrs.nasa.gov/search.jsp?R=20120011790, 2012.

Molod, A., Takacs, L., Suarez, M. and Bacmeister, J.: Development of the GEOS-5 atmospheric general circulation model: Evolution from MERRA to MERRA2, Geosci. Model Dev., 8(5), 1339–1356, 1365 doi:10.5194/gmd-8-1339-2015, 2015.

Monge-Sanz, B. M., Chipperfield, M. P., Dee, D. P., Simmonsc, A. J. and Uppalac, S. M.: Improvements in the stratospheric transport achieved by a chemistry transport model with ECMWF (re)analyses: Identifying effects and remaining challenges, Q. J. R. Meteorol. Soc., 139(672), 654–673, doi:10.1002/qj.1996, 2013a.

1370 Monge-Sanz, B. M., Chipperfield, M. P., Untch, A., Morcrette, J. J., Rap, A. and Simmons, A. J.: On the uses of a new linear scheme for stratospheric methane in global models: Water source, transport tracer and radiative forcing, Atmos. Chem. Phys., 13(18), 9641–9660, doi:10.5194/acp-13-9641-2013, 2013b.

Moore, F. L., Ray, E. A., Rosenlof, K. H., Elkins, J. W., Tans, P., Karion, A. and Sweeney, C.: A cost-1375 effective trace gas measurement program for long-term monitoring of the stratospheric circulation, Bull. Am. Meteorol. Soc., 95(1), 147–155, doi:10.1175/BAMS-D-12-00153.1, 2014.

Morgenstern, O., Braesicke, P., O'Connor, F. M., Bushell, A. C., Johnson, C. E., Osprey, S. M. and

Pyle, J. A.: Evaluation of the new UKCA climate-composition model-Part 1: The stratosphere, Geosci. Model Dev., 2(1), 43–57, doi:10.5194/gmd-2-43-2009, 2009.

1380    Morgenstern, O., Zeng, G., Abraham, N. L., Telford, P. J., Braesicke, P., Pyle, J. A., Hardiman, S. C., O'connor, F. M. and Johnson, C. E.: Impacts of climate change, ozone recovery, and increasing methane on surface ozone and the tropospheric oxidizing capacity, J. Geophys. Res. Atmos., 118(2), 1028–1041, doi:10.1029/2012JD018382, 2013.

Morgenstern, O., Hegglin, M., Rozanov, E., O'Connor, F., Luke Abraham, N., Akiyoshi, H., Archibald,
1385    A., Bekki, S., Butchart, N., Chipperfield, M., Deushi, M., Dhomse, S., Garcia, R., Hardiman, S., Horowitz, L., Jöckel, P., Josse, B., Kinnison, D., Lin, M., Mancini, E., Manyin, M., Marchand, M., Marécal, V., Michou, M., Oman, L., Pitari, G., Plummer, D., Revell, L., Saint-Martin, D., Schofield, R., Stenke, A., Stone, K., Sudo, K., Tanaka, T., Tilmes, S., Yamashita, Y., Yoshida, K. and Zeng, G.: Review of the global models used within phase 1 of the Chemistry-Climate Model Initiative (CCMI),
1390    Geosci. Model Dev., 10(2), 639–671, doi:10.5194/gmd-10-639-2017, 2017.

Morgenstern, O., Stone, K. A., Schofield, R., Akiyoshi, H., Yamashita, Y., Kinnison, D. E., Garcia, R. R., Sudo, K., Plummer, D. A., Scinocca, J., Oman, L. D., Manyin, M. E., Zeng, G., Rozanov, E., Stenke, A., Revell, L. E., Pitari, G., Mancini, E., DI Genova, G., Visioni, D., Dhomse, S. S. and Chipperfield, M. P.: Ozone sensitivity to varying greenhouse gases and ozone-depleting substances
1395    in CCMI-1 simulations, Atmos. Chem. Phys., 18(2), 1091–1114, doi:10.5194/acp-18-1091-2018, 2018.

Neu, J. L. and Plumb, R. A.: Age of air in a "leaky pipe" model of stratospheric transport, J. Geophys. Res. Atmos., 104(D16), 19243–19255, doi:10.1029/1999JD900251, 1999.

Oman, L. D., Ziemke, J. R., Douglass, A. R., Waugh, D. W., Lang, C., Rodriguez, J. M. and Nielsen, J.
1400    E.: The response of tropical tropospheric ozone to ENSO, Geophys. Res. Lett., 38(13), doi:10.1029/2011GL047865, 2011.

Oman, L. D., Douglass, A. R., Ziemke, J. R., Rodriguez, J. M., Waugh, D. W. and Nielsen, J. E.: The ozone response to enso in aura satellite measurements and a chemistry-climate simulation, J. Geophys. Res. Atmos., 118(2), 965–976, doi:10.1029/2012JD018546, 2013.

1405    Orbe, C., Yang, H., Waugh, D. W., Zeng, G., Morgenstern, O., Kinnison, D. E., Lamarque, J.-F., Tilmes, S., Plummer, D. A., Scinocca, J. F., Josse, B., Marecal, V., Jöckel, P., Oman, L. D., Strahan, S. E., Deushi, M., Tanaka, T. Y., Yoshida, K., Akiyoshi, H., Yamashita, Y., Stenke, A., Revell, L., Sukhodolov, T., Rozanov, E., Pitari, G., Visioni, D., Stone, K. A., Schofield, R. and Banerjee, A.: Large-scale tropospheric transport in the Chemistry–Climate Model Initiative (CCMI) simulations,
1410    Atmos. Chem. Phys., 18(10), 7217–7235, doi:10.5194/acp-18-7217-2018, 2018.

Pedatella, N. M., Fuller-Rowell, T., Wang, H., Jin, H., Miyoshi, Y., Fujiwara, H., Shinagawa, H., Liu, H.-L., Sassi, F., Schmidt, H., Matthias, V. and Goncharenko, L.: The neutral dynamics during the 2009 sudden stratosphere warming simulated by different whole atmosphere models, J. Geophys.

Res. Sp. Phys., 119(2), 1306–1324, doi:10.1002/2013JA019421, 2014.

Pitari, G. and Rizi, V.: Pitari, Rizi - 1993.pdf, J. Atmos. Sci., 50(19), 3260–3276, doi:10.1175/1520-0469(1993)050<3260:AEOTCA>2.0.CO;2, 1993.

Pitari, G., Aquila, V., Kravitz, B., Robock, A., Watanabe, S., Cionni, I., Luca, N., Genova, G., Mancini, E. and Tilmes, S.: Stratospheric ozone response to sulfate geoengineering: Results from the geoengineering model intercomparison project (GeoMip), J. Geophys. Res., 119(5), 2629–2653, doi:10.1002/2013JD020566, 2014.

Ploeger, F. and Birner, T.: Seasonal and inter-annual variability of lower stratospheric age of air spectra, Atmos. Chem. Phys., 16(15), 10195–10213, doi:10.5194/acp-16-10195-2016, 2016.

Ploeger, F., Abalos, M., Birner, T., Konopka, P., Legras, B., Müller, R. and Riese, M.: Quantifying the effects of mixing and residual circulation on trends of stratospheric mean age of air, Geophys. Res. Lett., 42(6), 2047–2054, doi:10.1002/2014GL062927, 2015a.

Ploeger, F., Riese, M., Haenel, F., Konopka, P., Müller, R. and Stiller, G.: Variability of stratospheric mean age of air and of the local effects of residual circulation and eddy mixing, J. Geophys. Res., 120(2), 716–733, doi:10.1002/2014JD022468, 2015b.

Ploeger, F., Legras, B., Charlesworth, E., Yan, X., Diallo, M., Konopka, P., Birner, T., Tao, M., Engel, A. and Riese, M.: How robust are stratospheric age of air trends from different reanalyses?, Atmos. Chem. Phys., 19(9), 6085–6105, doi:10.5194/acp-19-6085-2019, 2019.

PLUMB, R. A.: Stratospheric Transport, J. Meteorol. Soc. Japan. Ser. II, 80(4B), 793–809, doi:10.2151/jmsj.80.793, 2004.

Polvani, L. M., Abalos, M., Garcia, R., Kinnison, D. and Randel, W. J.: Significant Weakening of Brewer-Dobson Circulation Trends Over the 21st Century as a Consequence of the Montreal Protocol, Geophys. Res. Lett., 45(1), 401–409, doi:10.1002/2017GL075345, 2018.

Polvani, L. M., Wang, L., Abalos, M., Butchart, N., Chipperfield, M. P., Dameris, M., Deushi, M., Dhomse, S. S., Jöckel, P., Kinnison, D., Michou, M., Morgenstern, O., Oman, L. D., Plummer, D. A. and Stone, K. A.: Large impacts, past and future, of ozone-depleting substances on Brewer-Dobson circulation trends: A multi-model assessment, J. Geophys. Res. Atmos., 2018JD029516, doi:10.1029/2018JD029516, 2019.

Randel, W. J. and Thompson, A. M.: Interannual variability and trends in tropical ozone derived from SAGE II satellite data and SHADOZ ozonesondes, J. Geophys. Res. Atmos., 116(7), 1–9, doi:10.1029/2010JD015195, 2011.

Randel, W. J., Garcia, R. R., Calvo, N. and Marsh, D.: ENSO influence on zonal mean temperature and ozone in the tropical lower stratosphere, Geophys. Res. Lett., 36(15), 1–5, doi:10.1029/2009GL039343, 2009.

Ray, E. A., Moore, F. L., Rosenlof, K. H., Plummer, D. A., Kolonjari, F. and Walker, K. A.: An idealized stratospheric model useful for understanding differences between long-lived trace gas

1450 measurements and global chemistry-climate model output, J. Geophys. Res., 121(10), 5356–5367, doi:10.1002/2015JD024447, 2016.

Revell, L. E., Tummon, F., Stenke, A., Sukhodolov, T., Coulon, A., Rozanov, E., Garny, H., Grewe, V. and Peter, T.: Drivers of the tropospheric ozone budget throughout the 21st century under the medium-high climate scenario RCP 6.0, Atmos. Chem. Phys., 15(10), 5887–5902, doi:10.5194/acp-

1455 15-5887-2015, 2015a.

Revell, L. E., Tummon, F., Salawitch, R. J., Stenke, A. and Peter, T.: The changing ozone depletion potential of N2O in a future climate, Geophys. Res. Lett., 42(22), 10047–10055, doi:10.1002/2015GL065702, 2015b.

Richter, J. H., Sassi, F. and Garcia, R. R.: Toward a Physically Based Gravity Wave Source

1460 Parameterization in a General Circulation Model, J. Atmos. Sci., 67(1), 136–156, doi:10.1175/2009jas3112.1, 2010.

Rienecker, M. M., Suarez, M. J., Gelaro, R., Todling, R., Bacmeister, J., Liu, E., Bosilovich, M. G., Schubert, S. D., Takacs, L., Kim, G. K., Bloom, S., Chen, J., Collins, D., Conaty, A., Da Silva, A., Gu, W., Joiner, J., Koster, R. D., Lucchesi, R., Molod, A., Owens, T., Pawson, S., Pegion, P.,

1465 Redder, C. R., Reichle, R., Robertson, F. R., Ruddick, A. G., Sienkiewicz, M. and Woollen, J.: MERRA: NASA's modern-era retrospective analysis for research and applications, J. Clim., 24(14), 3624–3648, doi:10.1175/JCLI-D-11-00015.1, 2011.

Rood, R. B., Allen, D. J., Baker, W. E., Lamich, D. J. and Kaye, J. A.: The Use of Assimilated Stratospheric Data in Constituent Transport Calculations, J. Atmos. Sci., 46(5), 687–702,

1470 doi:10.1175/1520-0469(1989)046<0687:tuoasd>2.0.co;2, 2002.

Rosenlof, K. H.: Seasonal cycle of the residual mean meridional circulation in the stratosphere, J. Geophys. Res., 100(D3), 5173–5191, doi:10.1029/94JD03122, 1995.

Šácha, P., Eichinger, R., Garny, H., Pišoft, P., Dietmüller, S., de la Torre, L., Plummer, D. A., Jöckel, P., Morgenstern, O., Zeng, G., Butchart, N. and Añel, J. A.: Extratropical age of air trends and

1475 causative factors in climate projection simulations, Atmos. Chem. Phys., 19(11), 7627–7647, doi:10.5194/acp-19-7627-2019, 2019.

Scaife, A. A., Butchart, N., Warner, C. D. and Swinbank, R.: Impact of a Spectral Gravity Wave Parameterization on the Stratosphere in the Met Office Unified Model, J. Atmos. Sci., 59(9), 1473–1489, doi:10.1175/1520-0469(2002)059<1473:ioasgw>2.0.co;2, 2002.

1480 Schmidt, A., Mills, M. J., Ghan, S., Gregory, J. M., Allan, R. P., Andrews, T., Bardeen, C. G., Conley, A., Forster, P. M., Gettelman, A., Portmann, R. W., Solomon, S. and Toon, O. B.: Volcanic Radiative Forcing From 1979 to 2015, J. Geophys. Res. Atmos., 123(22), 12,491-12,508, doi:10.1029/2018JD028776, 2018.

Schmidt, U. and Khedim, A.: In situ measurements of carbon dioxide in the winter Arctic vortex and at

1485 midlatitudes: An indicator of the 'age' of stratospheric air, Geophys. Res. Lett., 18(4), 763–766,

doi:10.1029/91GL00022, 1991.

Scinocca, J. F.: An Accurate Spectral Nonorographic Gravity Wave Drag Parameterization for General Circulation Models, J. Atmos. Sci., 60(4), 667–682, doi:10.1175/1520-0469(2003)060<0667:aasngw>2.0.co;2, 2003.

1490 Scinocca, J. F., McFarlane, N. A., Lazare, M., Li, J. and Plummer, D.: Technical note: The CCCma third generation AGCM and its extension into the middle atmosphere, Atmos. Chem. Phys., 8(23), 7055–7074, doi:10.5194/acp-8-7055-2008, 2008.

Seviour, W. J. M., Butchart, N. and Hardiman, S. C.: The Brewer-Dobson circulation inferred from ERA-Interim, Q. J. R. Meteorol. Soc., 138(665), 878–888, doi:10.1002/qj.966, 2012.

1495 Smith, A. K., Pedatella, N. M., Marsh, D. R. and Matsuo, T.: On the Dynamical Control of the Mesosphere–Lower Thermosphere by the Lower and Middle Atmosphere, J. Atmos. Sci., 74(3), 933–947, doi:10.1175/jas-d-16-0226.1, 2017.

Solomon, S., Kinnison, D., Bandoro, J. and Garcia, R.: Simulation of polar ozone depletion: An update, J. Geophys. Res., 120(15), 7958–7974, doi:10.1002/2015JD023365, 2015.

1500 Solomon, S., Kinnison, D., Garcia, R. R., Bandoro, J., Mills, M., Wilka, C., Neely, R. R., Schmidt, A., Barnes, J. E., Vernier, J. P. and Höpfner, M.: Monsoon circulations and tropical heterogeneous chlorine chemistry in the stratosphere, Geophys. Res. Lett., 43(24), 12,624-12,633, doi:10.1002/2016GL071778, 2016.

SPARC: SPARC CCMVal Report on the Evaluation of Chemistry-Climate Models. V. Eyring, T. Shepherd and D. Waugh (Eds.), SPARC Rep. No. 5, WCRP-30/2010,WMO/TD-No.40, available at www.sparc-climate.org/publications/sp [online] Available from: http://www.sparc-climate.org/publications/sparc-reports/sparc-report-no5/, 2010.

Stenke, A., Schraner, M., Rozanov, E., Egorova, T., Luo, B. and Peter, T.: The SOCOL version 3.0 chemistry-climate model: Description, evaluation, and implications from an advanced transport algorithm, Geosci. Model Dev., 6(5), 1407–1427, doi:10.5194/gmd-6-1407-2013, 2013.

Stiller, G. P., Von Clarmann, T., Haenel, F., Funke, B., Glatthor, N., Grabowski, U., Kellmann, S., Kiefer, M., Linden, A., Lossow, S. and López-Puertas, M.: Observed temporal evolution of global mean age of stratospheric air for the 2002 to 2010 period, Atmos. Chem. Phys., 12(7), 3311–3331, doi:10.5194/acp-12-3311-2012, 2012.

1515 Stone, K. A., Morgenstern, O., Karoly, D. J., Klekociuk, A. R., French, W. J., Abraham, N. L. and Schofield, R.: Evaluation of the ACCESS - Chemistry-climate model for the Southern Hemisphere, Atmos. Chem. Phys., 16(4), 2401–2415, doi:10.5194/acp-16-2401-2016, 2016.

Telford, P. J., Braesicke, P., Morgenstern, O. and Pyle, J. A.: Technical note: Description and assessment of a nudged version of the new dynamics Unified Model, Atmos. Chem. Phys., 8(6), 1701–1712, doi:10.5194/acp-8-1701-2008, 2008.

Tweedy, O. V., Limpasuvan, V., Orsolini, Y. J., Smith, A. K., Garcia, R. R., Kinnison, D., Randall, C.

E., Kvissel, O. K., Stordal, F., Harvey, V. L. and Chandran, A.: Nighttime secondary ozone layer during major stratospheric sudden warmings in specified-dynamics WACCM, J. Geophys. Res. Atmos., 118(15), 8346–8358, doi:10.1002/jgrd.50651, 2013.

1525 UCAR/NCAR/CISL/TDD: The NCAR Command Language (NCL), Version 6.3.0, , doi:10.5065/D6WD3XH5, 2015.

[revised manuscript text omitted]

**REFC1SD Annual means DCP MLR Analysis**

[Figure]

**Figure 11.** Timeseries for REF-C1SD simulations of the components of the annual mean tropical upward mass flux [$\times 10^9$ kg s$^{-1}$] attributed to (a) volcanic aerosol, (b) ENSO, (c) linear trend, (d, e)  QBO, and (f) regression  residuals .

**Start/End Date Trend Sensitivity REFC1**

[Figure]

**Figure 12.** 70 hPa tropical upward mass flux trends [×10⁹ kg s⁻¹ decade⁻¹] for different start (abscissa) and end (ordinate) dates  over the period 1980-2009 for the REF-C1 (r1i1p1) simulations. Trends are not shown for periods of less than 10 years. Values with statistical significance greater than the 95% level are stippled.

**Start/End Date Trend Sensitivity REFC1SD**

[Figure]

**Figure 13.** As in Figure 12 but for the REF-C1SD simulations.

Mass flux linear trend – partial regressor from the MLR sensitivity plots (values with statistical significance are stippled) over period 1980-2009 for REF-C1SD simulations.

Mass flux linear trend – partial regressor from the MLR sensitivity plots (values with statistical significance are stippled) over period 1980-2009 for REF-C1SD simulations.